# HARNESSING DENSITY RATIOS FOR ONLINE REINFORCEMENT LEARNING

**Philip Amortila**[*]  **Dylan J. Foster**  **Nan Jiang**  **Ayush Sekhari**  **Tengyang Xie**
UIUC  Microsoft Research  UIUC  MIT  Microsoft Research

## ABSTRACT

The theories of offline and online reinforcement learning, despite having evolved in parallel, have begun to show signs of the possibility for a unification, with algorithms and analysis techniques for one setting often having natural counterparts in the other. However, the notion of *density ratio modeling*, an emerging paradigm in offline RL, has been largely absent from online RL, perhaps for good reason: the very existence and boundedness of density ratios relies on access to an exploratory dataset with good coverage, but the core challenge in online RL is to collect such a dataset without having one to start.

In this work we show—perhaps surprisingly—that density ratio-based algorithms have online counterparts. Assuming only the existence of an exploratory distribution with good coverage, a structural condition known as *coverability* (Xie et al., 2023), we give a new algorithm (GLOW) that uses density ratio realizability and value function realizability to perform sample-efficient online exploration. GLOW addresses unbounded density ratios via careful use of *truncation*, and combines this with optimism to guide exploration. GLOW is computationally inefficient; we complement it with a more efficient counterpart, HYGLOW, for the *Hybrid RL* setting (Song et al., 2023) wherein online RL is augmented with additional offline data. HYGLOW is derived as a special case of a more general meta-algorithm that provides a provable black-box reduction from hybrid RL to offline RL, which may be of independent interest.

## 1 INTRODUCTION

A fundamental problem in reinforcement learning (RL) is to understand what modeling assumptions and algorithmic principles lead to sample-efficient learning guarantees. Investigation into algorithms for sample-efficient reinforcement learning has primarily focused on two separate formulations: *Offline reinforcement learning*, where a learner must optimize a policy from logged transitions and rewards, and *online reinforcement learning*, where the learner can gather new data by interacting with the environment; both formulations share the common goal of learning a near-optimal policy. For the most part, the bodies of research on offline and online reinforcement have evolved in parallel, but they exhibit a number of curious similarities. Algorithmically, many design principles for offline RL (e.g., pessimism) have online counterparts (e.g., optimism), and statistically efficient algorithms for both frameworks typically require similar *representation conditions* (e.g., ability to model state-action value functions). Yet, the frameworks have notable differences: online RL algorithms require *exploration conditions* to address the issue of distribution shift (Russo and Van Roy, 2013; Jiang et al., 2017; Sun et al., 2019; Wang et al., 2020c; Du et al., 2021; Jin et al., 2021a; Foster et al., 2021), while offline RL algorithms require conceptually distinct *coverage conditions* to ensure the data logging distribution sufficiently covers the state space (Munos, 2003; Antos et al., 2008; Chen and Jiang, 2019; Xie and Jiang, 2020, 2021; Jin et al., 2021b; Rashidinejad et al., 2021; Foster et al., 2022; Zhan et al., 2022).

Recently, Xie et al. (2023) exposed a deeper connection between online and offline RL by showing that *coverability*—that is, *existence* of a data distribution with good coverage for offline RL—is itself a sufficient condition that enables sample-efficient exploration in online RL, even when the learner has no prior knowledge of said distribution. This suggests the possibility of a theoretical unification of online and offline RL, but the picture remains incomplete, and there are many gaps

---

[*]Authors listed in alphabetical order. Full version appears at [arXiv:2401.09681].

in our understanding. Notably, a promising emerging paradigm in offline RL makes use of the ability to model *density ratios* (also referred to as *marginalized importance weights* or simply *weight functions*) for the underlying MDP. Density ratio modeling offers an alternative to classical value function approximation (or, approximate dynamic programming) methods (Munos, 2007; Munos and Szepesvári, 2008; Chen and Jiang, 2019), as it avoids instability and typically succeeds under weaker representation conditions (requiring only realizability conditions as opposed to Bellman completeness-type assumptions). Yet despite extensive investigation into density ratio methods for offline RL—both in theory (Liu et al., 2018; Uehara et al., 2020; Yang et al., 2020; Uehara et al., 2021; Jiang and Huang, 2020; Xie and Jiang, 2020; Zhan et al., 2022; Chen and Jiang, 2022; Rashidinejad et al., 2023; Ozdaglar et al., 2023) and practice (Nachum et al., 2019; Kostrikov et al., 2019; Nachum and Dai, 2020; Zhang et al., 2020; Lee et al., 2021)—density ratio modeling has been conspicuously absent in the *online* reinforcement learning. This leads us to ask:

*Can online reinforcement learning benefit from the ability to model density ratios?*

Adapting density ratio-based methods to the online setting with provable guarantees presents a number of conceptual and technical challenges. First, since the data distribution in online RL is constantly changing, it is unclear *what* densities one should even attempt to model. Second, most offline reinforcement learning algorithms require relatively stringent notions of coverage for the data distribution. In online RL, it is unreasonable to expect data gathered early in the learning process to have good coverage, and naive algorithms may cycle or fail to explore as a result. As such, it may not be reasonable to expect density ratio modeling to benefit online RL in the same fashion as offline.

**Our contributions.** We show that in spite of these challenges, density ratio modeling enables guarantees for online reinforcement learning that were previously out of reach.

- **Density ratios for online RL.** We show (Section 3) that for any MDP with low *coverability*, density ratio realizability and realizability of the optimal state-action value function are sufficient for sample-efficient online RL. This result is obtained through a new algorithm, GLOW, which addresses the issue of distribution shift via careful use of *truncated* density ratios, which it combines with optimism to drive exploration. This complements Xie et al. (2023), who gave guarantees for coverable MDPs under a stronger Bellman completeness assumption for the value function class.

- **Density ratios for hybrid RL.** Our algorithm for online RL is computationally inefficient. We complement it (Section 4) with a more efficient counterpart, HYGLOW, for the *hybrid RL* framework (Song et al., 2023), where the learner has access to additional offline data that covers a high-quality policy.

- **Hybrid-to-offline reductions.** To achieve the result above, we investigate a broader question: when can offline RL algorithms be adapted *as-is* to online settings? We provide a new meta-algorithm, $H_2O$, which reduces hybrid RL to offline RL by repeatedly calling a given offline RL algorithm as a black box. We show that $H_2O$ enjoys low regret whenever the black-box offline algorithm satisfies certain conditions, and demonstrate that these conditions are satisfied by a range of existing offline algorithms, thus lifting them to the hybrid RL setting.

While our results are theoretical in nature, we are optimistic that they will lead to further investigation into the power of density ratio modeling in online RL and inspire practical algorithms.

## 1.1 PRELIMINARIES

**Markov Decision Processes.** We consider an episodic reinforcement learning setting. A Markov Decision Process (MDP) is a tuple $\mathcal{M} = (\mathcal{X}, \mathcal{A}, P, R, H, d_1)$, where $\mathcal{X}$ is the (large/potentially infinite) state space, $\mathcal{A}$ is the action space, $H \in \mathbb{N}$ is the horizon, $R = \{R_h\}_{h=1}^H$ is the reward function (where $R_h : \mathcal{X} \times \mathcal{A} \to \Delta([0,1])$), $P = \{P_h\}_{h \leq 1}$ is the transition distribution (where $P_h : \mathcal{X} \times \mathcal{A} \to \Delta(\mathcal{X})$), and $d_1$ is the initial state distribution. A randomized policy is a sequence of functions $\pi = \{\pi_h : \mathcal{X} \to \Delta(\mathcal{A})\}_{h=1}^H$. When a policy is executed, it generates a trajectory $(x_1, a_1, r_1), \ldots, (x_H, a_H, r_H)$ via the process $a_h \sim \pi_h(x_h), r_h \sim R_h(x_h, a_h), x_{h+1} \sim P_h(x_h, a_h)$, initialized at $x_1 \sim d_1$ (we use $x_{H+1}$ to denote a deterministic terminal state with zero reward). We write $\mathbb{P}^\pi[\cdot]$ and $\mathbb{E}^\pi[\cdot]$ to denote the law and corresponding expectation for the trajectory under this process.

For a policy $\pi$, the expected reward for is given by $J(\pi) := \mathbb{E}^\pi\left[\sum_{h=1}^H r_h\right]$, and the value functions given by $V_h^\pi(x) := \mathbb{E}^\pi\left[\sum_{h'=h}^H r_{h'} \mid x_h = x\right]$, and $Q_h^\pi(x, a) := \mathbb{E}^\pi\left[\sum_{h'=h}^H r_{h'} \mid x_h = x, a_h = a\right]$. We write $\pi^\star = \{\pi_h^\star\}_{h=1}^H$ to denote an optimal deterministic policy, which maximizes $V^\pi$ at all

states. We let $\mathcal{T}_h$ denote the Bellman (optimality) operator for layer $h$, defined via $[\mathcal{T}_h f](x, a) = \mathbb{E}[r_h + \max_{a'} f(x_{h+1}, a') \mid x_h = x, a_h = a]$ for $f : \mathcal{X} \times \mathcal{A} \to \mathbb{R}$.

**Online RL.** In the online reinforcement learning framework, the learner repeatedly interacts with an unknown MDP by executing a policy and observing the resulting trajectory. The goal is to maximize total reward. Formally, the protocol proceeds in $N$ rounds, where at each round $t \in [N]$, the learner selects a policy $\pi^{(t)} = \{\pi_h^{(t)}\}_{h=1}^H$ in the (unknown) underlying MDP $M^\star$ and observes the trajectory $\{(x_h^{(t)}, a_h^{(t)}, r_h^{(t)})\}_{h=1}^H$. Our results are most naturally stated in terms of PAC guarantees. Here, after the $N$ rounds of interaction conclude, the learner can use all of the data collected to produce a final policy $\widehat{\pi}$, with the goal of minimizing

$$\textbf{Risk} := \mathbb{E}_{\widehat{\pi} \sim p}[J(\pi^\star) - J(\widehat{\pi})], \tag{1}$$

where $p \in \Delta(\Pi)$ denotes a distribution that the algorithm can use to randomize the final policy.

**Additional definitions and assumptions.** We assume that rewards are normalized such that $\sum_{h=1}^H r_h \in [0, 1]$ almost surely for all trajectories (Jiang and Agarwal, 2018; Wang et al., 2020a; Zhang et al., 2021; Jin et al., 2021a). To simplify presentation, we assume that $\mathcal{X}$ and $\mathcal{A}$ are countable; we expect that our results extend to handle continuous variables with an appropriate measure-theoretic treatment. We define the occupancy measure for policy $\pi$ via $d_h^\pi(x, a) := \mathbb{P}^\pi[x_h = x, a_h = a]$.

## 2 PROBLEM SETUP: DENSITY RATIO MODELING AND COVERABILITY

To investigate the power of density ratio modeling in online RL, we make use of *function approximation*, and aim to provide sample complexity guarantees with no explicit dependence on the size of the state space. We begin by appealing to *value function approximation*, a standard approach in online and offline reinforcement learning, and assume access to a value-function class $\mathcal{F} \subset (\mathcal{X} \times \mathcal{A} \times [H] \to [0, 1])$ that can realize the optimal value function $Q^\star$.

**Assumption 2.1** (Value function realizability). *We have $Q^\star \in \mathcal{F}$.*

For $f \in \mathcal{F}$, we define the greedy policy $\pi_f$ via $\pi_{f,h}(x) = \arg\max_a f_h(x, a)$, with ties broken in an arbitrary consistent fashion. For the remainder of the paper, we define $\Pi := \{\pi_f \mid f \in \mathcal{F}\}$ as the policy class induced by $\mathcal{F}$, unless otherwise specified.

**Density ratio modeling.** While value function approximation is a natural modeling approach, prior works in both online (Du et al., 2020; Weisz et al., 2021; Wang et al., 2021) and offline RL (Wang et al., 2020b; Zanette, 2021; Foster et al., 2022) have shown that value function realizability alone is not sufficient for statistically tractable learning in many settings. As such, value function approximation methods in online (Zanette et al., 2020; Jin et al., 2021a; Xie et al., 2023) and offline RL (Antos et al., 2008; Chen and Jiang, 2019) typically require additional representation conditions that may not be satisfied in practice, such as the stringent *Bellman completeness* assumption (i.e., $\mathcal{T}_h \mathcal{F}_{h+1} \subseteq \mathcal{F}_h$).

In offline RL, a promising emerging paradigm that goes beyond pure value function approximation is to model *density ratios* (or, marginalized important weights), which typically take the form $d_h^\pi(x,a)/\mu_h(x,a)$ for a policy $\pi$, where $\mu_h$ denotes the offline data distribution.[1] A recent line of work (Xie and Jiang, 2020; Jiang and Huang, 2020; Zhan et al., 2022) shows, that given access to a realizable value function class and a weight function class $\mathcal{W}$ that can realize the ratio $d_h^\pi/\mu_h$ (typically either for all policies $\pi$, or for the optimal policy $\pi^\star$), one can learn a near-optimal policy offline in a sample-efficient fashion; such results sidestep the need for stringent value function representation conditions like Bellman completeness. To explore whether density ratio modeling has similar benefits in online RL, we make the following assumption.

**Assumption 2.2** (Density ratio realizability). *The learner has access to a weight function class $\mathcal{W} \subset (\mathcal{X} \times \mathcal{A} \times [H] \to \mathbb{R}_+)$ such that for any policy pair $\pi, \pi' \in \Pi$, and $h \in [H]$, we have $w_h^{\pi;\pi'}(x, a) := d_h^\pi(x,a)/d_h^{\pi'}(x,a) \in \mathcal{W}_h$.*[2]

Assumption 2.2 *does not* assume that the density ratios under consideration are finite. That is, we do not assume boundedness of the weights, and our results do not pay for their range; our algorithm will only access certain *clipped* versions of the weight functions (Remark B.1).

---

[1]Formally, in offline reinforcement learning the learner does not interact with $M^\star$ but is given a dataset of tuples $(x_h, a_h, r_h, x_{h+1})$ collected i.i.d. according to $(x_h, a_h) \sim \mu_h, r_h \sim R_h(x_h, a_h), x_{h+1} \sim P_h(x_h, a_h)$.

[2]We adopt the convention that $x/0 = +\infty$ when $x > 0$ and $0/0 = 1$.

Compared to density ratio approaches in the offline setting, which typically require either realizability of $d_h^\pi/\mu_h$ for all $\pi \in \Pi$ or realizability of $d_h^{\pi^\star}/\mu_h$, where $\mu$ is the offline data distribution, we require realizability of $d_h^\pi/d_h^{\pi'}$ for all pairs of policies $\pi, \pi' \in \Pi$. This assumption is natural because it facilitates transfer between historical data (which is algorithm-dependent) and future policies. We refer the reader to Appendix B for a detailed comparison to alternative assumptions.

**Coverability.** In addition to realizability assumptions, online RL methods require *exploration conditions* (Russo and Van Roy, 2013; Jiang et al., 2017; Sun et al., 2019; Wang et al., 2020c; Du et al., 2021; Jin et al., 2021a; Foster et al., 2021) that allow deliberately designed algorithms to control distribution shift or extrapolate to unseen states. Towards lifting density ratio modeling from offline to online RL, we make use of *coverability* (Xie et al., 2023), an exploration condition inspired by the notion of coverage in the offline setting.

**Definition 2.1** (Coverability coefficient (Xie et al., 2023)). *The coverability coefficient $C_{\mathsf{cov}} > 0$ for a policy class $\Pi$ is given by $C_{\mathsf{cov}} := \inf_{\mu_1,\dots,\mu_H \in \Delta(\mathcal{X} \times \mathcal{A})} \sup_{\pi \in \Pi, h \in [H]} \|d_h^\pi/\mu_h\|_\infty$. We refer to the distribution $\mu_h^\star$ that attains the minimum for $h$ as the coverability distribution.*

Coverability is a structural property of the underlying MDP, and can be interpreted as the best value one can achieve for the *concentrability coefficient* $C_{\mathsf{conc}}(\mu) := \sup_{\pi \in \Pi, h \in [H]} \|d_h^\pi/\mu_h\|_\infty$ (a standard coverage parameter in offline RL (Munos, 2007; Munos and Szepesvári, 2008; Chen and Jiang, 2019)) by optimally designing the offline data distribution $\mu$. However, in our setting the agent has no prior knowledge of $\mu^\star$ and no way to explicitly search for it. Examples that admit low coverability include tabular MDPs and Block MDPs (Xie et al., 2023), linear/low-rank MDPs (Huang et al., 2023), and analytically sparse low-rank MDPs (Golowich et al., 2023); see Appendix C for further examples.

Concretely, we aim for sample complexity guarantees scaling as $\mathrm{poly}(H, C_{\mathsf{cov}}, \log|\mathcal{F}|, \log|\mathcal{W}|, \varepsilon^{-1})$, where $\varepsilon$ is the desired bound on the risk in Eq. (1).[3] Such a guarantee complements Xie et al. (2023), who achieved similar sample complexity under the Bellman completeness assumption, and parallels the fashion in which density ratio modeling allows one to remove completeness in offline RL.

**Additional notation.** For $n \in \mathbb{N}$, we write $[n] = \{1, \dots, n\}$. For a countable set $\mathcal{Z}$, we write $\Delta(\mathcal{Z})$ for the set of probability distributions on $\mathcal{Z}$. We adopt standard big-oh notation, and use $\widetilde{O}(\cdot)$ and $\widetilde{\Omega}(\cdot)$ to suppress factors polylogarithmic in $H, T, \varepsilon^{-1}, \log|\mathcal{F}|, \log|\mathcal{W}|$, and other problem parameters. For each $h \in [H]$, we define $\mathcal{F}_h = \{f_h \mid f \in \mathcal{F}\}$ and $\mathcal{W}_h = \{w_h \mid w \in \mathcal{W}\}$.

## 3 ONLINE RL WITH DENSITY RATIO REALIZABILITY

This section presents our main results for the online RL setting. We first introduce our main algorithm, GLOW (Algorithm 1), and explain the intuition behind its design (Section 3.1). We then show (Section 3.2) that GLOW obtains polynomial sample complexity guarantees (Theorems 3.1 and 3.2) under density ratio realizability and coverability, and conclude with a proof sketch.

### 3.1 ALGORITHM AND KEY IDEAS

Our algorithm, GLOW (Algorithm 1), is based on the principle of optimism in the face of uncertainty. For each iteration $t \leq T \in \mathbb{N}$, the algorithm uses the density ratio class $\mathcal{W}$ to construct a confidence set (or, version space) $\mathcal{F}^{(t)} \subseteq \mathcal{F}$ with the property that $Q^\star \in \mathcal{F}^{(t)}$. It then chooses a policy $\pi^{(t)} = \pi_{f^{(t)}}$ based on the value function $f^{(t)} \in \mathcal{F}^{(t)}$ with the most optimistic estimate $\mathbb{E}[f_1(x_1, \pi_{f,1}(x_1))]$ for the initial value. Then, it uses the policy $\pi^{(t)}$ to gather $K \in \mathbb{N}$ trajectories, which are used to update the confidence set for subsequent iterations.

Within the scheme above, the main novelty to our approach lies in the confidence set construction. GLOW appeals to *global optimism* (Jiang et al., 2017; Zanette et al., 2020; Du et al., 2021; Jin et al., 2021a; Xie et al., 2023), and constructs the confidence set $\mathcal{F}^{(t)}$ by searching for value functions $f \in \mathcal{F}$ that satisfy certain Bellman residual constraints for all layers $h \in [H]$ simultaneously. For MDPs with low coverability, previous such approaches (Jin et al., 2021a; Xie et al., 2023) make use of constraints based on *squared Bellman error*, which requires Bellman completeness. The confidence set construction in GLOW (Eq. (3)) departs from this approach, and aims to find $f \in \mathcal{F}$ such that the

---

[3]To simplify presentation as much as possible, we assume finiteness of $\mathcal{F}$ and $\mathcal{W}$, but our results extend to infinite classes via standard uniform convergence arguments. Likewise, we do not require exact realizability, and an extension to misspecified classes is given in Appendix E.

*average Bellman error* is small for all weight functions. At the population level, this (informally) corresponds to requiring that for all $h \in [H]$ and $w \in \mathcal{W}$,[4]

$$\mathbb{E}_{\bar{d}^{(t)}}\big[w_h(x_h, a_h)(f_h(x_h, a_h) - [\mathcal{T}_h f_{h+1}](x_h, a_h))\big] - \alpha^{(t)} \cdot \mathbb{E}_{\bar{d}^{(t)}}\big[(w_h(x_h, a_h))^2\big] \leq \beta^{(t)}. \quad (2)$$

where $\bar{d}_h^{(t)} := \frac{1}{t-1} \sum_{i < t} d_h^{\pi^{(t)}}$ is the historical data distribution and $\alpha^{(t)} > 0$ and $\beta^{(t)} > 0$ are algorithm parameters; this is motivated by the fact that the optimal value function satisfies $\mathbb{E}^\pi\big[w_h(x_h, a_h)(Q_h^\star(x_h, a_h) - [\mathcal{T}_h Q_{h+1}^\star](x_h, a_h))\big] = 0$ for all functions $w$ and policies $\pi$. Our analysis uses that Eq. (2) holds for the weight function $w_h^{(t)} := d_h^{\pi^{(t)}}/\bar{d}_h^{(t)}$, which allows to transfer bounds on the off-policy Bellman error for the historical distribution $\bar{d}^{(t)}$ to the on-policy Bellman error for $\pi^{(t)}$.

**Remark 3.1.** *Among density ratio-based algorithms for* offline *reinforcement learning (Jiang and Huang, 2020; Xie and Jiang, 2020; Zhan et al., 2022; Chen and Jiang, 2022; Rashidinejad et al., 2023), the constraint (2) is most directly inspired by the Minimax Average Bellman Optimization (*MABO*) algorithm (Xie and Jiang, 2020), which uses a similar minimax approximation to the average Bellman error.*

**Partial coverage and clipping.** Compared to the offline setting, much extra work is required to handle the issue of *partial coverage*. Early in the learning process, the ratio $w_h^{(t)} := d_h^{\pi^{(t)}}/\bar{d}_h^{(t)}$ may be unbounded, which prevents the naive empirical approximation to Eq. (2) from concentrating. To address this issue, GLOW carefully truncates the weight functions under consideration.

**Definition 3.1** (Clipping operator). *For any $w : \mathcal{X} \times \mathcal{A} \to \mathbb{R} \cup \{\infty\}$ and $\gamma \in \mathbb{R}$, we define the clipped weight function (at scale $\gamma$) via $\mathsf{clip}_\gamma[w](x, a) := \min\{w(x, a), \gamma\}$.*

We replace the weight functions in Eq. (2) with clipped counterparts $\mathsf{clip}_{\gamma^{(t)}}[w](x, a)$, where $\gamma^{(t)} := \gamma \cdot t$ for a parameter $\gamma \in [0, 1]$. For a given iteration $t$, clipping in this fashion may render Eq. (2) a poor approximation to the on-policy Bellman error. The crux of our analysis is to show—via coverability—that *on average* across all iterations, the approximation error is small.

An important difference relative to MABO is that the weighted Bellman error in Eq. (2) incorporates a quadratic penalty $-\alpha^{(t)} \cdot \mathbb{E}_{\bar{d}^{(t)}}[(w_h(x_h, a_h))^2]$ for the weight function. This is not essential to derive polynomial sample complexity guarantees, but is critical to attain the $1/\varepsilon^2$-type rates we achieve under our strongest realizability assumption. Briefly, regularization is beneficial because it allows us to appeal to variance-dependent Bernstein-style concentration; our analysis shows that while the variance of the weight functions under consideration may not be small on a per-iteration basis, it is small on average across all iterations (again, via coverability). Interestingly, similar quadratic penalties have been used within empirical offline RL algorithms based on density ratio modeling (Yang et al., 2020; Lee et al., 2021), as well as recent theoretical results (Zhan et al., 2022), but for considerations seemingly unrelated to concentration.

## 3.2 MAIN RESULT: SAMPLE COMPLEXITY BOUND FOR GLOW

We now present the main sample complexity guarantees for GLOW. The first result we present, which gives the tightest sample complexity bound, is stated under a form of density ratio realizability that strengthens Assumption 2.2. Concretely, we assume that the class $\mathcal{W}$ can realize density ratios for certain *mixtures* of policies. For $t \in \mathbb{N}$, we write $\pi^{(1:t)}$ as a shorthand for a sequence $(\pi^{(1)}, \cdots, \pi^{(t)})$, where $\pi^{(i)} \in \Pi$, and let $d^{\pi^{(1:t)}} := \frac{1}{t} \sum_{i=1}^t d^{\pi^{(i)}}$.

**Assumption 2.2′** (Density ratio realizability, mixture version). *Let $T$ be the parameter to GLOW (Algorithm 1). For all $h \in [H]$, $\pi \in \Pi$, $t \leq T$, and $\pi^{(1)}, \ldots, \pi^{(t)} \in \Pi$, we have*

$$w_h^{\pi; \pi^{(1:t)}}(x, a) := \frac{d_h^\pi(x, a)}{d_h^{\pi^{(1:t)}}(x, a)} \in \mathcal{W}.$$

This assumption directly facilitates transfer from the algorithm's *historical distribution* $\bar{d}_h^{(t)} := \frac{1}{t-1} \sum_{i < t} d_h^{\pi^{(t)}}$ to on-policy error. Under Assumption 2.2′, GLOW obtains $1/\varepsilon^2$-PAC sample complexity and $\sqrt{T}$-regret.

---

[4]Average Bellman error *without weight functions* is used in algorithms such at OLIVE (Jiang et al., 2017) and BILIN-UCB Du et al. (2021). Without weighting, this approach is insufficient to derive guarantees based on coverability; see discussion in Xie et al. (2023).

---

**Algorithm 1** GLOW: Global Optimism via Weight Function Realizability

---

**input:** Value function class $\mathcal{F}$, Weight function class $\mathcal{W}$, Parameters $T, K \in \mathbb{N}, \gamma \in [0, 1]$.

1: `// For Thm. 3.1, set` $T = \widetilde{\Theta}((H^2 C_{\text{cov}}/\varepsilon^2) \cdot \log(|\mathcal{F}||\mathcal{W}|/\delta))$`,` $K = 1$`, and` $\gamma = \sqrt{C_{\text{cov}}/(T \log(|\mathcal{F}||\mathcal{W}|/\delta))}$`.`

2: `// For Thm. 3.2, set` $T = \widetilde{\Theta}(H^2 C_{\text{cov}}/\varepsilon^2)$`,` $K = \widetilde{\Theta}(T \log(|\mathcal{F}||\mathcal{W}|/\delta))$`, and` $\gamma = \sqrt{C_{\text{cov}}/T}$`.`

3: Set $\gamma^{(t)} = \gamma \cdot t$, $\alpha^{(t)} = 8/\gamma^{(t)}$ and $\beta^{(t)} = (36\gamma^{(t)}/K(t-1)) \cdot \log(6|\mathcal{F}||\mathcal{W}|TH/\delta)$.

4: Initialize $\mathcal{D}_h^{(1)} = \varnothing$ for all $h \leq H$.

5: **for** $t = 1, \ldots, T$ **do**

6:     Define confidence set based on (regularized) minimax average Bellman error:

$$\mathcal{F}^{(t)} = \left\{ f \in \mathcal{F} \mid \forall h : \sup_{w \in \mathcal{W}_h} \widehat{\mathbb{E}}_{\mathcal{D}_h^{(t)}} \left[ ([\widehat{\Delta}_h f](x, a, r, x')) \cdot \widetilde{w}_h(x, a) - \alpha^{(t)} \cdot (\widetilde{w}_h(x, a))^2 \right] \leq \beta^{(t)} \right\}, \tag{3}$$

    where $\widetilde{w} := \text{clip}_{\gamma^{(t)}}[w]$ and $[\widehat{\Delta}_h f](x, a, r, x') := f_h(x, a) - r - \max_{a'} f_{h+1}(x', a')$.

7:     Compute optimistic value function and policy:

$$f^{(t)} := \underset{f \in \mathcal{F}^{(t)}}{\arg\max} \, \widehat{\mathbb{E}}_{x_1 \sim \mathcal{D}_1^{(t)}} [f_1(x_1, \pi_f(x_1))], \quad \text{and} \quad \pi^{(t)} := \pi_{f^{(t)}}. \tag{4}$$

8:     Initialize $\mathcal{D}_h^{(t+1)} \leftarrow \mathcal{D}_h^{(t)}$ for $h \in [H]$.

9:     **for** $k = 1, \ldots, K$ **do**                                         `// Online data collection.`

10:         Collect a trajectory $(x_1, a_1, r_1), \ldots, (x_H, a_H, r_H)$ by executing $\pi^{(t)}$.

11:         Update $\mathcal{D}_h^{(t+1)} \leftarrow \mathcal{D}_h^{(t+1)} \cup \{(x_h, a_h, r_h, x_{h+1})\}$ for each $h \in [H]$.

12: **output:** policy $\widehat{\pi} = \text{Unif}(\pi^{(1)}, \ldots, \pi^{(T)})$.                                  `// For PAC guarantee only.`

---

**Theorem 3.1** (Risk bound for GLOW under strong density ratio realizability). *Let $\varepsilon > 0$ be given, and suppose that Assumptions 2.1 and 2.2' hold. Then, GLOW, with hyperparameters $T = \widetilde{\Theta}((H^2 C_{\text{cov}}/\varepsilon^2) \cdot \log(|\mathcal{F}||\mathcal{W}|/\delta))$, $K = 1$, and $\gamma = \sqrt{C_{\text{cov}}/(T \log(|\mathcal{F}||\mathcal{W}|/\delta))}$ returns an $\varepsilon$-suboptimal policy $\widehat{\pi}$ with probability at least $1 - \delta$ after collecting*

$$N = \widetilde{O}\left( \frac{H^2 C_{\text{cov}}}{\varepsilon^2} \log(|\mathcal{F}||\mathcal{W}|/\delta) \right) \tag{5}$$

*trajectories. Additionally, for any $T \in \mathbb{N}$, with the same choice for $K$ and $\gamma$ as above, GLOW enjoys the regret bound $\mathbf{Reg} := \sum_{t=1}^{T} J(\pi^\star) - J(\pi^{(t)}) = \widetilde{O}\left( H\sqrt{C_{\text{cov}} T \log(|\mathcal{F}||\mathcal{W}|/\delta)} \right)$.*

Next, we provide our main result, which gives a sample complexity guarantee under density ratio realizability for pure policies (Assumption 2.2). To obtain the result, we begin with a class $\mathcal{W}$ that satisfies Assumption 2.2, then expand it to obtain an augmented class $\overline{\mathcal{W}}$ that satisfies mixture realizability (Assumption 2.2'). This reduction increases $\log|\mathcal{W}|$ by a $T$ factor, which we offset by increasing the batch size $K$; this leads to a polynomial increase in sample complexity.[5]

**Theorem 3.2** (Risk bound for GLOW under weak density ratio realizability). *Let $\varepsilon > 0$ be given, and suppose that Assumptions 2.1 and 2.2 hold for the classes $\mathcal{F}$ and $\mathcal{W}$. Then, GLOW, when executed with a modified class $\overline{\mathcal{W}}$ defined in Eq. (30) in Appendix E, with hyperparameters $T = \widetilde{\Theta}(H^2 C_{\text{cov}}/\varepsilon^2)$, $K = \widetilde{\Theta}(T \log(|\mathcal{F}||\mathcal{W}|/\delta))$, and $\gamma = \sqrt{C_{\text{cov}}/T}$, returns an $\varepsilon$-suboptimal policy $\widehat{\pi}$ with probability at least $1 - \delta$ after collecting $N$ trajectories, for*

$$N = \widetilde{O}\left( \frac{H^4 C_{\text{cov}}^2}{\varepsilon^4} \log(|\mathcal{F}||\mathcal{W}|/\delta) \right). \tag{6}$$

Theorems 3.1 and 3.2 show for the first time that value function realizability and density ratio realizability alone are sufficient for sample-efficient online RL under coverability. In particular, the sample complexity and regret bound in Theorem 3.1 match the coverability-based guarantees obtained in Xie et al. (2023, Theorem 1) under the complementary Bellman completeness assumption, with the only difference being that they scale with $\log(|\mathcal{F}||\mathcal{W}|)$ instead of $\log|\mathcal{F}|$; as discussed in

---

[5]This reduction also prevents us from obtaining a regret bound directly under Assumption 2.2, though a (slower-than-$\sqrt{T}$) regret bound can be attained using an explore-then-commit strategy.

Xie et al. (2023), this rate is tight for the special case of contextual bandits ($H = 1$). Interesting open questions include (i) whether the sample complexity for learning with density ratio realizability for pure policies can be improved to $1/\varepsilon^2$, and (ii) whether value realizability and coverability alone are sufficient for sample-efficient RL. Extensions to Theorems 3.1 and 3.2 under misspecification are given in Appendix E; see also Remark B.1 We further refer to Appendix C for examples instantiating these results, Appendix E.1 for a proof sketch, and Remark B.3 for a connection of our algorithm to the GOLF algorithm (Jin et al., 2021a; Xie et al., 2023).

## 4 EFFICIENT HYBRID RL WITH DENSITY RATIO REALIZABILITY

Toward overcoming the challenges of intractable computation in online exploration, a number of recent works show that including additional offline data in online RL can lead to computational benefits in theory (e.g., Xie et al., 2021b; Wagenmaker and Pacchiano, 2023; Song et al., 2023; Zhou et al., 2023) and in practice (e.g., Cabi et al., 2020; Nair et al., 2020; Ball et al., 2023; Song et al., 2023; Zhou et al., 2023). Notably, combining offline and online data can enable algorithms that provably explore without having to appeal to optimism or pessimism, both of which are difficult to implement efficiently under general function approximation.

Song et al. (2023) formalize a version of this setting—in which online RL is augmented with offline data—as *hybrid reinforcement learning*. Formally, in hybrid RL, the learner interacts with the MDP online (as in Section 1.1) but is additionally given an offline dataset $\mathcal{D}_{\mathrm{off}}$ collected from a data distribution $\nu$. The data distribution $\nu$ is typically assumed to provide coverage for the optimal policy $\pi^\star$ (formalized in Definition 4.3), but not on all policies, and thus additional online exploration is required (see Appendix F.2 for further discussion).

### 4.1 $H_2O$: A PROVABLE BLACK-BOX HYBRID-TO-OFFLINE REDUCTION

Interestingly, many of the above approaches for the hybrid setting simply apply offline algorithms (with relatively little modification) on a mixture of online and offline data (e.g., Cabi et al., 2020; Nair et al., 2020; Ball et al., 2023). This raises the question: *when can we use a given offline algorithm as a black box to solve the problem of hybrid RL (or, more generally, of online RL?)*. To answer this, we give a general meta-algorithm, $H_2O$, which provides a provable black-box reduction to solve the hybrid RL problem by repeatedly invoking a given offline RL algorithm on a mixture of offline data and freshly gathered online trajectories. To present the result, we first describe the class of offline RL algorithms with which it will be applied.

**Offline RL and partial coverage.** We refer to a collection of distributions $\mu = \{\mu_h\}_{h=1}^H$, where $\mu_h \in \Delta(\mathcal{X} \times \mathcal{A})$, as a *data distribution*, and we say that a dataset $\mathcal{D} = \{\mathcal{D}_h\}_{h=1}^H$ has $H \cdot n$ *samples from data distributions* $\mu^{(1)}, \dots, \mu^{(n)}$ if $\mathcal{D}_h = \{(x_h^{(i)}, a_h^{(i)}, r_h^{(i)}, x_{h+1}^{(i)})\}_{i=1}^n$ where $(x_h^{(i)}, a_h^{(i)}) \sim \mu_h^{(i)}$, $r_h^{(i)} \sim R_h(x_h^{(i)}, a_h^{(i)})$, $x_{h+1}^{(i)} \sim P_h(x_h^{(i)}, a_h^{(i)})$. We denote the *mixture distribution* via $\mu^{(1:n)} = \{\mu_h^{(1:n)}\}_{h=1}^H$, where $\mu_h^{(1:n)} := \frac{1}{n} \sum_{i=1}^n \mu_h^{(i)}$. An *offline RL algorithm* $\mathbf{Alg}_{\mathrm{off}}$ takes as input a dataset $\mathcal{D} = \{\mathcal{D}_h\}_{h=1}^H$ of $H \cdot n$ samples from $\mu^{(1)}, \dots, \mu^{(n)}$ and outputs a policy $\pi = (\pi_h)_{h=1}^H$.[6] An offline algorithm is measured by its risk, as in Equation (1); we write $\mathbf{Risk}_{\mathrm{off}}$ when we are in the offline interaction protocol.

An immediate problem with directly invoking offline RL algorithms in the hybrid model is that typical algorithms—particularly, those that do not make use of pessimism (e.g., Xie and Jiang, 2020)—require relatively *uniform* notions of coverage (e.g., coverage for all policies as opposed to just coverage for $\pi^\star$) to provide guarantees, leading one to worry that their behaviour might be completely uncontrolled when applied with non-exploratory datasets. Fortunately, we will show that for a large class algorithms whose risk scales with a measure of coverage we refer to as *clipped concentrability*, this phenomenon cannot occur.

**Definition 4.1** (Clipped concentrability coefficient)**.** *The clipped concentrability coefficient (at scale $\gamma \in \mathbb{R}_+$) for $\pi \in \Pi$ relative to a data distribution $\mu = \{\mu_h\}_{h=1}^H$, where $\mu_h \in \Delta(\mathcal{X} \times \mathcal{A})$, is defined as*

$$\mathsf{CC}_h(\pi, \mu, \gamma) := \left\| \mathsf{clip}_\gamma \left[ \frac{d_h^\pi}{\mu_h} \right] \right\|_{1, d_h^\pi}.$$

---

[6]When parameters are needed, $\mathbf{Alg}_{\mathrm{off}}$ should instead be thought of as the algorithm for a fixed choice of parameter. Likewise, we treat function approximators ($\mathcal{F}$, $\mathcal{W}$, etc.) as part of the algorithm. We consider adaptively chosen $(\mu^{(i)})_{i=1}^n$ because, in the reduction, $\mathbf{Alg}_{\mathrm{off}}$ will be invoked on history-dependent datasets.

This coefficient should be thought of as a generalization of the standard (squared) $L_2(\mu)$ concentrability coefficient $C_{\mathsf{conc},2,h}^2(\pi,\mu) := \|d_h^\pi/\mu_h\|_{2,\mu_h}^2 = \|d_h^\pi/\mu_h\|_{1,d_h^\pi}$, a fundamental object in the analysis of offline RL algorithms (e.g., Farahmand et al., 2010), but incorporates clipping to better handle partial coverage. We consider offline RL algorithms with the property that for any offline distribution $\mu$, the algorithm's risk can be bounded by the clipped concentrability coefficients for (i) the output policy $\widehat{\pi}$, and (ii) the optimal policy $\pi^\star$. For the following definition, we recall the notation $\gamma^{(n)} := \gamma \cdot n$.

**Definition 4.2** (CC-bounded offline RL algorithm). *An offline algorithm* $\mathbf{Alg}_{\mathsf{off}}$ *is* CC-*bounded at scale* $\gamma \in \mathbb{R}_+$ *under an assumption* Assumption$(\cdot)$ *if there exists scalars* $\mathfrak{a}_\gamma, \mathfrak{b}_\gamma$ *such that for all* $n \in \mathbb{N}$ *and data distributions* $\mu^{(1)}, \ldots, \mu^{(n)}$, $\mathbf{Alg}_{\mathsf{off}}$ *outputs* $p \in \Delta(\Pi)$ *satisfying*

$$\mathbf{Risk}_{\mathsf{off}} = \mathbb{E}_{\widehat{\pi} \sim p}[J(\pi^\star) - J(\widehat{\pi})] \leq \sum_{h=1}^H \frac{\mathfrak{a}_\gamma}{n} \left( \mathsf{CC}_h(\pi^\star, \mu^{(1:n)}, \gamma^{(n)}) + \mathbb{E}_{\widehat{\pi} \sim p}[\mathsf{CC}_h(\widehat{\pi}, \mu^{(1:n)}, \gamma^{(n)})] \right) + \mathfrak{b}_\gamma$$

$$(7)$$

*with probability at least* $1 - \delta$, *when given* $H \cdot n$ *samples from* $\mu^{(1)}, \ldots, \mu^{(n)}$ *such that* Assumption$(\mu^{(1:n)}, M^\star)$ *is satisfied.*

This definition does not automatically imply that the offline algorithm has low offline risk, but simply that the risk can be bounded in terms of clipped coverage (which may be large if the dataset has poor coverage). In the sequel, we will show that many natural offline RL algorithms have this property (Appendix F.1).[7]

**The $H_2O$ algorithm.** Our reduction, $H_2O$, is given in Algorithm 2. For any dataset $\mathcal{D}$, we will write $\mathcal{D}|_{1:t}$ for the subset consisting of its first $t$ elements. The algorithm is initialized with an offline dataset $\mathcal{D}_{\mathsf{off}} = \{\mathcal{D}_{\mathsf{off},h}\}_{h=1}^H$, and at each iteration $t \in [T]$ invokes the black-box offline RL algorithm $\mathbf{Alg}_{\mathsf{off}}$ with a dataset $\mathcal{D}_{\mathsf{hybrid}} = \{\mathcal{D}_{\mathsf{hybrid},h}\}_{h=1}^H$ that mixes the first $t$ elements of $\mathcal{D}_{\mathsf{off},h}$ with all of the online data gathered so far. This produces a policy $\pi^{(t)}$, which is executed to gather trajectories that are then added to the online dataset and used at the next iteration. $H_2O$ is inspired by empirical methods for the hybrid setting (e.g., Cabi et al., 2020; Nair et al., 2020; Ball et al., 2023), and in particular the meta-algorithm is efficient whenever $\mathbf{Alg}_{\mathsf{off}}$ is.

---

**Algorithm 2** $H_2O$: Hybrid-to-Offline Reduction

    **input:** Parameter $T \in \mathbb{N}$, offline algorithm $\mathbf{Alg}_{\mathsf{off}}$, offline datasets $\mathcal{D}_{\mathsf{off}} = \{\mathcal{D}_{\mathsf{off},h}\}_h$ each of
    size $T$.
1: Initialize $\mathcal{D}_{\mathsf{on},h}^{(1)} = \mathcal{D}_{\mathsf{hybrid},h}^{(1)} = \varnothing$ for all $h \in [H]$.
2: **for** $t = 1, \ldots, T$ **do**
3:     Get policy $\pi^{(t)}$ from $\mathbf{Alg}_{\mathsf{off}}$ on dataset $\mathcal{D}_{\mathsf{hybrid}}^{(t)} = \{\mathcal{D}_{\mathsf{hybrid},h}^{(t)}\}_h$.
4:     Collect $(x_1, a_1, r_1), \ldots, (x_H, a_H, r_H)$ using $\pi^{(t)}$; $\mathcal{D}_{\mathsf{on},h}^{(t+1)} := \mathcal{D}_{\mathsf{on},h}^{(t)} \cup \{(x_h, a_h, r_h, x_{h+1})\}$.
5:     Aggregate offline and online data: $\mathcal{D}_{\mathsf{hybrid},h}^{(t+1)} := \mathcal{D}_{\mathsf{off},h}|_{1:t} \cup \mathcal{D}_{\mathsf{on},h}^{(t+1)}$ for all $h \in [H]$.
6: **output:** policy $\widehat{\pi} = \mathsf{Unif}(\pi^{(1)}, \ldots, \pi^{(T)})$.

---

**Main risk bound for $H_2O$.** We now present the main result for this section: a risk bound for the $H_2O$ reduction. Our bound depends on the coverability parameter for the underlying MDP, as well as the quality of the offline data distribution $\nu$, quantified by *single-policy concentrability*.

**Definition 4.3** (Single-policy concentrability). *A data distribution* $\nu = \{\nu_h\}_{h=1}^H$ *satisfies* $C_\star$-*single-policy concentrability if* $\max_h \|d_h^{\pi^\star}/\nu_h\|_\infty \leq C_\star$.

**Theorem 4.1** (Risk bound for $H_2O$). *Let* $T \in \mathbb{N}$ *be given, let* $\mathcal{D}_{\mathsf{off}}$ *consist of* $H \cdot T$ *samples from data distribution* $\nu$, *and suppose that* $\nu$ *satisfies* $C_\star$-*single-policy concentrability. Let* $\mathbf{Alg}_{\mathsf{off}}$ *be* CC-*bounded at scale* $\gamma \in (0,1)$ *under* Assumption$(\cdot)$, *with parameters* $\mathfrak{a}_\gamma$ *and* $\mathfrak{b}_\gamma$. *Suppose that* $\forall\, t \in [T]$ *and* $\pi^{(1)}, \ldots, \pi^{(t)} \in \Pi$, Assumption$(\mu^{(t)}, M^\star)$ *holds for* $\mu^{(t)} := \{1/2(\nu_h + 1/t \sum_{i=1}^t d_h^{\pi^{(i)}})\}_{h=1}^H$. *Then, with probability at least* $1 - \delta T$, *the risk of* $H_2O$ *(Algorithm 2) with* $T$, $\mathbf{Alg}_{\mathsf{off}}$, *and* $\mathcal{D}_{\mathsf{off}}$ *is*

---

[7]Examples of assumptions for Assumption include value function completeness (e.g., for FQI (Chen and Jiang, 2019)) and realizability of value functions and density ratios (e.g., for MABO (Xie and Jiang, 2020)).

*bounded as*[8]

$$\textbf{Risk} \leq \widetilde{O}\left(H\left(\frac{\mathfrak{a}_\gamma(C_\star + C_{\mathsf{cov}})}{T} + \mathfrak{b}_\gamma\right)\right). \tag{8}$$

For the algorithms we consider, one can take $\mathfrak{a}_\gamma \propto \mathfrak{a}/\gamma$ and $\mathfrak{b}_\gamma \propto \mathfrak{b}\gamma$ for scalar-valued problem parameters $\mathfrak{a}, \mathfrak{b} > 0$, so that choosing $\gamma$ optimally gives $\textbf{Risk} \leq \widetilde{O}\big(H\sqrt{(C_\star + C_{\mathsf{cov}})\mathfrak{a}\mathfrak{b}/T}\big)$. This is a special case of a more general result, Theorem F.6, which handles the general case where $\nu$ need not satisfy single-policy concentrability.

## 4.2 Applying the Reduction: HYGLOW

We now apply $H_2O$ to give a hybrid counterpart to GLOW (Algorithm 1), using a variant of MABO (Xie and Jiang, 2020) as the black box offline RL algorithm $\textbf{Alg}_{\mathsf{off}}$ in $H_2O$. Further examples, which apply Fitted Q-Iteration (FQI) and Model-Based Maximum Likelihood Estimation as the black box, are deferred to Appendix F.1.

More specifically, the offline RL algorithm we consider is a variant of MABO that incorporates clipping and regularization in the same fashion as GLOW, which we call MABO.CR. Our algorithm takes as input a dataset $\mathcal{D} = \{\mathcal{D}_h\}$ with $H \cdot n$ samples, has parameters consisting of a value function class $\mathcal{F}$, a weight function class $\mathcal{W}$, and a clipping scale $\gamma$, and computes the following estimator:

$$\widehat{f} \in \arg\min_{f \in \mathcal{F}} \max_{w \in \mathcal{W}} \sum_{h=1}^{H} \left|\widehat{\mathbb{E}}_{\mathcal{D}_h}\left[\check{w}_h(x_h, a_h)[\widehat{\Delta}_h f](x_h, a_h, r_h, x'_{h+1})\right]\right| - \alpha^{(n)}\widehat{\mathbb{E}}_{\mathcal{D}_h}\left[\check{w}_h^2(x_h, a_h)\right], \quad (9)$$

where $\alpha^{(n)} := 8/\gamma^{(n)}$ and $\check{w}_h := \mathsf{clip}_{\gamma^{(n)}}[w_h]$. By instantiating $H_2O$ with this algorithm, we obtain a density ratio-based algorithm for the hybrid RL setting (HYGLOW; Algorithm 3) that is statistically and computationally efficient.

**Theorem 4.2** (Risk bound for HYGLOW). *Let $\varepsilon > 0$ be given, let $\mathcal{D}_{\mathsf{off}}$ consist of $H \cdot T$ samples from data distribution $\nu$, and suppose that $\nu$ satisfies $C_\star$-single-policy concentrability. Suppose that $Q^\star \in \mathcal{F}$ and that for all $t \in [T]$, $\pi \in \Pi$, and $h \in [H]$, we have $d_h^\pi/\mu_h^{(t)} \in \mathcal{W}$, where $\mu_h^{(t)} := 1/2(\nu_h + 1/t\sum_{i=1}^{t} d_h^{\pi^{(i)}})$. Then, HYGLOW (Algorithm 3) with inputs $T = \widetilde{\Theta}\big((H^4(C_{\mathsf{cov}} + C_\star)/\varepsilon^2) \cdot \log(|\mathcal{F}||\mathcal{W}|/\delta)\big)$, $\mathcal{F}$, augmented $\overline{\mathcal{W}}$ defined in Eq. (34), $\gamma = \widetilde{\Theta}\left(\sqrt{(C_\star + C_{\mathsf{cov}})/TH^2 \log(|\mathcal{F}||\mathcal{W}|/\delta)}\right)$, and $\mathcal{D}_{\mathsf{off}}$ returns an $\varepsilon$-suboptimal policy with probability at least $1 - \delta T$ after collecting $N = \widetilde{O}\left(\frac{H^2(C_{\mathsf{cov}} + C_\star)}{\varepsilon^2}\log(|\mathcal{F}||\mathcal{W}|/\delta)\right)$ trajectories.*

This result is obtained from Theorem 4.1 by showing that MABO.CR is CC-bounded under $Q^\star$-realizability and a density ratio realizability assumption (cf. Theorem F.1). While the sample complexity and realizability assumptions parallel those of GLOW, the computational efficiency is improved because we remove the need for global optimism. More specifically, note that when clipping and the absolute value signs are removed from Eq. (9), the optimization problem is concave-convex in the function class $\mathcal{F}$ and weight function class $\mathcal{W}$, a desirable property shared by standard density-ratio based offline algorithms (Xie and Jiang, 2020; Zhan et al., 2022). To accommodate clipping and the absolute value signs efficiently, we note that Eq. (9) can be written as a convex-concave program in which the max player optimizes over the set $\widetilde{\mathcal{W}}_{\gamma,h} := \{\pm\mathsf{clip}_{\gamma n}[w_h] \mid w_h \in \mathcal{W}_h\}$ and the result continues to hold if the max player optimizes over any expanded weight function class $\widetilde{\mathcal{W}}_\gamma \subseteq \mathcal{W}'$ such that $\|w\|_\infty \leq \gamma n$ for all $w \in \mathcal{W}'$; we defer the details to Appendix F.1.2.

## 5 Conclusion

Our work shows for the first time that density ratio modeling has provable benefits for online reinforcement learning, and serves as step in a broader research program that aims to clarify connections between online and offline reinforcement learning. On the theoretical side, promising directions for future research include (i) resolving whether sample efficiency is possible under only coverability and value function realizability, and (ii) generic reductions from offline to online RL (see Appendix F.3 for a brief discussion). In addition, we are excited to explore the possibility of developing practical and computationally efficient online reinforcement learning algorithms based on density ratio modeling.

---

[8]We define risk for the hybrid setting as in Eq. (1). Our result is stated as a bound on the risk to the optimal policy $\pi^\star$, but extends to give a bound on the risk of any comparator $\pi$ with $C_\star$ replaced by coverage for $\pi$.

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

# Contents of Appendix

## A  ADDITIONAL RELATED WORK

**Online reinforcement learning.** Xie et al. (2023) introduce the notion of coverability and provide regret bounds for online reinforcement learning under the assumption of access to a value function class $\mathcal{F}$ satisfying Bellman completeness. Liu et al. (2023) extend their result to more general coverage conditions under the same Bellman completeness assumption. Our work complements these results by providing guarantees based on coverability that do not require Bellman completeness.

To the best of our knowledge, our work is the first to provide provable sample complexity guarantees for online reinforcement learning that take advantage of the ability to model density ratios, but a number of closely related works bear mentioning. Recent work of Huang et al. (2023) considers the low-rank MDP model and provides an algorithms which takes advantage of a form of *occupancy realizability*. Occupancy realizability, while related to density ratio realizability, is stronger assumption in general: For example, the Block MDP model (Krishnamurthy et al., 2016; Du et al., 2019; Misra et al., 2019; Zhang et al., 2022; Mhammedi et al., 2023) admits a density ratio class of low complexity, but does not admit a small occupancy class. Overall, however, their results are somewhat complementary, as they do not require any form of value function realizability. A number of other recent works in online reinforcement learning also make use of occupancy measures, but restrict to linear function approximation (Neu and Pike-Burke, 2020; Neu and Okolo, 2023). Lastly, a number of works apply density ratio modeling in online settings with an empirical focus (Feng et al., 2019; Nachum et al., 2019), but do not address the exploration problem.

**Offline reinforcement learning.** Within the literature on offline reinforcement learning theory, density ratio modeling has been widely used as a means to avoid strong Bellman completeness requirements, with theoretical guarantees for policy evaluation (Liu et al., 2018; Uehara et al., 2020; Yang et al., 2020; Uehara et al., 2021) and policy optimization (Jiang and Huang, 2020; Xie and Jiang, 2020; Zhan et al., 2022; Chen and Jiang, 2022; Rashidinejad et al., 2023; Ozdaglar et al., 2023). A number of additional works investigate density ratio modeling with an empirical focus, and do not provide finite-sample guarantees (Nachum et al., 2019; Kostrikov et al., 2019; Nachum and Dai, 2020; Zhang et al., 2020; Lee et al., 2021).

**Hybrid reinforcement learning.** Song et al. (2023) were the first to show, theoretically, that the hybrid reinforcement learning model can lead to computational benefits over online and offline RL individually. Our reduction, $H_2O$, can be viewed as generalization of their Hybrid Q-Learning algorithm, with their result corresponding to the special case in which FQI is applied as a the base algorithm. Our guarantees under coverability complement their guarantees based on bilinear rank.

Other recent works on hybrid reinforcement learning in specialized settings (e.g., tabular MDPs or linear MDPs) include Wagenmaker and Pacchiano (2023); Li et al. (2023); Zhou et al. (2023).

## B    COMPARING WEIGHT FUNCTION REALIZABILITY TO ALTERNATIVE REALIZABILITY ASSUMPTIONS

In this section, we compare the density ratio realizability assumption in Assumption 2.2 to a number of alternative realizability assumptions.

**Comparison to Bellman completeness.**    A traditional assumption in the analysis of value function approximation methods is to assume that $\mathcal{F}$ satisfies a representation condition called *Bellman completeness*, which asserts that $\mathcal{T}_h\mathcal{F}_{h+1} \subseteq \mathcal{F}_h$; this assumption is significantly stronger than just assuming realizability of $Q^\star$ (Assumption 2.1), and has been used throughout offline RL (Antos et al., 2008; Chen and Jiang, 2019), and online RL (Zanette et al., 2020; Jin et al., 2021a; Xie et al., 2023).

Bellman completeness is incomparable to our density ratio realizability assumption. For example, the low-rank MDP model (Jin et al., 2020) in which $P_h(x' \mid x, a) = \langle \phi_h(x, a), \psi_h(x') \rangle$ satisfies Bellman completeness when the feature map $\phi$ is known to the learner even if $\psi$ is unknown (but may not satisfy it otherwise), and satisfies weight function realizability when the feature map $\psi$ is known even if $\phi$ is unknown (but may not satisfy it otherwise). Examples C.1 and C.2 in Appendix C give further examples that satisfy weight function realizability but are not known to satisfy Bellman completeness.

**Comparison to model-based realizability.**    Weight function realizability is strictly weaker than model-based realizability (e.g., Foster et al., 2021), in which one assumes access to a model class $\mathcal{M}$ of MDPs that contains the true MDP.[9] Since each MDP induces an occupancy for every policy, it is straightforward to see that given such a class, we can construct a weight function class $\mathcal{W}$ that satisfies Assumption 2.2 with

$$\log|\mathcal{W}| \leq O(\log|\mathcal{M}| + \log|\Pi|),$$

as well as a realizable value function class with $\log|\mathcal{F}| \leq O(\log|\mathcal{M}|)$. On the other hand, weight function realizability does not imply model-based realizability; a canonical example that witnesses this is the *Block MDP*.

**Example B.1** (Block MDP).  In the well-studied Block MDP model (Krishnamurthy et al., 2016; Jiang et al., 2017; Du et al., 2019; Misra et al., 2019; Zhang et al., 2022; Mhammedi et al., 2023), there exists realizable value function class $\mathcal{F}$ and weight function class $\mathcal{W}$ with $\log|\mathcal{F}|, \log|\mathcal{W}| \lesssim \mathrm{poly}(|\mathcal{S}|, |\mathcal{A}|, \log|\Phi|)$, where $\mathcal{S}$ is the *latent state space* and $\Phi$ is a class of *decoder functions*. However, there does not exist a realizable model class with bounded statistical complexity (see discussion in, e.g., Mhammedi et al. (2023)).                                                                ◁

**Alternative forms of density ratio realizability.**

**Remark B.1** (Clipped density ratio realizability).  *Since* GLOW *only accesses the weight function class $\mathcal{W}$ through the clipped weight functions, we can replace Assumption 2.2 with the assumption that for all $\pi, \pi' \in \Pi$, $t \leq T$, we have $\mathrm{clip}_{\gamma t}\left[w^{\pi;\pi'}\right] \in \mathcal{W}$,   where $T \in \mathbb{N}$ and $\gamma \in [0, 1]$ are chosen as in Theorem 3.2. Likewise, we can replace Assumption 2.2' with the assumption that for all $\pi \in \Pi$, and for all $t \leq T$, and $\pi_{1:t} \in \Pi^t$, we have $\mathrm{clip}_{\gamma t}\left[w^{\pi;\pi^{(1:t)}}\right] \in \mathcal{W}$,   for $T \in \mathbb{N}$ and $\gamma \in [0, 1]$ chosen as in Theorem 3.1.*

**Remark B.2** (Density ratio realizability relative to a fixed reference distribution).  *Assumption 2.2 is weaker than assuming access to a class $\mathcal{W}$ that can realize the ratio $d_h^\pi/\nu_h$ (or alternatively $\nu_h/d_h^\pi$) for all $\pi \in \Pi$, where $\nu_h$ is an arbitrary fixed distribution (natural choices might include $\nu_h = \mu_h^\star$ or $\nu_h = d_h^{\pi^\star}$). Indeed, given access to such a class, the expanded class $\mathcal{W}' := \{w/w' \mid w, w' \in \mathcal{W}\}$ satisfies Assumption 2.2, and has $\log|\mathcal{W}'| \leq 2\log|\mathcal{W}|$.*

**Remark B.3** (Connection to GOLF).  *Prior work (Xie et al., 2023) analyzed the GOLF algorithm of Jin et al. and established positive results under coverability and Bellman completeness. We remark that*

---

[9]Up to $H$ factors, this is equivalent to assuming access to a class of transition distributions that can realize the true transition distribution, and a class of reward distributions that can realize the true reward distribution.

*by allowing for weight functions that take negative values,*[10] GLOW *can be viewed as a generalization of* GOLF*, and can be configured to obtain comparable results. Indeed, given a value function class $\mathcal{F}$ that satisfies Bellman completeness, the weight function class $\mathcal{W} := \{f - f' \mid f, f' \in \mathcal{F}\}$ leads to a confidence set construction at least as tight as that of* GOLF*. To see this, observe that if we set $\gamma \geq 2$ so that no clipping occurs, our construction for $\mathcal{F}^{(t)}$ (Eq. (3)) implies (after standard concentration arguments) that $Q^\star \in \mathcal{F}^{(t)}$ and that in-sample squared Bellman errors are small with high probability. These ingredients are all that is required to repeat the analysis of* GOLF *from Xie et al. (2023).*

## C    EXAMPLES FOR GLOW

In this section, we give two new examples in which our main results for GLOW, Theorems 3.1 and 3.2, can be applied: MDPs with low-rank density features and general value functions, and a class of MDPs we refer to as *Generalized Block MDPs*.

**Example C.1** (Realizable $Q^\star$ with low-rank density features). Consider a setting in which (i) $Q^\star \in \mathcal{F}$, and (ii), there is a known feature map $\psi_h : \mathcal{X} \times \mathcal{A} \to \mathbb{R}^d$ with $\|\psi_h(x, a)\|_2 \leq 1$ such that for all $\pi \in \Pi$, $d_h^\pi(x, a) = \langle \psi_h(x, a), \theta^\pi \rangle$ for an unknown parameter $\theta^\pi \in \mathbb{R}^d$ with $\|\theta^\pi\|_2 \leq 1$. This assumption is sometimes referred to as *low occupancy complexity* (Du et al., 2021), and has been studied with and without known features. In this case, we have $C_{\text{cov}} \leq d$, and one can construct a weight function class $\mathcal{W}$ that satisfies Assumption 2.2′ with $\log|\mathcal{W}| \leq \widetilde{O}(dH)$ (Huang et al., 2023).[11] As a result, Theorem 3.1 gives sample complexity $\widetilde{O}(H^3 d^2 \log|\mathcal{F}|/\varepsilon^2)$. Note that while this setup requires that the occupancies themselves have low-rank structure, the class $\mathcal{F}$ can consist of arbitrary, potentially nonlinear functions (e.g., neural networks). We remark that when the feature map $\psi$ is not known, but instead belongs to a known class $\Psi$, the result continues to hold, at the cost of expanding $\mathcal{W}$ to have size $\log|\mathcal{W}| \leq \widetilde{O}(dH + \log|\Psi|)$.

This example is similar to but complementary to Huang et al. (2023), who give guarantees for reward-free exploration under low-rank occupancies. Their results do not require any form of value realizability, but require a low-rank MDP assumption (which, in particular, implies Bellman completeness).[12]                                                                                                                     ◁

Our next example concerns a generalization of the well-studied Block MDP framework (Krishnamurthy et al., 2016; Jiang et al., 2017; Du et al., 2019; Misra et al., 2019; Zhang et al., 2022; Mhammedi et al., 2023) that we refer to as the *Generalized Block MDP*.

**Definition C.1** (Generalized Block MDP). *A Generalized Block MDP $\mathcal{M} = (\mathcal{X}, \mathcal{S}, \mathcal{A}, P_{\text{latent}}, R_{\text{latent}}, q, H, d_1)$ is comprised of an observation space $\mathcal{X}$, latent state space $\mathcal{S}$, action space $\mathcal{A}$, latent space transition kernel $P_{\text{latent}} : \mathcal{S} \times \mathcal{A} \to \Delta(\mathcal{S})$, and emission distribution $q : \mathcal{S} \to \Delta(\mathcal{X})$. The latent state space evolves based on the agent's action $a_h \in \mathcal{A}$ via the process*

$$r_h \sim R_{\text{latent}}(s_h, a_h), \quad s_{h+1} \sim P_{\text{latent}}(\cdot \mid s_h, a_h), \tag{10}$$

*with $s_1 \sim d_1$; we refer to $s_h$ as the latent state. The latent state is not observed directly, and instead we observe observations $x_h \in \mathcal{X}$ generated by the emission process*

$$x_h \sim q(\cdot \mid s_h). \tag{11}$$

*We assume that the emission process satisfies the decodability property:*

$$\operatorname{supp} q(\cdot \mid s) \cap \operatorname{supp} q(\cdot \mid s') = \varnothing, \quad \forall s' \neq s \in \mathcal{S}. \tag{12}$$

*Decodability implies that there exists a (unknown to the agent) decoder $\phi_\star : \mathcal{X} \to \mathcal{S}$ such that $\phi_\star(x_h) = s_h$ a.s. for all $h \in [H]$, meaning that latent states can be uniquely decoded from observations. Prior work on the Block MDP framework (Krishnamurthy et al., 2016; Jiang et al., 2017; Du et al., 2019; Misra et al., 2019; Zhang et al., 2022; Mhammedi et al., 2023) assumes that the latent*

---

[10]In this context, $\mathcal{W}$ can be thought of more generally as a class of *test functions*.

[11]To be precise, $\mathcal{W}$ is infinite, and this result requires a covering number bound. We omit a formal treatment, and refer to Huang et al. (2023) for details.

[12]The low-rank MDP assumption is incomparable to the assumption we consider here, as it implies that $d_h^\pi(x) = \langle \psi_h(x), \theta^\pi \rangle$, but does not necessarily imply that the *state-action* occupancies (i.e. $d_h^\pi(x, a)$) are low-rank.

*space $\mathcal{S}$ and action space $\mathcal{A}$ are finite, but allow the observation space $\mathcal{X}$ to be large or potentially infinite. They provide sample complexity guarantees that scale as* $\mathrm{poly}(|\mathcal{S}|, |\mathcal{A}|, H, \log|\Phi|, \varepsilon^{-1})$, *where $\Phi$ is a known class of decoders that contains $\phi_\star$. We use the term* Generalized Block MDP *to refer to Block MDPs in which the latent space not tabular, and can be arbitrarily large.*

**Example C.2** (Generalized Block MDPs with coverable latent states)**.** We can use GLOW to give sample complexity guarantees for Generalized Block MDPs in which the latent space is large, but has low coverability. Let $\Pi_{\mathrm{latent}} = (\mathcal{S} \times [H] \to \Delta(\mathcal{A}))$ denote the set of all randomized policy that operate on the latent space. Assume that the following conditions hold:

- We have a value function class $\mathcal{F}_{\mathrm{latent}}$ such that $Q^\star_{\mathrm{latent}} \in \mathcal{F}$, where $Q^\star_{\mathrm{latent}}$ is the optimal $Q$-function for the latent space.

- We have access to a class of *latent space* density ratios $\mathcal{W}_{\mathrm{latent}}$ such that for all $h \in [H]$, and all $\pi, \pi' \in \Pi_{\mathrm{latent}}$,

$$w^{\pi,\pi'}_{\mathrm{latent},h}(s,a) := \frac{d^\pi_{\mathrm{latent},h}(s,a)}{d^{\pi'}_{\mathrm{latent},h}(s,a)} \in \mathcal{W}_{\mathrm{latent}},$$

where $d^\pi_{\mathrm{latent},h} = \mathbb{P}^\pi(s_h = s, a_h = a)$ is the latent occupancy measure.

- The *latent coverability coefficient* is bounded:

$$C_{\mathrm{cov,latent}} := \inf_{\mu_1,\dots,\mu_H \in \Delta(\mathcal{S} \times \mathcal{A})} \sup_{\pi \in \Pi_{\mathrm{latent}}, h \in [H]} \left\| \frac{d^\pi_{\mathrm{latent},h}}{\mu_h} \right\|_\infty.$$

We claim that whenever these conditions hold, analogous conditions hold in observation space (viewing the Generalized BMDP as a large MDP), allowing GLOW and Theorem 3.2 to be applied. Namely, we have:

- There exists a class $\mathcal{F}$ satisfying Assumption 2.1 in observation space such that $\log|\mathcal{F}| \leq O(\log|\mathcal{F}_{\mathrm{latent}}| + \log|\Phi|)$.

- There exists a weight function class $\mathcal{W}$ satisfying Assumption 2.2 in observation space such that $\log|\mathcal{W}| \leq O(\log|\mathcal{W}_{\mathrm{latent}}| + \log|\Phi|)$.

- We have $C_{\mathrm{cov}} \leq C_{\mathrm{cov,latent}}$.

As a result, GLOW attains sample complexity $\mathrm{poly}(C_{\mathrm{cov.latent}}, H, \log|\mathcal{F}_{\mathrm{latent}}|, \log|\mathcal{W}_{\mathrm{latent}}|, \log|\Phi|, \varepsilon^{-1})$. This generalizes existing results for Block MDPs with tabular latent state spaces, which have $\log|\mathcal{F}_{\mathrm{latent}}|, \log|\mathcal{W}_{\mathrm{latent}}|, C_{\mathrm{cov,latent}} = \mathrm{poly}(|\mathcal{S}|, |\mathcal{A}|, H)$. ◁

# D  TECHNICAL TOOLS

**Lemma D.1** (Azuma-Hoeffding). *Let $M \in \mathbb{N}$ and $(Y_m)_{m \leq M}$ be a sequence of random variables adapted to a filtration $(\mathscr{F}_m)_{m \leq M}$. If $|Y_m| \leq R$ almost surely, then with probability at least $1 - \delta$,*

$$\left| \sum_{m=1}^{M} Y_m - \mathbb{E}_{m-1}[Y_m] \right| \leq R \cdot \sqrt{8M \log(2\delta^{-1})}.$$

**Lemma D.2** (Freedman's inequality (e.g., Agarwal et al., 2014)). *Let $M \in \mathbb{N}$ and $(Y_m)_{m \leq M}$ be a real-valued martingale difference sequence adapted to a filtration $(\mathscr{F}_m)_{m \leq M}$. If $|Y_m| \leq R$ almost surely, then for any $\eta \in (0, 1/R)$, with probability at least $1 - \delta$,*

$$\left| \sum_{m=1}^{M} Y_m \right| \leq \eta \sum_{m=1}^{M} \mathbb{E}_{m-1}\big[(Y_m)^2\big] + \frac{\log(2\delta^{-1})}{\eta}.$$

The following lemma is a standard consequence of Lemma D.2 (e.g., Foster et al., 2021).

**Lemma D.3.** *Let $M \in \mathbb{N}$ and $(Y_m)_{m \leq M}$ be a sequence of random variables adapted to a filtration $(\mathscr{F}_m)_{m \leq M}$. If $0 \leq Y_m \leq R$ almost surely, then with probability at least $1 - \delta$,*

$$\sum_{m=1}^{M} Y_m \leq \frac{3}{2} \sum_{m=1}^{M} \mathbb{E}_{m-1}[Y_m] + 4R \log(2\delta^{-1}),$$

*and*

$$\sum_{m=1}^{M} \mathbb{E}_{m-1}[Y_m] \leq 2 \sum_{m=1}^{M} Y_m + 8R \log(2\delta^{-1}).$$

## D.1  REINFORCEMENT LEARNING PRELIMINARIES

**Lemma D.4** (Jiang et al. (2017, Lemma 1)). *For any value function $f = (f_1, \ldots, f_H)$,*

$$\mathbb{E}_{x_1 \sim d_1}[f_1(x_1, \pi_{f_1}(x_1))] - J(\pi_f) = \sum_{h=1}^{H} \mathbb{E}_{d_h^{\pi_f}}[f_h(x_h, a_h) - [\mathcal{T}_h f_{h+1}](x_h, a_h)].$$

**Lemma D.5** (Per-state-action elliptic potential lemma; Xie et al. (2023, Lemma 4)). *Let $d^{(1)}, \ldots, d^{(T)}$ be an arbitrary sequence of distributions over a set $\mathcal{Z}$, and let $\mu \in \Delta(\mathcal{Z})$ be a distribution such that $d^{(t)}(z)/\mu(z) \leq C$ for all $z \in \mathcal{Z}$ and $t \in [T]$. Then, for all $z \in \mathcal{Z}$, we have*

$$\sum_{t=1}^{T} \frac{d^{(t)}(z)}{\sum_{i<t} d^{(m)}(z) + C\mu(z)} \leq 2 \log(1 + T).$$

# E PROOFS FROM SECTION 3 (ONLINE RL)

This section of the appendix is organized as follows:

- Appendix E.1 gives a high-level sketch of the proof of Theorems 3.1 and 3.2, highlighting the role of clipped density ratios and coverability.

- Appendix E.2 provides supporting technical results for GLOW, including concentration guarantees.

- Appendix E.3 presents our main technical result for GLOW, Lemma E.4, which bounds the cumulative suboptimality of the iterates $\pi^{(1)}, \ldots, \pi^{(T)}$ produced by the algorithm for general choices of the parameters $T$, $K$, and $\gamma > 0$.

- Finally, in Appendices E.4 and E.5, we invoke with specific parameter choices to prove Theorems 3.1 and 3.2, as well as more general results (Theorems 3.1′ and 3.2′) that allow for misspecification error.

## E.1 OVERVIEW OF PROOF TECHNIQUES

In this section we give a proof sketch for Theorem 3.1, highlighting the role of truncated weight functions in addressing partial coverage. We focus on the regret bound; the sample complexity bound in Eq. (5) is an immediate consequence.

By design, the constraint in Eq. (3) ensures that $Q^\star \in \mathcal{F}^{(t)}$ for all $t \leq T$ with high probability. Thus, by a standard regret decomposition for optimistic algorithms (Lemma D.4), we have

$$\mathbf{Reg} = \sum_{t=1}^{T} J(\pi^\star) - J(\pi^{(t)}) \lesssim \sum_{t=1}^{T} \sum_{h=1}^{H} \underbrace{\mathbb{E}_{d_h^{(t)}} \big[ f_h^{(t)}(x_h, a_h) - [\mathcal{T} f_{h+1}^{(t)}](x_h, a_h) \big]}_{\text{On-policy Bellman error for } f^{(t)} \text{ under } \pi^{(t)}}, \qquad (13)$$

up to lower-order terms, where we abbreviate $d^{(t)} = d^{\pi^{(t)}}$. Defining $[\Delta_h f^{(t)}](x, a) := f_h^{(t)}(x, a) - [\mathcal{T}_h f_{h+1}^{(t)}](x, a)$, it remains to bound the on-policy expected bellman error $\mathbb{E}_{d_h^{(t)}}[[\Delta_h f^{(t)}](x_h, a_h)]$. To do so, a natural approach is to relate this quantity to the weighted off-policy Bellman error under $\bar{d}^{(t)} := \frac{1}{t-1} \sum_{i < t} d^{\pi^{(i)}}$ by introducing the weight function $d^{(t)}/\bar{d}^{(t)} \in \mathcal{W}$:

$$\mathbb{E}_{d_h^{(t)}}[[\Delta_h f^{(t)}](x_h, a_h)] \approx \mathbb{E}_{\bar{d}_h^{(t)}}\left[ [\Delta_h f^{(t)}](x_h, a_h) \cdot \frac{d_h^{(t)}(x_h, a_h)}{\bar{d}_h^{(t)}(x_h, a_h)} \right].$$

Unfortunately, this equality is not true as-is because the ratio $d^{(t)}/\bar{d}^{(t)}$ can be unbounded. We address this by replacing $\bar{d}^{(t)}$ by $\bar{d}^{(t+1)}$ throughout the analysis (at the cost of small approximation error), and work with the weight function $w_h^{(t)} := d_h^{(t)}/\bar{d}_h^{(t+1)} \in \mathcal{W}$, which is always bounded in magnitude $t$. However, while boundedness is a desirable property, the range $t$ is still too large to obtain non-vacuous concentration guarantees. This motivates us to introduce clipped/truncated weight functions via the following decomposition.

$$\underbrace{\mathbb{E}_{d_h^{(t)}}[[\Delta_h f^{(t)}](x_h, a_h)]}_{\text{On-policy Bellman error}} \leq \underbrace{\mathbb{E}_{\bar{d}_h^{(t+1)}}\big[ [\Delta_h f^{(t)}](x_h, a_h) \cdot \mathsf{clip}_{\gamma^{(t)}}\big[w_h^{(t)}\big](x_h, a_h) \big]}_{(A_t): \text{ Clipped off-policy Bellman error}} + \underbrace{\mathbb{E}_{d_h^{(t)}}\big[ \mathbb{I}\big\{ w_h^{(t)}(x_h, a_h) \geq \gamma^{(t)} \big\} \big]}_{(B_t): \text{ Loss due to clipping}}.$$

Recall that $\breve{w}_h^{(t)} := \mathsf{clip}_{\gamma^{(t)}}\big[w_h^{(t)}\big](x_h, a_h)$. As $w^{(t)} \in \mathcal{W}$, it follows from the constraint in Eq. (3) and Freedman-type concentration that the clipped Bellman error in term $(A_t)$ has order $\alpha^{(t)} \cdot \mathbb{E}_{\bar{d}^{(t+1)}}\big[(\breve{w}_h^{(t)})^2\big] + \beta^{(t)}$, so that $\sum_{t=1}^{T} A_t \leq \sum_{t=1}^{T} \alpha^{(t)} \cdot \mathbb{E}_{\bar{d}^{(t+1)}}\big[(\breve{w}_h^{(t)})^2\big] + \beta^{(t)}$. Since we clip to $\gamma^{(t)} = \gamma t$, we have $\sum_{t=1}^{T} \beta^{(t)} \lesssim \gamma \cdot T \log(|\mathcal{F}||\mathcal{W}|/\delta)$; bounding the sum of weight functions requires a more involved argument that we defer for a moment.

We now focus on bounding the terms $(B_t)$. Each term $(B_t)$ captures the extent to which the weighted off-policy Bellman error at iteration $t$ fails to approximate the true Bellman error due to clipping. This occurs when $\bar{d}^{(t+1)}$ has poor coverage relative to $d^{(t)}$, which happens when $\pi^{(t)}$ visits a portion of the state space not previously covered. We begin by applying Markov's inequality ($\mathbb{I}\{u \geq v\} \leq u/v$ for $u, v \geq 0$) to bound

$$B_t \leq \frac{1}{\gamma^{(t)}} \mathbb{E}_{d_h^{(t)}}\big[w_h^{(t)}(x_h, a_h)\big] = \frac{1}{\gamma^{(t)}} \mathbb{E}_{d_h^{(t)}}\left[ \frac{d_h^{(t)}(x_h, a_h)}{\bar{d}_h^{(t+1)}(x_h, a_h)} \right] = \frac{1}{\gamma} \mathbb{E}_{d_h^{(t)}}\left[ \frac{d_h^{(t)}(x_h, a_h)}{\widetilde{d}_h^{(t+1)}(x_h, a_h)} \right], \qquad (14)$$

where the equality uses that $\gamma^{(t)} := \gamma \cdot t$ and $\widetilde{d}^{(t+1)} := \bar{d}^{(t+1)} \cdot t$. Our most important insight is that even though each term in Eq. (14) might be large on a given iteration $t$ (if a previously unexplored portion of the state space is visited), *coverability* implies that on average across all iterations the error incurred by clipping must be small. In particular, using a variant of a coverability-based potential argument from Xie et al. (2023) (Lemma D.5), we show that

$$\sum_{t=1}^{T} \mathbb{E}_{d_h^{(t)}} \left[ \frac{d_h^{(t)}(x_h, a_h)}{\widetilde{d}_h^{(t+1)}(x_h, a_h)} \right] \leq O(C_{\mathsf{cov}} \cdot \log(T)),$$

so that $\sum_{t=1}^{T} B_t \leq \widetilde{O}(C_{\mathsf{cov}}/\gamma)$. To conclude the proof, we use an analogous potential argument to show the sum of weight functions in our bound on $\sum_{t=1}^{T} A_t$ also satisfies $\sum_{t=1}^{T} \alpha^{(t)} \cdot \mathbb{E}_{\bar{d}^{(t+1)}} \left[ (\check{w}_h^{(t)})^2 \right] \leq \widetilde{O}(C_{\mathsf{cov}}/\gamma)$. The intuition is similar: the squared weight functions (corresponding to variance of the weighted Bellman error) may be large in a given round, but cannot be large for all rounds under coverability. Altogether, combining the bounds on $A_t$ and $B_t$ gives $\mathbf{Reg} = \widetilde{O}\big( H \big( \frac{C_{\mathsf{cov}}}{\gamma} + \gamma \cdot T \log(|\mathcal{F}||\mathcal{W}|HT\delta^{-1}) \big) \big)$. The final result follows by choosing $\gamma > 0$ to balance the terms.

We find it interesting that the way in which this proof makes use of coverability—to handle the cumulative loss incurred by clipping—is quite different from the analysis in Xie et al. (2023), where it more directly facilitates a change-of-measure argument.

### E.2 SUPPORTING TECHNICAL RESULTS

For $x, x' \in \mathcal{X}$, $a \in \mathcal{A}$, $r \in [0, 1]$, and $h \in [H]$, recall the notation

$$[\widehat{\Delta}_h f](x, a, r, x') = f_h(x, a) - r - \max_{a'} f_h(x', a'),$$
$$[\Delta_h f](x, a) = f_h(x, a) - [\mathcal{T}_h f_{h+1}](x, a),$$
$$\check{w}_h(x, a) = \mathsf{clip}_{\gamma^{(t)}}[w_h](x, a).$$

**Lemma E.1** (Basic concentration for GLOW). *Let $\gamma^{(t)} \geq 0$ for $t \in [T]$. With probability at least $1 - \delta$, all of the following inequalities hold for all $f \in \mathcal{F}$, $w \in \mathcal{W}$, $t \in [T]$ and $h \in [H]$:*

$(a)$ $\left| \widehat{\mathbb{E}}_{\mathcal{D}_h^{(t)}} \left[ [\widehat{\Delta}_h f](x_h, a_h, r_h, x'_{h+1}) \cdot \check{w}_h(x_h, a_h) \right] - \mathbb{E}_{\bar{d}_h^{(t)}} \left[ [\Delta_h f](x_h, a_h) \cdot \check{w}_h(x_h, a_h) \right] \right| \leq \frac{10}{3\gamma^{(t)}} \mathbb{E}_{\bar{d}_h^{(t)}} \left[ (\check{w}_h(x_h, a_h))^2 \right] + \frac{\beta^{(t)}}{12}$

$(b)$ $\frac{1}{\gamma^{(t)}} \mathbb{E}_{\bar{d}_h^{(t)}} \left[ \check{w}_h^2(x_h, a_h) \right] \leq \frac{2}{\gamma^{(t)}} \widehat{\mathbb{E}}_{\mathcal{D}_h^{(t)}} \left[ \check{w}_h^2(x_h, a_h) \right] + \frac{2\beta^{(t)}}{9}$,

$(c)$ $\frac{1}{\gamma^{(t)}} \widehat{\mathbb{E}}_{\mathcal{D}_h^{(t)}} \left[ \check{w}_h^2(x_h, a_h) \right] \leq \frac{3}{2\gamma^{(t)}} \mathbb{E}_{\bar{d}_h^{(t)}} \left[ \check{w}_h^2(x_h, a_h) \right] + \frac{\beta^{(t)}}{9}$,

$(d)$ $J(\pi^\star) - \mathbb{E}_{x_1 \sim d_1}[f_1(x_1, \pi_{f_1}(x_1))] \leq \widehat{\mathbb{E}}_{x_1 \sim \mathcal{D}_1^{(t)}}[\max_a Q_1^\star(x_1, a_1) - f_1(x_1, \pi_{f_1}(x_1))] + \sqrt{\frac{8 \log(6|\mathcal{F}||\mathcal{W}|TH/\delta)}{K(t-1)}}$,

*where $\check{w}_h := \mathsf{clip}_{\gamma^{(t)}}[w_h]$ and $\beta^{(t)} := \frac{36\gamma^{(t)}}{K(t-1)} \log(6|\mathcal{F}||\mathcal{W}|TH/\delta)$.*

**Proof of Lemma E.1.** Fix any $h \in [H]$ and $t \in [T]$. Let $M = K(t-1)$ and recall that the dataset $\mathcal{D}_h^t$ consists of $M$ tuples of the form $\{(x_h^{(m)}, a_h^{(m)}, r_h^{(m)}, x_{h+1}^{(m)})\}_{m \leq M}$ where $x_{h+1}^{(m)} \sim P(\cdot \mid x_h^{(m)}, a_h^{(m)})$, and $a_h^{(m)} = \pi_{\tau(m)}(x_h^{(m)})$ where $\tau(m) = \lceil m/K \rceil$. Fix any $f \in \mathcal{F}$ and $w \in \mathcal{W}$.

**Proof of** $(a)$**.** For each $m \in [M]$, define the random variable

$$Y_m = [\widehat{\Delta}_h f](x_h^{(m)}, a_h^{(m)}, r_h^{(m)}, x_{h+1}^{(m)}) \cdot \check{w}_h(x_h^{(m)}, a_h^{(m)}) - \mathbb{E}_{d_h^{(\tau(m))}} [[\Delta_h f](x_h, a_h) \cdot \check{w}_h(x_h, a_h)]$$

Clearly, $\mathbb{E}_{m-1}[Y_m] = 0$ and thus $\{Y_m\}_{m \leq M}$ is a martingale difference sequence with

$$|Y_m| \leq 3 \sup_{x_h, a_h} |\check{w}_h(x_h, a_h)| \leq 3\gamma^{(t)},$$

since $|[\widehat{\Delta}_h f](x_h^{(m)}, a_h^{(m)}, r_h^{(m)}, x_{h+1}^{(m)})| \le 2$ and $|[\Delta_h f](x_h, a_h)| \le 1$. Furthermore,

$$
\begin{aligned}
\sum_{m=1}^{M} Y_m &= \sum_{m=1}^{M} [\widehat{\Delta}_h f](x_h^{(m)}, a_h^{(m)}, r_h^{(m)}, x_{h+1}^{(m)}) \cdot \check{w}_h(x_h^{(m)}, a_h^{(m)}) - \sum_{m=1}^{M} \mathbb{E}_{d_h^{(\tau(m))}}[[\Delta_h f](x_h, a_h) \cdot \check{w}_h(x_h, a_h)] \\
&= K(t-1)\widehat{\mathbb{E}}_{\mathcal{D}_h^{(t)}}\Big[[\widehat{\Delta}_h f](x_h, a_h, r_h, x'_{h+1}) \cdot \check{w}_h(x_h, a_h)\Big] - K\sum_{\tau=1}^{t-1} \mathbb{E}_{d_h^{(\tau)}}[[\Delta_h f](x_h, a_h) \cdot \check{w}_h(x_h, a_h)] \\
&= K(t-1)\widehat{\mathbb{E}}_{\mathcal{D}_h^{(t)}}\Big[[\widehat{\Delta}_h f](x_h, a_h, r_h, x'_{h+1}) \cdot \check{w}_h(x_h, a_h)\Big] - K(t-1)\mathbb{E}_{\bar{d}_h^{(t)}}[[\Delta_h f](x_h, a_h) \cdot \check{w}_h(x_h, a_h)].
\end{aligned}
$$

Additionally, we also have that

$$
\begin{aligned}
\mathbb{E}_{m-1}\big[(Y_m)^2\big] &\le 2\,\mathbb{E}_{m-1}\Big[([\widehat{\Delta}_h f](x_h^{(m)}, a_h^{(m)}, r_h^{(m)}, x_{h+1}^{(m)}) \cdot \check{w}_h(x_h^{(m)}, a_h^{(m)}))^2\Big] \\
&\qquad + 2\,\mathbb{E}_{m-1}\bigg[\Big(\mathbb{E}_{d_h^{(\tau(m))}}[[\Delta_h f](x_h, a_h) \cdot \check{w}_h(x_h, a_h)]\Big)^2\bigg] \\
&\le \mathbb{E}_{m-1}\Big[8\check{w}_h(x_h^{(m)}, a_h^{(m)})^2 + 2\,\mathbb{E}_{d_h^{(\tau(m))}}\Big[([\Delta_h f](x_h, a_h) \cdot \check{w}_h(x_h, a_h))^2\Big]\Big] \\
&\le \mathbb{E}_{m-1}\Big[8\check{w}_h(x_h^{(m)}, a_h^{(m)})^2 + 2\,\mathbb{E}_{d_h^{(\tau(m))}}\Big[(\check{w}_h(x_h, a_h))^2\Big]\Big] \\
&= 10\,\mathbb{E}_{d_h^{(\tau(m))}}\Big[(\check{w}_h(x_h, a_h))^2\Big]
\end{aligned}
$$

where the second line follows since $|[\widehat{\Delta}_h f](x_h^{(m)}, a_h^{(m)}, r_h^{(m)}, x_{h+1}^{(m)})| \le 2$ and by using Jensen's inequality, and the third line uses $|[\Delta_h f](x_h, a_h)| \le 1$.

Thus, using Lemma D.2 with $\eta = 1/3\gamma^{(t)}$, we get that with probability at least $1 - \delta'$,

$$
\begin{aligned}
\Big|\sum_{m=1}^{M} Y_m\Big| &= K(t-1)\Big|\widehat{\mathbb{E}}_{\mathcal{D}_h^{(t)}}\Big[[\widehat{\Delta}_h f](x_h, a_h, r_h, x'_{h+1}) \cdot \check{w}_h(x_h, a_h)\Big] - \mathbb{E}_{\bar{d}_h^{(t)}}[[\Delta_h f](x_h, a_h) \cdot \check{w}_h(x_h, a_h)]\Big| \\
&\le \frac{10K}{3\gamma^{(t)}}\sum_{\tau=1}^{t-1} \mathbb{E}_{d_h^{(\tau)}}\big[(\check{w}_h(x_h, a_h))^2\big] + 3\gamma^{(t)}\log(2/\delta') \\
&= \frac{10K(t-1)}{3\gamma^{(t)}}\,\mathbb{E}_{\bar{d}_h^{(t)}}\big[(\check{w}_h(x_h, a_h))^2\big] + 3\gamma^{(t)}\log(2/\delta').
\end{aligned}
$$

The above bound implies that

$$
\begin{aligned}
\Big|\widehat{\mathbb{E}}_{\mathcal{D}_h^{(t)}}\Big[[\widehat{\Delta}_h f](x_h, a_h, r_h, x'_{h+1}) \cdot \check{w}_h(x_h, a_h)\Big] &- \mathbb{E}_{\bar{d}_h^{(t)}}[[\Delta_h f](x_h, a_h) \cdot \check{w}_h(x_h, a_h)]\Big| \\
&\le \frac{10}{3\gamma^{(t)}}\,\mathbb{E}_{\bar{d}_h^{(t)}}\big[(\check{w}_h(x_h, a_h))^2\big] + \frac{3\gamma^{(t)}}{K(t-1)}\log(2/\delta').
\end{aligned}
$$

Plugging in the value of $\beta^{(t)}$ gives the desired bound. The final result follows by setting $\delta' = \delta/3|\mathcal{F}||\mathcal{W}|TH$, and taking another union bound over the choice of $f, w, t$ and $h$.

**Proof of** $(b)$ **and** $(c)$**.** For each $m \in [M]$, define the random variable

$$
Y_m = \big(\check{w}_h(x_h^{(m)}, a_h^{(m)})\big)^2.
$$

Clearly, the sequence $\{Y_m\}_{m \le M}$ is adapted to an increasing filtration, with $Y_t \ge 0$ and $|Y_t| = |(\check{w}_h(x_h^{(m)}, a_h^{(m)}))^2| \le (\gamma^{(t)})^2$. Furthermore,

$$
\sum_{m=1}^{M} \mathbb{E}_{m-1}[Y_m] = K\sum_{\tau=1}^{t-1} \mathbb{E}_{d_h^{(\tau)}}\big[(\check{w}_h(x_h, a_h))^2\big] = K(t-1)\mathbb{E}_{\bar{d}_h^{(t)}}\big[(\check{w}_h(x_h, a_h))^2\big],
$$

and,

$$
\sum_{m=1}^{M} Y_m = \sum_{(x,a) \in \mathcal{D}_h^{(t)}} (\check{w}_h(x_h, a_h))^2 = K(t-1)\widehat{\mathbb{E}}_{\mathcal{D}_h^{(t)}}\big[(\check{w}_h(x_h, a_h))^2\big].
$$

Thus, by Lemma D.3, we have that with probability at least $1 - \delta'$,

$$\mathbb{E}_{\bar{d}_h^{(t)}}\left[(\breve{w}_h(x_h, a_h))^2\right] \leq 2\widehat{\mathbb{E}}_{\mathcal{D}_h^{(t)}}\left[(\breve{w}_h(x_h, a_h))^2\right] + \frac{8(\gamma^{(t)})^2 \log(2/\delta')}{K(t-1)},$$

and

$$\widehat{\mathbb{E}}_{\mathcal{D}_h^{(t)}}\left[(\breve{w}_h(x_h, a_h))^2\right] \leq \frac{3}{2}\mathbb{E}_{\bar{d}_h^{(t)}}\left[(\breve{w}_h(x_h, a_h))^2\right] + \frac{4(\gamma^{(t)})^2 \log(2/\delta')}{K(t-1)}.$$

The final result follows by setting $\delta' = \delta/3|\mathcal{F}||\mathcal{W}|TH$, and taking another union bound over the choice of $f, w, t$ and $h$.

**Proof of** $(d)$. For each $m \in [M]$, define the random variable

$$Y_m = \max_a Q_1^\star(x_1^{(m)}, a) - f_1(x_1^{(m)}, \pi_{f_1}(x_1^{(m)})).$$

Clearly, $|Y_m| \leq 1$. Thus, using Lemma D.1, we get that with probability at least $1 - \delta'$,

$$\sum_{m=1}^{M} \mathbb{E}_{m-1}[Y_m] \leq \sum_{m=1}^{M}\left(\max_a Q_1^\star(x_1^{(m)}, a) - f_1(x_1^{(m)}, \pi_{f_1}(x_1^{(m)}))\right) + \sqrt{8M\log(2/\delta')}.$$

Setting $M = K(t-1)$ and noting that $\mathbb{E}_{m-1}[Y_m] = J(\pi^\star) - \mathbb{E}_{x_1 \sim d_1}[f_1(x_1, \pi_{f_1}(x_1)]$ since $x_1^{(m)} \sim d_1$ for any $m \in [M]$, we get that

$$J(\pi^\star) - \mathbb{E}_{x_1 \sim d_1}[f_1(x_1, \pi_{f_1}(x_1)] \leq \mathbb{E}_{x \sim \mathcal{D}_1^{(t)}}\left[\max_a Q_1^\star(x_1, a) - f_1(x_1, \pi_{f_1}(x_1))\right] + \sqrt{\frac{8\log(2/\delta')}{K(t-1)}}$$

The final result follows by setting $\delta' = \delta/3|\mathcal{F}||\mathcal{W}|TH$, and taking another union bound over the choice of $f, w, t$ and $h$. $\qquad\square$

**Lemma E.2** (Properties of GLOW confidence set). *Let $\gamma^{(t)} \geq 0$ for $t \in [T]$. With probability at least $1 - \delta$, all of the following events hold:*

($a$) *For all $t \geq 1$, $Q^\star \in \mathcal{F}^{(t)}$*

($b$) *For all $t \geq 2$, $h \in [H]$, $f \in \mathcal{F}^{(t)}$, and $w \in \mathcal{W}$, we have*

$$\mathbb{E}_{\bar{d}_h^{(t)}}[[\Delta_h f](x_h, a_h) \cdot \breve{w}_h(x_h, a_h)] \leq \frac{20}{\gamma^{(t)}}\mathbb{E}_{\bar{d}_h^{(t)}}\left[(\breve{w}_h(x_h, a_h))^2\right] + \frac{7\beta^{(t)}}{18}.$$

*Furthermore,*

$$\mathbb{E}_{\bar{d}_h^{(t+1)}}[[\Delta_h f](x_h, a_h) \cdot \breve{w}_h(x_h, a_h)] \leq \frac{40}{\gamma^{(t)}}\mathbb{E}_{\bar{d}_h^{(t+1)}}\left[(\breve{w}_h(x_h, a_h))^2\right] + \frac{7\beta^{(t)}}{9} + \frac{\gamma^{(t)}}{160t^2},$$

($c$) *For all $t \geq 2$, we have*

$$\mathbb{E}_{x_1 \sim d_1}\left[\max_a Q_1^\star(x_1, a) - f_1^{(t)}(x_1, \pi_1^{(t)}(x_1))\right] \leq \sqrt{\frac{8\log(6|\mathcal{F}||\mathcal{W}|TH/\delta)}{K(t-1)}},$$

*where $\breve{w}_h := \mathsf{clip}_{\gamma^{(t)}}[w_h]$ and $\beta^{(t)} = \frac{36\gamma^{(t)}}{K(t-1)}\log(6|\mathcal{F}||\mathcal{W}|TH/\delta)$.*

**Proof of Lemma E.2.** Using Lemma E.1, we have that with probability at least $1 - \delta$, for all $f \in \mathcal{F}$, $w \in \mathcal{W}$, $t \in [T]$ and $h \in [H]$,

$$\left|\widehat{\mathbb{E}}_{\mathcal{D}_h^{(t)}}\left[[\widehat{\Delta}_h f](x_h, a_h, r_h, x'_{h+1}) \cdot \breve{w}_h(x_h, a_h)\right] - \mathbb{E}_{\bar{d}_h^{(t)}}[[\Delta_h f](x_h, a_h) \cdot \breve{w}_h(x_h, a_h)]\right|$$

$$\leq \frac{10}{3\gamma^{(t)}}\mathbb{E}_{\bar{d}_h^{(t)}}\left[(\breve{w}_h(x_h, a_h))^2\right] + \frac{\beta^{(t)}}{12}, \tag{15}$$

$$\frac{1}{\gamma^{(t)}} \, \mathbb{E}_{\bar{d}_h^{(t)}} \left[ (\check{w}_h(x_h, a_h))^2 \right] \leq \frac{2}{\gamma^{(t)}} \widehat{\mathbb{E}}_{\mathcal{D}_h^{(t)}} \left[ (\check{w}_h(x_h, a_h))^2 \right] + \frac{2\beta^{(t)}}{9}, \tag{16}$$

$$\frac{1}{\gamma^{(t)}} \widehat{\mathbb{E}}_{\mathcal{D}_h^{(t)}} \left[ (\check{w}_h(x_h, a_h))^2 \right] \leq \frac{3}{2\gamma^{(t)}} \, \mathbb{E}_{\bar{d}_h^{(t)}} \left[ (\check{w}_h(x_h, a_h))^2 \right] + \frac{\beta^{(t)}}{9}, \tag{17}$$

and,

$$J(\pi^\star) - \mathbb{E}_{d_1}[f_1(x_1, \pi_{f_1}(x_1)] \leq \widehat{\mathbb{E}}_{\mathcal{D}_1^{(t)}} \left[ \max_a Q_1^\star(x_1, a) - f_1(x_1, \pi_{f_1}(x_1)) \right]$$
$$+ \sqrt{\frac{8 \log(6|\mathcal{F}||\mathcal{W}|TH/\delta)}{K(t-1)}}. \tag{18}$$

For the rest of the proof, we condition on the event in which (15-18) hold.

**Proof of** $(a)$. Consider any $t \in [T]$, and observe that the optimal state-action value function $Q^\star$ satisfies for any $w_h \in \mathcal{W}$,

$$\mathbb{E}_{\bar{d}_h^{(t)}} \left[ (Q_h^\star(x_h, a_h) - r_h - \max_{a'} Q_{h+1}^\star(x_{h+1}', a')) \cdot \check{w}_h(x_h, a_h) \right] = 0,$$

where $\check{w}_h := \mathsf{clip}_{\gamma^{(t)}}[w_h]$. Using the above relation with (15), we get that

$$\widehat{\mathbb{E}}_{\mathcal{D}_h^{(t)}} \left[ (Q_h^\star(x_h, a_h) - r_h - \max_{a'} Q_{h+1}^\star(x_{h+1}', a')) \cdot \check{w}_h(x_h, a_h) \right] \leq \frac{10}{3\gamma^{(t)}} \, \mathbb{E}_{\bar{d}_h^{(t)}} \left[ (\check{w}_h(x_h, a_h))^2 \right] + \frac{\beta^{(t)}}{12}$$
$$\leq \frac{4}{\gamma^{(t)}} \, \mathbb{E}_{\bar{d}_h^{(t)}} \left[ (\check{w}_h(x_h, a_h))^2 \right] + \frac{\beta^{(t)}}{12}$$
$$\leq \frac{8}{\gamma^{(t)}} \widehat{\mathbb{E}}_{\mathcal{D}_h^{(t)}} \left[ (\check{w}_h(x_h, a_h))^2 \right] + \beta^{(t)},$$

where the second-last inequality follows from (16).

Plugging in the values of $\alpha^{(t)}$ and $\beta^{(t)}$, rearranging the terms, we get that

$$\widehat{\mathbb{E}}_{\mathcal{D}_h^{(t)}} \left[ (Q_h^\star(x_h, a_h) - r_h - \max_{a'} Q_{h+1}^\star(x_{h+1}', a')) \cdot \check{w}_h(x_h, a_h) - \alpha^{(t)} (\check{w}_h(x_h, a_h))^2 \right] \leq \beta^{(t)}.$$

Since the above inequality holds for all $w \in \mathcal{W}$, we have that $Q^\star \in \mathcal{F}^{(t)}$.

**Proof of** $(b)$. Fix any $t$, and note that by the definition of $\mathcal{F}^{(t)}$, any $f \in \mathcal{F}^{(t)}$ satisfies for any $w \in \mathcal{W}$, the bound

$$\widehat{\mathbb{E}}_{\mathcal{D}_h^{(t)}} \left[ (f_h(x_h, a_h) - r_h - \max_{a'} f_{h+1}(x_{h+1}', a')) \cdot \check{w}_h(x_h, a_h) \right] \leq \frac{10}{\gamma^{(t)}} \widehat{\mathbb{E}}_{\mathcal{D}_h^{(t)}} \left[ (\check{w}_h(x_h, a_h))^2 \right] + \beta^{(t)}.$$

Using the above bound with (15), we get that

$$\mathbb{E}_{\bar{d}_h^{(t)}}[[\Delta_h f](x_h, a_h) \cdot \check{w}_h(x_h, a_h)] \leq \frac{10}{3\gamma^{(t)}} \, \mathbb{E}_{\bar{d}_h^{(t)}} \left[ (\check{w}_h(x_h, a_h))^2 \right] + \frac{10}{\gamma^{(t)}} \widehat{\mathbb{E}}_{\mathcal{D}_h^{(t)}} \left[ (\check{w}_h(x_h, a_h))^2 \right] + \frac{13}{12} \beta^{(t)}.$$

Plugging the bound from (17) for the second term above, we get that

$$\mathbb{E}_{\bar{d}_h^{(t)}}[[\Delta_h f](x_h, a_h) \cdot \check{w}_h(x_h, a_h)] \leq \frac{20}{\gamma^{(t)}} \, \mathbb{E}_{\bar{d}_h^{(t)}} \left[ (\check{w}_h(x_h, a_h))^2 \right] + \frac{7\beta^{(t)}}{18}.$$

Finally, noting that $\bar{d}^{(t+1)} = \frac{(t-1)\bar{d}^{(t)} + d^{(t)}}{t}$, we can further upper bound as:

$$\mathbb{E}_{\bar{d}_h^{(t+1)}}[[\Delta_h f](x_h, a_h) \cdot \check{w}_h(x_h, a_h)]$$
$$\leq \frac{t-1}{t} \left( \frac{20}{\gamma^{(t)}} \, \mathbb{E}_{\bar{d}_h^{(t)}} \left[ (\check{w}_h(x_h, a_h))^2 \right] + \frac{7\beta^{(t)}}{18} \right) + \frac{1}{t} \, \mathbb{E}_{d_h^{(t)}}[[\Delta_h f](x_h, a_h) \cdot \check{w}_h(x_h, a_h)]$$
$$\leq 2 \left( \frac{20}{\gamma^{(t)}} \, \mathbb{E}_{\bar{d}_h^{(t)}} \left[ (\check{w}_h(x_h, a_h))^2 \right] + \frac{7\beta^{(t)}}{18} \right) + \frac{1}{t} \, \mathbb{E}_{d_h^{(t)}}[|\check{w}_h(x_h, a_h)|]$$
$$\leq \frac{40}{\gamma^{(t)}} \, \mathbb{E}_{\bar{d}_h^{(t)}} \left[ (\check{w}_h(x_h, a_h))^2 \right] + \frac{7\beta^{(t)}}{9} + \frac{40}{\gamma^{(t)}} \, \mathbb{E}_{d_h^{(t)}} \left[ \check{w}_h(x_h, a_h)^2 \right] + \frac{\gamma^{(t)}}{160t^2}$$
$$= \frac{40}{\gamma^{(t)}} \, \mathbb{E}_{\bar{d}_h^{(t+1)}} \left[ (\check{w}_h(x_h, a_h))^2 \right] + \frac{7\beta^{(t)}}{9} + \frac{\gamma^{(t)}}{160t^2},$$

where the second-to-last line follows from an application of AM-GM inequality.

**Proof of** $(c)$. Plugging in $f = f^{(t)}$ in (18) and noting that $\max_a Q_1^\star(x_1, a) = Q_1^\star(x, \pi^\star(x))$ for any $x \in \mathcal{X}$, we get that

$$J(\pi^\star) - \mathbb{E}_{x_1 \sim d_1}\left[f_1^{(t)}(x_1, \pi_{f_1^{(t)}}(x_1))\right] \leq \widehat{\mathbb{E}}_{x_1 \sim \mathcal{D}_1^{(t)}}\left[Q_1^\star(x_1, \pi_1^\star(x_1)) - f_1^{(t)}(x_1, \pi_{f_1^{(t)}}(x_1))\right] + \sqrt{\frac{8\log(6|\mathcal{F}||\mathcal{W}|TH/\delta)}{K(t-1)}}.$$

However, note that by definition, $f^{(t)} \in \arg\max_f \widehat{\mathbb{E}}[f_1(x_1, \pi_{f_1}(x_1)]$, and using part-(a), $Q^\star \in \mathcal{F}^{(t)}$. Thus, $\widehat{\mathbb{E}}_{\mathcal{D}_1^{(t)}}\left[Q_1^\star(x_1, \pi_1^\star(x_1)) - f_1^{(t)}(x_1, \pi_{f_1^{(t)}}(x_1))\right] \leq 0$, which implies that

$$J(\pi^\star) - \mathbb{E}_{x_1 \sim d_1}\left[f_1^{(t)}(x_1, \pi_{f_1^{(t)}}(x_1))\right] \leq \sqrt{\frac{8\log(6|\mathcal{F}||\mathcal{W}|TH/\delta)}{K(t-1)}}.$$

$\square$

**Lemma E.3** (Coverability potential bound). *Let $d^{(1)}, \ldots, d^{(T)}$ be an arbitrary sequence of distributions over $\mathcal{X} \times \mathcal{A}$, such that there exists a distribution $\mu \in \Delta(\mathcal{X} \times \mathcal{A})$ that satisfies $\|d^{(t)}/\mu\|_\infty \leq C$ for all $(x, a) \in \mathcal{X} \times \mathcal{A}$ and $t \in [T]$. Then,*

$$\sum_{t=1}^T \mathbb{E}_{(x,a) \sim d^{(t)}}\left[\frac{d^{(t)}(x, a)}{\widetilde{d}^{(t+1)}(x, a)}\right] \leq 5C\log(T),$$

*where recall that $\widetilde{d}^{(t+1)} := \sum_{s=1}^t d^{(t)}$ for all $t \in [T]$.*

**Proof of Lemma E.3.** Let $\tau(x, a) = \min\{t \mid \widetilde{d}^{(t+1)}(x, a) \geq C\mu(x, a)\}$. With this definition, we can bound

$$\sum_{t=1}^T \mathbb{E}_{d^{(t)}}\left[\frac{d^{(t)}(x, a)}{\widetilde{d}^{(t+1)}(x, a)}\right] = \sum_{t=1}^T \mathbb{E}_{d^{(t)}}\left[\frac{d^{(t)}(x, a)}{\widetilde{d}^{(t+1)}(x, a)} \cdot \mathbb{I}\{t < \tau(x, a)\}\right]$$
$$+ \sum_{t=1}^T \mathbb{E}_{d^{(t)}}\left[\frac{d^{(t)}(x, a)}{\widetilde{d}^{(t+1)}(x, a)} \cdot \mathbb{I}\{t \geq \tau(x, a)\}\right]$$
$$\leq \underbrace{\sum_{t=1}^T \mathbb{E}_{d^{(t)}}[\mathbb{I}\{t < \tau(x, a)\}]}_{\text{(I): burn-in phase}} + \underbrace{\sum_{t=1}^T \mathbb{E}_{d^{(t)}}\left[\frac{d^{(t)}(x, a)}{\widetilde{d}^{(t+1)}(x, a)} \cdot \mathbb{I}\{t \geq \tau(x, a)\}\right]}_{\text{(II): stable phase}},$$

where the second line uses that $d^{(t)}(x,a)/\widetilde{d}^{(t+1)}(x,a) \leq 1$.

For the burn-in phase, note that

$$\sum_{t=1}^T \mathbb{E}_{d^{(t)}}[\mathbb{I}\{t < \tau(x, a)\}] = \sum_{x,a} \sum_{t < \tau(x,a)} d^t(x, a) = \sum_{x,a} \widetilde{d}^{(\tau(x,a))}(x, a) \leq \sum_{x,a} C\mu(x, a) = C,$$

where the last inequality uses that by definition, $\tilde{d}^t(x, a) \leq C\mu(x, a)$ for all $t \leq \tau(x, a)$.

For the stable phase, whenever $t \geq \tau(x, a)$, by definition, we have $\widetilde{d}^{(t+1)}(x, a) \geq C\mu(x, a)$ which implies that $\widetilde{d}^{(t+1)}(x, a) \geq \frac{1}{2}(\widetilde{d}^{(t+1)}(x, a) + C\mu(x, a))$. Thus,

$$\text{(II)} \leq 2\sum_{t=1}^T \mathbb{E}_{d^{(t)}}\left[\frac{d^{(t)}(x, a)}{\widetilde{d}^{(t)}(x, a) + C\mu(x, a)}\right] \tag{19}$$
$$= 2\sum_{x,a} \sum_{t=1}^T \frac{d^{(t)}(x, a) \cdot d^{(t)}(x, a)}{\widetilde{d}^{(t)}(x, a) + C\mu(x, a)}$$
$$\leq 2\sum_{x,a} \max_{t' \in [T]} d^{(t')}(x, a) \max_{x,a}\left(\sum_{t=1}^T \frac{d^{(t)}(x, a)}{\widetilde{d}^{(t)}(x, a) + C\mu(x, a)}\right).$$

Using the per-state elliptical potential lemma (Lemma D.5) in the above inequality, we get that

$$\text{(II)} \leq 4\log(T+1)\sum_{x,a}\max_{t'\in[T]}d^{(t')}(x,a) \leq 4C\log(T+1)\sum_{x,a}\mu(x,a) = 4C\log(1+T), \quad (20)$$

where the second inequality follows from the fact that $\left\|\frac{d^{(t)}}{\mu}\right\|_\infty \leq C$ (by definition), and the last equality uses that $\sum_{x,a}\mu(x,a) = 1$. Combining the above bound, we get that

$$\sum_{t=1}^{T}\mathbb{E}_{d^{(t)}}\left[\frac{d^{(t)}(x,a)}{\widetilde{d}^{(t+1)}(x,a)}\right] \leq 5C\log(T).$$

$\square$

### E.3   Main Technical Result: Bound on Cumulative Suboptimality for Glow

In this section we prove a key technical lemma, Lemma E.4, which gives a bound on the cumulative suboptimality of the sequence of policies generated by GLOW. Both the proofs of Theorem 3.2 and Theorem 3.1 build on this result. To facilitate more general sample complexity bounds that allow for misspecification error in $\mathcal{W}$. In particular, for each $t \in [T]$, we define $\xi_t$ as the misspecification error of the clipped density ratio $d_h^{(t)}/\bar{d}_h^{(t+1)}$ in class $\mathcal{W}_h$, defined as

$$\xi_t := \sup_{h\in[H]}\inf_{w\in\mathcal{W}_h}\sup_{\pi\in\Pi}\left\|\mathsf{clip}_{\gamma^{(t)}}\left[\frac{d_h^{(t)}}{\bar{d}_h^{(t+1)}}\right] - \mathsf{clip}_{\gamma^{(t)}}[w_h]\right\|_{1,\bar{d}_h^\pi}, \quad (21)$$

where recall that for any function $u : \mathcal{X} \times \mathcal{A} \mapsto \mathbb{R}$ and distribution $d \in \Delta(\mathcal{X} \times \mathcal{A})$, the norm $\|u\|_{1,d} := \mathbb{E}_{(x,a)\sim d}[|u(x,a)|]$. Note that under Assumption 2.2 or 2.2', $\xi_t = 0$ for all $t \in [T]$.

**Lemma E.4** (Bound on cumulative suboptimality). *Let $\pi^{(1)}, \ldots, \pi^{(T)}$ be the sequence of policies generated by GLOW, when executed on classes $\mathcal{F}$ and $\mathcal{W}$ with parameters $T, K$ and $\gamma$. Then the cumulative suboptimality of the sequence of policies $\{\pi^{(t)}\}_{t\in[T]}$ is bounded as*

$$\sum_{t=1}^{T}J(\pi^\star) - J(\pi^{(t)}) = O\left(H\left(\frac{C_{\mathsf{cov}}\log(1+T)}{\gamma} + \frac{\gamma T\log(|\mathcal{F}||\mathcal{W}|HT\delta^{-1})}{K} + \sum_{t=1}^{T}\xi_t + \gamma\log(T)\right)\right).$$

**Proof of Lemma E.4.** Fix any $t \geq 2$. We begin by establishing optimism as follows:

$$J(\pi^\star) - J(\pi^{(t)}) = \mathbb{E}_{x_1\sim d_1}\left[\max_a Q_1^\star(x_1,a)\right] - J(\pi^{(t)})$$

$$= \mathbb{E}_{x_1\sim d_1}\left[\max_a Q_1^\star(x_1,a) - f_1^{(t)}(x_1,\pi^{(t)}(x_1))\right] + \mathbb{E}_{x_1\sim d_1}\left[f_1^{(t)}(x_1,\pi^{(t)}(x_1))\right] - J(\pi^{(t)})$$

$$\leq \sqrt{\frac{8\log(6|\mathcal{F}||\mathcal{W}|TH/\delta)}{K(t-1)}} + \mathbb{E}_{x_1\sim d_1}\left[f_1^{(t)}(x_1,\pi^{(t)}(x_1))\right] - J(\pi^{(t)}),$$

where the last line follows from Lemma E.2-(c). Using Lemma D.4 for the second term, we get that

$$J(\pi^\star) - J(\pi^{(t)}) \leq \sqrt{\frac{8\log(6|\mathcal{F}||\mathcal{W}|TH/\delta)}{K(t-1)}} + \sum_{h=1}^{H}\mathbb{E}_{(x_h,a_h)\sim d_h^{(t)}}[[\Delta_h f^{(t)}](x_h,a_h)],$$

where recall that $[\Delta_h f^{(t)}](x_h,a_h) := f_h^{(t)}(x_h,a_h) - [\mathcal{T}_h f_{h+1}^{(t)}](x_h,a_h)$. Thus,

$$\sum_{t=1}^{T}J(\pi^\star) - J(\pi^{(t)}) \leq J(\pi^\star) - J(\pi^{(1)}) + \sum_{t=2}^{T}J(\pi^\star) - J(\pi^{(t)})$$

$$\leq 1 + \sum_{t=2}^{T}\sqrt{\frac{8\log(6|\mathcal{F}||\mathcal{W}|TH/\delta)}{K(t-1)}} + \sum_{t=2}^{T}\sum_{h=1}^{H}\mathbb{E}_{(x_h,a_h)\sim d_h^{(t)}}[[\Delta_h f^{(t)}](x_h,a_h)],$$

$$(22)$$

where the second inequality uses that $J(\pi^\star) \leq 1$ and $J(\pi^{(1)}) \geq 0$.

We next bound the expected Bellman error terms that appear in the right-hand-side above. Consider any $t \geq 2$ and $h \in [H]$, and note that via a straightforward change of measure,

$$\mathbb{E}_{d_h^{(t)}}[[\Delta_h f^{(t)}](x_h, a_h)] = \mathbb{E}_{\bar{d}_h^{(t+1)}}\left[[\Delta_h f^{(t)}](x_h, a_h) \cdot \frac{d_h^{(t)}(x_h, a_h)}{\bar{d}_h^{(t+1)}(x_h, a_h)}\right]$$

Since $u \leq \min\{u, v\} + u\mathbb{I}\{u \geq v\}$ for any $u, v$, we further decompose as

$$\mathbb{E}_{d_h^{(t)}}[[\Delta_h f^{(t)}](x_h, a_h)] \leq \underbrace{\mathbb{E}_{\bar{d}_h^{(t+1)}}\left[[\Delta_h f^{(t)}](x_h, a_h) \cdot \min\left\{\frac{d_h^{(t)}(x_h, a_h)}{\bar{d}_h^{(t+1)}(x_h, a_h)}, \gamma^{(t)}\right\}\right]}_{\text{(A): Expected clipped Bellman error}}$$

$$+ \underbrace{\mathbb{E}_{d_h^{(t)}}\left[\mathbb{I}\left\{\frac{d_h^{(t)}(x_h, a_h)}{\bar{d}_h^{(t+1)}(x_h, a_h)} \geq \gamma^{(t)}\right\}\right]}_{\text{(B): clipping violation}},$$

where in the second term we have changed the measure back to $d_h^{(t)}$ and used that $|[\Delta_h f^{(t)}](x_h, a_h)| \leq 1$. We bound the terms (A) and (B) separately below.

**Bound on expected clipped Bellman error.** Let $w_h^{(t)}(x_h, a_h) \in \mathcal{W}$ denote a weight function which satisfies

$$\sup_\pi \left\|\mathsf{clip}_{\gamma^{(t)}}\left[\frac{d_h^{(t)}}{\bar{d}_h^{(t+1)}}\right] - \mathsf{clip}_{\gamma^{(t)}}\left[w_h^{(t)}\right]\right\|_{1, d_h^\pi} \leq \xi_t, \tag{23}$$

which is guaranteed to exist by the definition of $\xi_t$. Then, we have

$$\text{(A)} = \mathbb{E}_{\bar{d}_h^{(t+1)}}\left[[\Delta_h f^{(t)}](x_h, a_h) \cdot \mathsf{clip}_{\gamma^{(t)}}\left[\frac{d_h^{(t)}(x_h, a_h)}{\bar{d}_h^{(t+1)}(x_h, a_h)}\right]\right]$$

$$\leq \mathbb{E}_{\bar{d}_h^{(t+1)}}\left[[\Delta_h f^{(t)}](x_h, a_h) \cdot \mathsf{clip}_{\gamma^{(t)}}\left[w_h^{(t)}(x_h, a_h)\right]\right] + \left\|\mathsf{clip}_{\gamma^{(t)}}\left[\frac{d_h^{(t)}}{\bar{d}_h^{(t+1)}}\right] - \mathsf{clip}_{\gamma^{(t)}}\left[w_h^{(t)}\right]\right\|_{1, \bar{d}_h^{(t+1)}}$$

$$\leq \mathbb{E}_{\bar{d}_h^{(t+1)}}\left[[\Delta_h f^{(t)}](x_h, a_h) \cdot \mathsf{clip}_{\gamma^{(t)}}\left[w_h^{(t)}(x_h, a_h)\right]\right] + \xi_t,$$

where the second line uses that $|[\Delta_h f^{(t)}](x_h, a_h)| \leq 1$, and the last line plugs in (23). Next, using Lemma E.2-(b) in the above inequality, we get that

$$\text{(A)} \leq \frac{40}{\gamma^{(t)}} \mathbb{E}_{\bar{d}_h^{(t+1)}}\left[\left(\mathsf{clip}_{\gamma^{(t)}}\left[w_h^{(t)}(x_h, a_h)\right]\right)^2\right] + \frac{7\beta^{(t)}}{9} + \frac{\gamma^{(t)}}{160t^2} + \xi_t. \tag{24}$$

Further splitting the first term, and using that $(a+b)^2 \leq 2a^2 + 2b^2$, we have that

$$\mathbb{E}_{\bar{d}_h^{(t+1)}}\left[\left(\mathsf{clip}_{\gamma^{(t)}}\left[w_h^{(t)}(x_h, a_h)\right]\right)^2\right]$$

$$\leq 2\mathbb{E}_{\bar{d}_h^{(t+1)}}\left[\left(\mathsf{clip}_{\gamma^{(t)}}\left[\frac{d_h^{(t)}(x_h, a_h)}{\bar{d}_h^{(t+1)}(x_h, a_h)}\right]\right)^2\right] + 2\left\|\mathsf{clip}_{\gamma^{(t)}}\left[\frac{d_h^{(t)}}{\bar{d}_h^{(t+1)}}\right] - \mathsf{clip}_{\gamma^{(t)}}\left[w_h^{(t)}\right]\right\|_{2, \bar{d}_h^{(t+1)}}^2$$

$$\leq 2\mathbb{E}_{\bar{d}_h^{(t+1)}}\left[\left(\mathsf{clip}_{\gamma^{(t)}}\left[\frac{d_h^{(t)}(x_h, a_h)}{\bar{d}_h^{(t+1)}(x_h, a_h)}\right]\right)^2\right] + 2\gamma^{(t)}\left\|\mathsf{clip}_{\gamma^{(t)}}\left[\frac{d_h^{(t)}}{\bar{d}_h^{(t+1)}}\right] - \mathsf{clip}_{\gamma^{(t)}}\left[w_h^{(t)}\right]\right\|_{1, \bar{d}_h^{(t+1)}}$$

$$\leq 2\mathbb{E}_{\bar{d}_h^{(t+1)}}\left[\left(\mathsf{clip}_{\gamma^{(t)}}\left[\frac{d_h^{(t)}(x_h, a_h)}{\bar{d}_h^{(t+1)}(x_h, a_h)}\right]\right)^2\right] + 2\gamma^{(t)}\xi_t,$$

where the second line holds since $\|w\|_{2,d}^2 \leq \|w\|_\infty \|w\|_{1,d}$ and $\mathsf{clip}_{\gamma^{(t)}}[w] \leq \gamma^{(t)}$ for any $w$, and the last line is due to (23). Using the above bound in (24), we get that

$$\text{(A)} \leq \frac{80}{\gamma^{(t)}} \mathbb{E}_{\bar{d}_h^{(t+1)}}\left[\left(\min\left\{\frac{d_h^{(t)}(x_h, a_h)}{\bar{d}_h^{(t+1)}(x_h, a_h)}, \gamma^{(t)}\right\}\right)^2\right] + 4\xi_t + 10\beta^{(t)} + \frac{\gamma^{(t)}}{80t^2}$$

$$\leq \frac{80}{\gamma^{(t)}} \mathbb{E}_{d_h^{(t)}}\left[\frac{d_h^{(t)}(x_h, a_h)}{\bar{d}_h^{(t+1)}(x_h, a_h)}\right] + 4\xi_t + 10\beta^{(t)} + \frac{\gamma^{(t)}}{80t^2}$$

$$= \frac{80}{\gamma} \mathbb{E}_{d_h^{(t)}} \left[ \frac{d_h^{(t)}(x_h, a_h)}{\widetilde{d}_h^{(t+1)}(x_h, a_h)} \right] + 4\xi_t + 10\beta^{(t)} + \frac{\gamma}{80t},$$

where the second line simply follows from a change of measure, and the last line holds since $\gamma^{(t)} = \gamma t$, and $\widetilde{d}^{(t+1)} = t\overline{d}^{(t+1)}$.

**Bound on clipping violation.** Since $\mathbb{I}\{u \geq v\} \leq \frac{u}{v}$ for any $u, v \geq 0$, we get that

$$(\mathrm{B}) \leq \frac{1}{\gamma^{(t)}} \mathbb{E}_{d_h^{(t)}} \left[ \frac{d_h^{(t)}(x_h, a_h)}{\widetilde{d}_h^{(t+1)}(x_h, a_h)} \right] = \frac{1}{\gamma} \mathbb{E}_{d_h^{(t)}} \left[ \frac{d_h^{(t)}(x_h, a_h)}{\widetilde{d}_h^{(t+1)}(x_h, a_h)} \right],$$

where the last line holds since $\gamma^{(t)} = \gamma t$.

Combining the bounds on the terms (A) and (B) above, and summing over the rounds $t = 2, \dots, T$, we get

$$\sum_{t=2}^{T} \mathbb{E}_{d_h^{(t)}}[[\Delta_h f^{(t)}](x_h, a_h)] \leq \frac{81}{\gamma} \sum_{t=2}^{T} \mathbb{E}_{d_h^{(t)}} \left[ \frac{d_h^{(t)}(x_h, a_h)}{\widetilde{d}_h^{(t+1)}(x_h, a_h)} \right] + 10 \sum_{t=2}^{T} \beta^{(t)} + 4 \sum_{t=2}^{T} \xi_t + \frac{\gamma}{80}, \quad (25)$$

For the first term, using Lemma E.3, along with the bound $\left\| d_h^{(t)} / \mu_h^\star \right\|_\infty \leq C_{\mathsf{cov}}$, we get that

$$\sum_{t=2}^{T} \mathbb{E}_{d_h^{(t)}} \left[ \frac{d_h^{(t)}(x_h, a_h)}{\widetilde{d}_h^{(t+1)}(x_h, a_h)} \right] \leq 5C_{\mathsf{cov}} \log(1 + T).$$

For the second term, we have

$$\sum_{t=2}^{T} \beta^{(t)} = \sum_{t=2}^{T} \frac{36\gamma t \log(6|\mathcal{F}||\mathcal{W}|HT\delta^{-1})}{K(t-1)} \leq \frac{72\gamma T \log(6|\mathcal{F}||\mathcal{W}|HT\delta^{-1})}{K}.$$

Combining these bounds, we get that

$$\sum_{t=2}^{T} \mathbb{E}_{d_h^{(t)}}[[\Delta_h f^{(t)}](x_h, a_h)] = O\left( \frac{C_{\mathsf{cov}} \log(1 + T)}{\gamma} + \frac{\gamma T \log(|\mathcal{F}||\mathcal{W}|HT\delta^{-1})}{K} + \sum_{t=1}^{T} \xi_t + \gamma \log(T) \right),$$
$$(26)$$

Plugging this bound in to (22) for each $h \in [H]$ gives the desired result.

$\square$

## E.4 PROOF OF THEOREM 3.1

In this section, we prove a generalization of Theorem 3.1 that accounts for misspecification error when the class $\mathcal{W}$ can only approximately realize the density ratios of mixed policies. Formally, we make the following assumption on the class $\mathcal{W}$:

**Assumption 2.2**[†] (Density ratio realizability, mixture version, with misspecification error). *Let $T$ be the parameter to* GLOW *(Algorithm 1). For all $h \in [H]$, $\pi \in \Pi$, $t \in [T]$, and $\pi^{(1:t)} = (\pi^{(1)}, \dots, \pi^{(t)}) \in \Pi$, there exists a weight function $w_h^{\pi;\pi^{(1:t)}}(x, a) \in \mathcal{W}_h$ such that*

$$\sup_{\widetilde{\pi} \in \Pi} \left\| \frac{d_h^\pi}{d_h^{\pi^{(1:t)}}} - w_h^{\pi;\pi^{(1:t)}} \right\|_{1, d_h^{\widetilde{\pi}}} \leq \varepsilon_{\mathrm{apx}}.$$

Note that setting $\varepsilon_{\mathrm{apx}} = 0$ above recovers Assumption 2.2′ given in the main body.

**Theorem 3.1′.** *Let $\varepsilon > 0$ be given, and suppose that Assumption 2.1 holds. Further, suppose that Assumption 2.2[†] (above) holds with $\varepsilon_{\mathrm{apx}} \leq \varepsilon/18H$. Then,* GLOW*, when executed on classes $\mathcal{F}$ and $\mathcal{W}$*

*with hyperparameters $T = \widetilde{\Theta}\big((H^2 C_{\mathsf{cov}}/\varepsilon^2) \cdot \log(|\mathcal{F}||\mathcal{W}|/\delta)\big)$, $K = 1$, and $\gamma = \sqrt{C_{\mathsf{cov}}/(T \log(|\mathcal{F}||\mathcal{W}|/\delta))}$ returns an $\varepsilon$-suboptimal policy $\widehat{\pi}$ with probability at least $1 - \delta$ after collecting*

$$N = \widetilde{O}\left( \frac{H^2 C_{\mathsf{cov}}}{\varepsilon^2} \log(|\mathcal{F}||\mathcal{W}|/\delta) \right) \tag{27}$$

*trajectories. In addition, for any $T \in \mathbb{N}$, with the same choice for $K$ and $\gamma$ as above, the algorithm enjoys a regret bound of the form*

$$\mathbf{Reg} := \sum_{t=1}^{T} J(\pi^\star) - J(\pi^{(t)}) = \widetilde{O}\big( H \sqrt{C_{\mathsf{cov}} T \log(|\mathcal{F}||\mathcal{W}|/\delta)} + HT\varepsilon_{\mathsf{apx}} \big). \tag{28}$$

Clearly, setting the misspecification error $\varepsilon_{\mathsf{apx}} = 0$ above recovers Theorem 3.1.

**Proof of Theorem 3.1′.** First note that by combining Assumption 2.2† with the fact that $\mathsf{clip}_\gamma[z]$ is 1-Lipschitz for any $\gamma > 0$, we have that for any $h \in [H]$ and $t \in [T]$, there exists a weight function $w_h^{(t)} \in \mathcal{W}_h$ such that

$$\sup_{\pi \in \Pi} \left\| \mathsf{clip}_{\gamma^{(t)}}\left[ \frac{d_h^{(t)}}{\overline{d}_h^{(t+1)}} \right] - \mathsf{clip}_{\gamma^{(t)}}\left[ w_h^{(t)} \right] \right\|_{1, d_h^\pi} \leq \varepsilon_{\mathsf{apx}}. \tag{29}$$

Using this misspecification bound, and setting $K = 1$ in Lemma E.4, we get that with probability at least $1 - \delta$,

$$\mathbf{Reg} = \sum_{t=1}^{T} J(\pi^\star) - J(\pi^{(t)}) = O\left( \frac{H C_{\mathsf{cov}} \log(1 + T)}{\gamma} + \gamma HT \log(6|\mathcal{F}||\mathcal{W}|HT\delta^{-1}) + HT\varepsilon_{\mathsf{apx}} \right).$$

Further setting $\gamma = \sqrt{C_{\mathsf{cov}}/(T \log(6|\mathcal{F}||\mathcal{W}|HT\delta^{-1}))}$ implies that

$$\mathbf{Reg} \leq O\left( H \sqrt{C_{\mathsf{cov}} T \log(T) \log(6|\mathcal{F}||\mathcal{W}|HT\delta^{-1})} + HT\varepsilon_{\mathsf{apx}} \right).$$

For the sample complexity bound, note that the returned policy $\widehat{\pi}$ is chosen via $\widehat{\pi} \sim \mathsf{Unif}(\{\pi^{(1)}, \ldots, \pi^{(T)}\})$, and thus

$$\mathbb{E}[J(\pi^\star) - J(\widehat{\pi})] = \frac{1}{T} \sum_{t=1}^{T} J(\pi^\star) - J(\pi^{(t)}) \leq O\left( H \sqrt{\frac{C_{\mathsf{cov}}}{T} \log(T) \log(6|\mathcal{F}||\mathcal{W}|HT\delta^{-1})} + H\varepsilon_{\mathsf{apx}} \right).$$

Hence, when $\varepsilon_{\mathsf{apx}} \leq O(\varepsilon/H)$, setting $T = \widetilde{\Theta}\left( \frac{H^2 C_{\mathsf{cov}}}{\varepsilon^2} \log(6|\mathcal{F}||\mathcal{W}|HT\delta^{-1}) \right)$ implies that the returned policy $\widehat{\pi}$ satisfies

$$\mathbb{E}[J(\pi^\star) - J(\widehat{\pi})] \leq \varepsilon.$$

The total number of trajectories collected to return an $\varepsilon$-suboptimal policy is given by

$$T \cdot K \leq \widetilde{O}\left( \frac{H^2 C_{\mathsf{cov}}}{\varepsilon^2} \log(6|\mathcal{F}||\mathcal{W}|H\delta^{-1}) \right).$$

$\square$

### E.5 PROOF OF THEOREM 3.2

In this section, we prove a generalization of Theorem 3.2 in Theorem 3.2′ (below) which accounts for misspecification error when the class $\mathcal{W}$ can only approximately realize the density ratios of pure policies. Formally, we make the following assumption on the class $\mathcal{W}$.

**Assumption 2.2‡** (Density ratio realizability, with misspecification error). *For any policy pair $\pi_1, \pi_2 \in \Pi$ and $h \in [H]$, there exists some weight function $w_h^{(\pi_1, \pi_2)} \in \mathcal{W}_h$ such that*

$$\sup_{\pi} \left\| \frac{d_h^{\pi_1}}{d_h^{\pi_2}} - w_h^{(\pi_1, \pi_2)} \right\|_{1, d_h^\pi} \leq \varepsilon_{\mathsf{apx}}.$$

Setting $\varepsilon_{\text{apx}} = 0$ above recovers Assumption 2.2 given in the main body.

Note that Assumption 2.2$^{\ddagger}$ only states that density ratios of pure policies are approximately realized by $\mathcal{W}_h$. On the other hand, the proof of Lemma E.4, our key tool in sample complexity analysis, requires (approximate) realizability for the ratio $d^{(t)}/\bar{d}^{(t+1)}$ in $\mathcal{W}_h$, which involves a mixture of occupancies. We fix this problem by running GLOW on a larger class $\overline{\mathcal{W}}$ that is constructed using $\mathcal{W}$ and has small misspecification error for $d^{(t)}/\bar{d}^{(t+1)}$ for all $t \leq T$. Before delving into the proof of Theorem 3.2', we first describe the class $\overline{\mathcal{W}}$.

**Construction of the class $\overline{\mathcal{W}}$.** Define an operator Mixture that takes in a sequence of weight functions $\{w^{(1)}, \ldots, w^{(t)}\}$ and a parameter $t \leq T$, and outputs a function $[\text{Mixture}(w^{(1)}, \ldots, w^{(t)}; t)]$ such that for any $x, a \in \mathcal{X} \times \mathcal{A}$,

$$[\text{Mixture}(w^{(1)}, \ldots, w^{(t)}; t)]_h(x, a) := \frac{1}{\mathbb{E}_{s \sim \text{Unif}([t])}\left[w_h^{(s)}(x, a)\right]}.$$

Using the operator Mixture, we define $\overline{\mathcal{W}}^{(t)}$ via

$$\overline{\mathcal{W}}^{(t)} = \{\text{Mixture}(w^{(1)}, \ldots, w^{(t)}; t) \mid w^{(1)}, \ldots, w^{(t)} \in \mathcal{W}\},$$

and then define

$$\overline{\mathcal{W}} = \cup_{t \leq T} \overline{\mathcal{W}}^{(t)}. \tag{30}$$

As a result of this construction, we have that

$$|\overline{\mathcal{W}}| \leq (|\mathcal{W}| + 1)^T \leq (2|\mathcal{W}|)^T. \tag{31}$$

In addition, we define $\overline{\mathcal{W}}_h = \{w_h \mid w \in \overline{\mathcal{W}}\}$. The following lemma shows that $\overline{\mathcal{W}}$ has small misspecification error for density ratios of mixture policies.

**Lemma E.5.** *Let $t \geq 0$ be given, and suppose Assumption 2.2$^{\ddagger}$ holds. For any sequence of policies $\pi^{(1)}, \ldots, \pi^{(t)} \in \Pi$, and $h \in [H]$, there exists a weight function $\bar{w}_h \in \overline{\mathcal{W}}_h^{(t)}$ such that for any $\gamma > 0$,*

$$\sup_{\pi} \left\| \text{clip}_\gamma\left[\frac{d_h^{(t)}}{\bar{d}_h^{(t+1)}}\right] - \text{clip}_\gamma[\bar{w}_h] \right\|_{1, d_h^\pi} \leq \gamma^2 \varepsilon_{\text{apx}},$$

*where recall that $\bar{d}_h^{(t+1)} = \frac{1}{t}\sum_{s=1}^t d_h^{(s)}$.*

The following theorem, which is our main sample complexity bound under misspecification error, is obtained by running GLOW on the weight function class $\overline{\mathcal{W}}$.

**Theorem 3.2'.** *Let $\varepsilon > 0$ be given, and suppose that Assumption 2.1 holds. Further, suppose that Assumption 2.2$^{\ddagger}$ holds with $\varepsilon_{\text{apx}} \leq \widetilde{O}(\varepsilon^5/C_{\text{cov}}^3 H^5)$. Then, GLOW, when executed on classes $\mathcal{F}$ and $\overline{\mathcal{W}}$ (defined in Eq. (30)) with hyperparameters $T = \widetilde{\Theta}(H^2 C_{\text{cov}}/\varepsilon^2)$, $K = \widetilde{\Theta}(T \log(|\mathcal{F}||\mathcal{W}|/\delta))$, and $\gamma = \sqrt{C_{\text{cov}}/T}$ returns an $\varepsilon$-suboptimal policy $\widehat{\pi}$ with probability at least $1 - \delta$ after collecting*

$$N = \widetilde{O}\left(\frac{H^4 C_{\text{cov}}^2}{\varepsilon^4} \log(|\mathcal{F}||\mathcal{W}|/\delta)\right).$$

*trajectories.*

Setting the misspecification error $\varepsilon_{\text{apx}} = 0$ above recovers Theorem 3.2 in the main body.

**Proof of Theorem 3.2'.** Using the misspecification bound from Lemma E.5 in Lemma E.4 implies that with probability at least $1 - \delta$,

$$\sum_{t=1}^T J(\pi^\star) - J(\pi^{(t)}) = O\left(\frac{HC_{\text{cov}} \log(1+T)}{\gamma} + \frac{\gamma HT \log(6|\mathcal{F}||\overline{\mathcal{W}}|HT\delta^{-1})}{K} + H\gamma^2 T^3 \varepsilon_{\text{apx}} + \gamma H \log(T)\right).$$

Using the relation in Eq. (31), we get that $|\overline{\mathcal{W}}| \leq (2|\mathcal{W}|)^T$ and thus

$$\sum_{t=1}^T J(\pi^\star) - J(\pi^{(t)}) = O\left(\frac{HC_{\text{cov}} \log(1+T)}{\gamma} + \frac{\gamma HT \log(6|\mathcal{F}||\overline{\mathcal{W}}|HT\delta^{-1})}{K} + H\gamma^2 T^3 \varepsilon_{\text{apx}} + \gamma H \log(T)\right).$$

Setting $K = 2T \log(6|\mathcal{F}||\mathcal{W}|HT\delta^{-1})$ and $\gamma = \sqrt{\frac{C_{\text{cov}}}{T}}$ in the above bound, we get

$$\sum_{t=1}^{T} J(\pi^\star) - J(\pi^{(t)}) \leq O\Big(\Big(H\sqrt{C_{\text{cov}}T}\log(T) + HC_{\text{cov}}T^2\varepsilon_{\text{apx}}\Big)\Big).$$

Finally, observing that the returned policy $\widehat{\pi} \sim \mathsf{Unif}(\{\pi^{(1)}, \ldots, \pi^{(T)}\})$, we get

$$\sum_{t=1}^{T} J(\pi^\star) - J(\pi^{(t)}) \leq O\Bigg(\Bigg(H\sqrt{\frac{C_{\text{cov}}}{T}}\log(T) + HC_{\text{cov}}T^2\varepsilon_{\text{apx}}\Bigg)\Bigg).$$

Thus, when $\varepsilon_{\text{apx}} \leq \widetilde{O}(\varepsilon^5/C_{\text{cov}}^3 H^5)$, setting $T = \widetilde{\Theta}\big(\frac{C_{\text{cov}}}{\varepsilon^2}\log^2(\frac{C_{\text{cov}}}{\varepsilon^2})\big)$ in the above bound implies that
$$\mathbb{E}[J(\pi^\star) - J(\widehat{\pi})] \leq \varepsilon.$$
The total number of trajectories collected to return $\varepsilon$-suboptimal policy is given by:

$$T \cdot K = O\bigg(\frac{H^4 C_{\text{cov}}^2}{\varepsilon^4}\log(|\mathcal{F}||\mathcal{W}|HT\delta^{-1})\bigg).$$

$\square$

**Proof of Lemma E.5.** Fix any $h \in [H]$ and $t \in [T]$. Using Assumption 2.2[‡], we have that for any $s \leq t$, there exists a function $w_h^{(s,t)} \in \mathcal{W}_h$ such that

$$\sup_{\pi \in \Pi}\bigg\|\frac{d_h^{(s)}}{d_h^{(t)}} - w_h^{(s,t)}\bigg\|_{1,d_h^\pi} \leq \varepsilon_{\text{apx}}. \tag{32}$$

Let $\bar{w}_h = \big[\mathsf{Mixture}(w^{(1,t)}, \ldots, w^{(t,t)}; t)\big]_h \in \overline{\mathcal{W}}_h$, and recall that $\bar{d}^{(t+1)} = \mathbb{E}_{s\sim\mathsf{Unif}([t])}\big[d_h^{(s)}\big]$. For any $x, a \in \mathcal{X} \times \mathcal{A}$, define

$$\zeta(x,a) := \Bigg|\min\Bigg\{\frac{1}{\mathbb{E}_{s\sim\mathsf{Unif}([t])}\big[d_h^{(s)}(x,a)/d_h^{(t)}(x,a)\big]}, \gamma\Bigg\} - \min\Bigg\{\frac{1}{\mathbb{E}_{s\sim\mathsf{Unif}([t])}\big[w^{(s,t)}(x,a)\big]}, \gamma\Bigg\}\Bigg|.$$

Using that the function $g(z) = \min\big\{\frac{1}{z}, \gamma\big\} = \frac{1}{\max\{z,\gamma^{-1}\}}$ is $\gamma^2$-Lipschitz, we get that

$$\zeta(x,a) \leq \gamma^2 \cdot \Bigg|\mathbb{E}_{s\sim\mathsf{Unif}([t])}\Bigg[\frac{d_h^{(s)}(x,a)}{d_h^{(t)}(x,a)}\Bigg] - \mathbb{E}_{s\sim\mathsf{Unif}([t])}\big[w_h^{(s,t)}(x,a)\big]\Bigg|$$

$$\leq \gamma^2 \mathbb{E}_{s\sim\mathsf{Unif}([t])}\Bigg[\Bigg|\frac{d_h^{(s)}(x,a)}{d_h^{(t)}(x,a)} - w_h^{(s,t)}(x,a)\Bigg|\Bigg],$$

where the last inequality follows from the linearity of expectation, and by using Jensen's inequality.
Thus,

$$\sup_{\pi \in \Pi} \mathbb{E}_{x_h, a_h \sim d_h^\pi}\Bigg[\Bigg|\min\Bigg\{\frac{d_h^{(t)}(x_h, a_h)}{\bar{d}_h^{(t+1)}(x_h, a_h)}, \gamma\Bigg\} - \min\{\bar{w}_h(x_h, a_h), \gamma\}\Bigg|\Bigg]$$

$$= \sup_{\pi \in \Pi} \mathbb{E}_{x_h, a_h \sim d_h^\pi}[\zeta(x_h, a_h)]$$

$$\leq \gamma^2 \sup_{\pi \in \Pi} \mathbb{E}_{s\sim\mathsf{Unif}([t])}\Bigg[\bigg\|\frac{d_h^{(s)}}{d_h^{(t)}} - w_h^{(s,t)}\bigg\|_{1,d_h^\pi}\Bigg]$$

$$\leq \gamma^2 \mathbb{E}_{s\sim\mathsf{Unif}([t])} \sup_{\pi \in \Pi}\Bigg[\bigg\|\frac{d_h^{(s)}}{d_h^{(t)}} - w_h^{(s,t)}\bigg\|_{1,d_h^\pi}\Bigg]$$

$$\leq \gamma^2 \varepsilon_{\text{apx}},$$

where the second-to-last line is due to Jensen's inequality, and the last line follows from the bound in (32). $\square$

# F    Proofs and Additional Results from Section 4 (Hybrid RL)

This section is organized as follows:

- Appendix F.1 gives additional details and further examples of offline algorithms with which we can apply $H_2O$, including an example that uses FQI as the base algorithm.

- Appendix F.2 includes a detailed comparison between HYGLOW and purely offline algorithms based on single-policy concentrability.

- Appendix F.3 includes a discussion on generic reductions from offline to online RL using a class of algorithms we call *optimistic offline RL algorithms*.

- Appendix F.4 contains the proof of the main result for $H_2O$, Theorem 4.1.

- Appendix F.5 contains supporting proofs for the offline RL algorithms (Appendix F.1) we use within $H_2O$.

## F.1    Examples for $H_2O$

This section contains additional examples of base algorithms that can be applied within the $H_2O$ reduction.

### F.1.1    HYGLOW Algorithm

For completeness, we state full pseudocode for the HYGLOW algorithm described in Section 4.2 as Algorithm 3. As described in the main body, this algorithm simply invokes $H_2O$ with a clipped and regularized variant of the MABO algorithm (MABO.CR, Eq. (9)) as the base algorithm.

We will invoke MABO.CR with a certain augmented weight function class $\overline{\mathcal{W}}$, which we now define. For any $w \in \mathcal{W}$, let $w^{(h)} \in (\mathcal{X} \times \mathcal{A} \times [H] \to \mathbb{R} \cup \{+\infty\})$ be defined by

$$w_{h'}^{(h)}(x,a) := \begin{cases} w_h(x,a) & \text{if } h = h' \\ 0 & \text{if } h \neq h' \end{cases}, \tag{33}$$

and let $\mathcal{W}^{(h)} = \{w^{(h)} \mid w \in \mathcal{W}\}$. Define the set

$$\overline{\mathcal{W}} := \mathcal{W} \cup (-\mathcal{W}) \cup_{h \in [H]} (\mathcal{W}^{(h)} \cup (-\mathcal{W}^{(h)})). \tag{34}$$

Note that the size satisfies $\overline{\mathcal{W}}$ is $|\overline{\mathcal{W}}| \leq 2(H+1)|\mathcal{W}| \leq 4H|\mathcal{W}|$.

We recall that the MABO.CR algorithm with dataset $\mathcal{D}$ and parameters $\gamma, \mathcal{F}, \mathcal{W}$ is defined via

$$\widehat{f} \in \arg\min_{f \in \mathcal{F}} \max_{w \in \mathcal{W}} \sum_{h=1}^{H} \left| \widehat{\mathbb{E}}_{\mathcal{D}_h} \left[ \check{w}_h(x_h, a_h)[\widehat{\Delta}_h f](x_h, a_h, r_h, x'_{h+1}) \right] \right| - \alpha^{(n)} \widehat{\mathbb{E}}_{\mathcal{D}_h} \left[ \check{w}_h^2(x_h, a_h) \right], \tag{35}$$

where $\alpha^{(n)} := 8/\gamma^{(n)}$ and $\check{w}_h := \mathsf{clip}_{\gamma^{(n)}}[w_h]$.

In Appendix F.5.2 we will prove the following result.

**Theorem F.1** (MABO.CR is CC-bounded). *Let $\mathcal{D} = \{\mathcal{D}_h\}_{h=1}^{H}$ consist of $H \cdot n$ samples from $\mu^{(1)}, \ldots, \mu^{(n)}$. For any $\gamma \in \mathbb{R}_+$, the MABO.CR algorithm (Eq. (9)) with parameters $\mathcal{F}$, augmented class $\overline{\mathcal{W}}$ defined in Eq. (34) in Appendix F.1.1, and $\gamma$ is CC-bounded at scale $\gamma$ under the Assumption that $Q^\star \in \mathcal{F}$ and that for all $\pi \in \Pi$ and $h \in [H]$, $d_h^\pi / \mu_h^{(1:n)} \in \mathcal{W}$.*

The risk bound for HYGLOW will follow by Theorem 4.1. Namely, we have the following.

**Theorem 4.2** (Risk bound for HYGLOW). *Let $\varepsilon > 0$ be given, let $\mathcal{D}_{\mathsf{off}}$ consist of $H \cdot T$ samples from data distribution $\nu$, and suppose that $\nu$ satisfies $C_\star$-single-policy concentrability. Suppose that $Q^\star \in \mathcal{F}$ and that for all $t \in [T]$, $\pi \in \Pi$, and $h \in [H]$, we have $d_h^\pi / \mu_h^{(t)} \in \mathcal{W}$, where $\mu_h^{(t)} := 1/2(\nu_h + 1/t \sum_{i=1}^{t} d_h^{\pi^{(i)}})$. Then, HYGLOW (Algorithm 3) with inputs $T = \widetilde{\Theta}((H^4(C_{\mathsf{cov}} + C_\star)/\varepsilon^2) \cdot \log(|\mathcal{F}||\mathcal{W}|/\delta))$, $\mathcal{F}$, augmented $\overline{\mathcal{W}}$ defined in Eq. (34), $\gamma = \widetilde{\Theta}\left(\sqrt{(C_\star + C_{\mathsf{cov}})/TH^2 \log(|\mathcal{F}||\mathcal{W}|/\delta)}\right)$, and $\mathcal{D}_{\mathsf{off}}$ returns an $\varepsilon$-suboptimal policy with probability at least $1 - \delta T$ after collecting $N = \widetilde{O}\left(\frac{H^2(C_{\mathsf{cov}} + C_\star)}{\varepsilon^2} \log(|\mathcal{F}||\mathcal{W}|/\delta)\right)$ trajectories.*

---

**Algorithm 3** HYGLOW: $H_2O$ + MABO.CR

---

   **input:** Parameter $T \in \mathbb{N}$, value function class $\mathcal{F}$, weight function class $\mathcal{W}$, parameter $\gamma \in [0, 1]$, offline datasets $\mathcal{D}_{\mathrm{off}} = \{\mathcal{D}_{\mathrm{off},h}\}_h$ each of size $T$.

1: Set $\gamma^{(t)} = \gamma \cdot t$, $\alpha^{(t)} = 4/\gamma^{(t)}$ and $\beta^{(t)} = (12\gamma/K) \cdot \log(6|\mathcal{F}||\mathcal{W}|TH/\delta)$.

2: Initialize $\mathcal{D}_{\mathrm{on},h}^{(1)} = \mathcal{D}_{\mathrm{hybrid},h}^{(1)} = \varnothing$ for all $h \in [H]$.

3: **for** $t = 1, \ldots, T$ **do**

4:     Compute value function $f^{(t)}$ such that

$$f^{(t)} \in \underset{f \in \mathcal{F}}{\arg\min} \max_{w \in \mathcal{W}} \sum_{h=1}^{H} \left| \widehat{\mathbb{E}}_{\mathcal{D}_{\mathrm{hybrid},h}^{(t)}} \left[ \check{w}_h(x_h, a_h)[\widehat{\Delta}_h f](x_h, a_h, r_h, x'_{h+1}) \right] \right| - \alpha^{(n)} \widehat{\mathbb{E}}_{\mathcal{D}_{\mathrm{hybrid},h}^{(t)}} \left[ \check{w}_h^2(x_h, a_h) \right],$$

(36)

        where $\check{w} := \mathsf{clip}_{\gamma^{(t)}}[w]$ and $[\widehat{\Delta}_h f](x, a, r, x') := f_h(x, a) - r - \max_{a'} f_{h+1}(x', a')$.

5:     Compute policy $\pi^{(t)} \leftarrow \pi_{f^{(t)}}$.

6:     Collect $(x_1, a_1, r_1), \ldots, (x_H, a_H, r_H)$ using $\pi^{(t)}$; $\mathcal{D}_{\mathrm{on},h}^{(t+1)} := \mathcal{D}_{\mathrm{on},h}^{(t)} \cup \{(x_h, a_h, r_h, x_{h+1})\}$.

7:     Aggregate offline and online data: $\mathcal{D}_{\mathrm{hybrid},h}^{(t+1)} := \mathcal{D}_{\mathrm{off},h}\big|_{1:t} \cup \mathcal{D}_{\mathrm{on},h}^{(t+1)}$ for all $h \in [H]$.

8: **output:** $\widehat{\pi} = \mathsf{Unif}(\pi^{(1)}, \ldots, \pi^{(T)})$.

---

### F.1.2 COMPUTATIONALLY EFFICIENT IMPLEMENTATION FOR HYGLOW

In the following, we expand on the discussion after Theorem F.1 regarding computationally efficient implementation of the optimization problem in Eq. (9) and Eq. (36) via reparameterization. Let $\gamma > 0$ and $n > 0$ be given to the learner, and suppose that in addition to the class $\mathcal{W}$, the learner has access to a function class $\mathcal{W}^{(\gamma,n)}$ that satisfies the following assumption.

**Assumption F.1.** *The function class $\mathcal{W}^{(\gamma,n)}$ satisfies*

   *(a) For all $w \in \mathcal{W}^{(\gamma,n)}$ and $h \in [H]$, $\|w_h\|_\infty \leq \gamma n$.*

   *(b) For all $h \in [H]$, $\{\pm\mathsf{clip}_{\gamma n}[w] \mid w \in \mathcal{W}\} \subseteq \mathcal{W}^{(\gamma,n)}$.*

   *(c) For all $h \in [H]$, $\{\pm\mathsf{clip}_{\gamma n}[w^{(h)}] \mid w \in \mathcal{W}\} \subseteq \mathcal{W}^{(\gamma,n)}$, where $w^{(h)}$ is defined as in Eq. (33).*

Note that $\mathcal{W}^{(\gamma,n)}$ also satisfies the density ratio realizability required by MABO.CR (cf. Theorem F.1). We claim that optimizing directly over the class $\mathcal{W}^{(\gamma,n)}$, which does not involve explicitly clipping, leads to the same guarantee as solving Eq. (9). In more detail, consider the following offline RL algorithm, which given offline datasets $\{\mathcal{D}_h\}_{h \leq H}$, returns

$$\widehat{f} \in \underset{f \in \mathcal{F}}{\arg\min} \max_{w \in \mathcal{W}^{(\gamma,n)}} \sum_{h=1}^{H} \widehat{\mathbb{E}}_{\mathcal{D}_h} \left[ w_h(x_h, a_h)[\widehat{\Delta}_h f](x_h, a_h, r_h, x'_{h+1}) \right] - \alpha^{(n)} \widehat{\mathbb{E}}_{\mathcal{D}_h} \left[ w_h^2(x_h, a_h) \right].$$

(37)

Using the next lemma, we show that the function obtained by solving Eq. (37) leads to the same offline RL guarantee as MABO.CR when $|\log(\mathcal{W}^{(\gamma,n)})| = O(\log(|\mathcal{W}|) + \mathrm{poly}(H))$. In particular, substituting the bound from Lemma F.1 in place of the corresponding bound from Lemma F.6 in the proof of Theorem F.1, while keeping rest of the analysis same, shows that the above described computationally efficient implementation of MABO.CR is also CC-bounded. Using this fact with Theorem 4.1 implies the desired performance guarantee for HYGLOW (similar to Theorem 4.2).

**Lemma F.1.** *Suppose $\mathcal{F}$ and $\mathcal{W}$ satisfy Assumption 2.1 and Assumption F.4. Additionally, let $\gamma \in (0, 1)$ and $n > 0$ be given constants, and for $h \in [H]$, let $\mathcal{D}_h$ be datasets of size $n$ sampled from the offline distribution $\mu_h^{(1:n)}$. Furthermore, let $\mathcal{W}^{(\gamma,n)}$ be a reparameterized function class that satisfies Assumption F.1 w.r.t. $\mathcal{W}$. Then, the function $\widehat{f}$ returned by Eq. (37), when executed with datasets $\{\mathcal{D}_h\}_{h \leq H}$, weight function class $\mathcal{W}^{(\gamma,n)}$, and parameters $\alpha^{(n)} = \frac{8}{\gamma n}$, satisfies with probability at least $1 - \delta$,*

$$\sum_{h=1}^{H} \left| \mathbb{E}_{\mu_h^{(1:n)}} \left[ [\Delta_h \widehat{f}](x_h, a_h) \cdot \check{w}_h(x_h, a_h) \right] \right| \leq \mathcal{O}\left( \sum_{h=1}^{H} \frac{1}{\gamma n} \mathbb{E}_{\mu_h^{(1:n)}} \left[ (\check{w}_h(x_h, a_h))^2 \right] + H^2 \beta^{(n)} \right),$$

*for all $w \in \mathcal{W}$, where $\beta^{(n)} = O\left( \frac{\gamma n}{n-1} \log(24|\mathcal{F}||\mathcal{W}|H^2/\delta) \right)$.*

**Proof of Lemma F.1.** Repeating the same arguments as in the proof of Lemma E.2 with $K = 1$ (we avoid repeating the arguments for conciseness), along with the fact that $\|w'_h\|_\infty \leq \gamma n$ for all $h \in [H]$ and $w' \in \mathcal{W}^{(\gamma,n)}$, we get that the returned function $\widehat{f}$ satisfies for all $w' \in \mathcal{W}^{(\gamma,n)}$,

$$\sum_{h=1}^{H} \mathbb{E}_{\mu_h^{(1:n)}}\left[[\Delta_h \widehat{f}](x_h, a_h) \cdot w'_h(x_h, a_h)\right] \leq \mathcal{O}\left(\sum_{h=1}^{H} \frac{1}{\gamma n} \mathbb{E}_{\mu_h^{(1:n)}}\left[(w'_h(x_h, a_h))^2\right] + \frac{H\gamma n}{n-1} \log(|\mathcal{F}||\mathcal{W}^{(\gamma,n)}|H/\delta)\right).$$

However, as in the proof of Lemma F.6, since for any $w \in \mathcal{W}$ we have that both $\mathsf{clip}_{\gamma n}[w^{(h)}] \in \mathcal{W}_h^{(\gamma,n)}$ and $-\mathsf{clip}_{\gamma n}[w^{(h)}] \in \mathcal{W}_h^{(\gamma,n)}$ for every $h \in [H]$, the above inequality immediately implies that for every $w \in \mathcal{W}$,

$$\sum_{h=1}^{H} \left|\mathbb{E}_{\mu_h^{(1:n)}}\left[[\Delta_h \widehat{f}](x_h, a_h) \cdot \check{w}_h(x_h, a_h)\right]\right| \leq \mathcal{O}\left(\sum_{h=1}^{H} \frac{1}{\gamma n} \mathbb{E}_{\mu_h^{(1:n)}}\left[(\check{w}_h(x_h, a_h))^2\right] + \frac{H^2\gamma n}{n-1} \log(|\mathcal{F}||\mathcal{W}^{(\gamma,n)}|H/\delta)\right),$$

where we used that the RHS is independent of the sign. The final statement follows by plugging in that $|\log(\mathcal{W}^{(\gamma,n)})| = O(\log(|\mathcal{W}|) + \mathrm{poly}(H))$. $\qquad\square$

### F.1.3 FITTED Q-ITERATION

In this section, we apply H$_2$O with the Fitted Q-Iteration (FQI) algorithm (Munos, 2007; Munos and Szepesvári, 2008; Chen and Jiang, 2019) as the base algorithm. For an offline dataset $\mathcal{D}$ and value function class $\mathcal{F}$, the FQI algorithm is defined as follows:

**Algorithm F.2** (Fitted Q-Iteration (FQI)).

1. *Set $\widehat{f}_{H+1}(x, a) = 0$ for all $(x, a)$*

2. *For $h = H, \ldots, 1$:*

$$\widehat{f}_h \in \arg\min_{f_h \in \mathcal{F}_h} \widehat{\mathbb{E}}_{\mathcal{D}_h}\left[(f_h(x_h, a_h) - r_h - \max_{a'} \widehat{f}_{h+1}(x_{h+1}, a')))^2\right].$$

3. *Output $\widehat{\pi} = \pi_{\widehat{f}}$.*

We analyze FQI under the following standard Bellman completeness assumption.

**Assumption F.2** (Bellman completeness). *For all $h \in [H]$, we have that $\mathcal{T}_h \mathcal{F}_{h+1} \subseteq \mathcal{F}_h$*

**Theorem F.3.** *The FQI algorithm is CC-bounded under Assumption F.2 with scaling functions $\mathfrak{a}_\gamma = \frac{6}{\gamma}$ and $\mathfrak{b}_\gamma = 1024 \log(2n|\mathcal{F}|)\gamma$, for all $\gamma > 0$ simultaneously. As a consequence, when invoked within H$_2$O, we have $\mathbf{Risk} \leq \widetilde{O}\left(H\sqrt{(C_\star + C_{\mathsf{cov}}) \log(|\mathcal{F}|\delta^{-1})/T}\right)$ with probability at least $1 - \delta T$.*

The proof can be found in Appendix F.5.3.

The full pseudocode for H$_2$O with FQI is essentially identical to the Hy-Q algorithm of Song et al. (2023), except for a slightly different data aggregattion strategy in Line 7. Thus, Hy-Q can be interpreted as a special case of the H$_2$O algorithm when instantiated with FQI as a base algorithm. The risk bounds in Song et al. (2023) are proven under an a structural condition known as bilinear rank (Du et al., 2021), which is complementary to coverability. Our result recovers a special case of a risk bound for Hy-Q given in the follow-up work of Liu et al. (2023), which analyzes Hy-Q under coverability instead of bilinear rank.

### F.1.4 MODEL-BASED MLE

In this section we apply H$_2$O with Model-Based Maximum Likelihood Estimation (MLE) algorithm as the base algorithm. The algorithm is parameterized by a model class $\mathcal{M} = \{\mathcal{M}_h\}_{h=1}^{H}$, where $\mathcal{M}_h \subset \{M_h : \mathcal{X} \times \mathcal{A} \to \Delta(\mathbb{R} \times \mathcal{X})\}$. Each model $M = \{M_h\}_{h=1}^{H} \in \mathcal{M}$ has the same state space, action space, initial distribution, and horizon, and each $M_h \in \mathcal{M}_h$ is a conditional distribution over rewards and next states for layer $h$. For a dataset $\mathcal{D}$, the algorithm proceeds as follows.

**Algorithm F.4** (Model-Based MLE).

- *For $h \in [H]$:*

– *Compute the maximum likelihood estimator for layer $h$ as*

$$\widehat{M}_h = \arg\max_{M_h \in \mathcal{M}_h} \sum_{(x_h, a_h, r_h, x_{h+1}) \in \mathcal{D}_h} \log(M_h(r_h, x_{h+1} \mid x_h, a_h)) \tag{38}$$

- *Output $\pi^\star_{\widehat{M}}$, the optimal policy for $\widehat{M} = \{\widehat{M}_h\}_h$*

We analyze Model-Based MLE under a standard realizability assumption.

**Assumption F.3** (Model realizability)**.** *We have that $M^\star \in \mathcal{M}$.*

**Theorem F.5.** *The model-based MLE algorithm is* CC*-bounded under Assumption F.3 for all $\gamma > 0$ simultaneously, with scaling functions $\mathfrak{a}_\gamma = \frac{6}{\gamma}$ and $\mathfrak{b}_\gamma = 8\log(|\mathcal{M}|H/\delta)\gamma$. As a consequence, when invoked within* $H_2O$*, we have* $\mathbf{Risk} \le \widetilde{O}\big(H\sqrt{(C_\star + C_{\mathsf{cov}})\log(|\mathcal{M}|H\delta^{-1})/T}\big)$ *with probability at least $1 - \delta T$.*

The proof can be found in Appendix F.5.4.

## F.2   Comparison to Offline RL

In this section, we compare the performance of HYGLOW to existing results in purely offline RL which assume access to a data distribution $\nu$ with single-policy concentrability (Definition 4.3)

Let $\mu$ be the offline data distribution. Let us write $w^\pi := {}^{d^\pi}/\mu$, $w^\star := {}^{d^{\pi^\star}}/\mu$ and $V^\star$ for the optimal value function . The most relevant work is the PRO-RL algorithm of Zhan et al. . Their algorithm is computationally efficient and establishes a polynomial sample complexity bound under the realizability of certain *regularized versions* of $w^\star$ and $V^\star$. By contrast, our result requires $Q^\star$-realizability and the density ratio realizability of $w^\pi$ for all $\pi \in \Pi$, but for the unregularized problem. These assumptions are not comparable, as either may hold without the other. Their sample complexity result is also slightly larger, scaling roughly as $\widetilde{O}\big(\frac{H^6 C_\star^4 C_{\star,\varepsilon}^2}{\varepsilon^6}\big)$, where $C_{\star,\varepsilon}$ is the single-policy concentrability for the regularized problem, as opposed to our $\widetilde{O}\big({}^{H^2(C_\star + C_{\mathsf{cov}})}/\varepsilon^2\big)$. However, our approach requires additional online access while their algorithm does not.

To the best of our knowledge, all other algorithms for the purely offline setting that only require single-policy concentrability either need stronger representation conditions (such as value-function completeness Xie et al. (2021a)), or are not known to be computationally efficient in the general function approximation setting due to the need for implementing pessimism (e.g., Chen and Jiang (2022)).

## F.3   Generic Reductions from Online to Offline RL?

Our hybrid-to-offline reduction $H_2O$ and the CC-boundedness definition also shed light on the question of when offline RL methods can be lifted to the *purely online setting*. Indeed, observe that any offline algorithm which satisfies CC-boundedness (Definition 4.2) with only a $\widehat{\pi}$-coverage term, namely which satisfies an offline risk bound of the form

$$\mathbf{Risk}_{\mathsf{off}} \le \sum_{h=1}^{H} \frac{\mathfrak{a}_\gamma}{n} \mathbb{E}_{\widehat{\pi} \sim p}[\mathsf{CC}_h(\widehat{\pi}, \mu^{(1:n)}, \gamma n)] + \mathfrak{b}_\gamma, \tag{39}$$

can be repeatedly invoked within $H_2O$ (with $\mathcal{D}_{\mathsf{off}} = \varnothing$) to achieve a small $\sqrt{C_{\mathsf{cov}}/T}$-type risk bound for the purely online setting, with no hybrid data. This can be seen immediately by inspecting our proof for the hybrid setting (Theorem F.6 and Theorem 4.1).

We can think of algorithms satisfying Eq. (39) as *optimistic offline RL algorithms*, since their risk only scales with a term depending on their own output policy; this is typically achieved using optimism. In particular, it is easy to see, that GLOW and GOLF (Jin et al., 2021a; Xie et al., 2023) can be interpreted as repeatedly invoking such an optimistic offline RL algorithm within the $H_2O$ reduction. This class of algorithms has not been considered in the offline RL literature since they inherit both the computational drawbacks of pessimistic algorithms and the statistical drawbacks of "neutral" (i.e. non-pessimistic) algorithms (at least, when viewed only in the context of offline RL).

In more detail, as with pessimism, optimism is often not computationally efficient, although it furthermore requires all-policy concentrability (as opposed to single-policy concentrability) to

obtain low *offline* risk. On the other hand, neutral (non-pessimistic) algorithms such as FQI (Chen and Jiang, 2019) and MABO (Xie and Jiang, 2020) also require all-policy concentrability, but are more computationally efficient. However, our reduction shows that these algorithms might merit further investigation. In particular, it uncovers that they can automatically solve the online setting (without hybrid data) under coverability and when repeatedly invoked on datasets generated from their previous policies. We find that this reduction advances the fundamental understanding of sample-efficient algorithms in both the online and offline settings, and are optimistic that this understanding can be used for future algorithm design.

### F.4   PROOFS FOR H$_2$O (THEOREM 4.1)

The following theorem is a slight generalization of Theorem 4.1. In the sequel, we prove Theorem 4.1 as a consequence of this result.

**Theorem F.6.** *Let $T \in \mathbb{N}$ be given, let $\mathcal{D}_{\mathrm{off}}$ consist of $H \cdot T$ samples from data distribution $\nu$. Let $\mathbf{Alg}_{\mathrm{off}}$ be CC-bounded at scale $\gamma \in [0,1]$ under Assumption$(\cdot)$, with parameters $\mathfrak{a}_\gamma$ and $\mathfrak{b}_\gamma$. Suppose that $\forall\ t \in [T]$ and $\pi^{(1)}, \dots, \pi^{(t)} \in \Pi$, Assumption$(\mu^{(t)}, M^\star)$ holds for $\mu^{(t)} := \{1/2(\nu_h + 1/t \sum_{i=1}^{t} d_h^{\pi^{(i)}})\}_{h=1}^{H}$. Then, with probability at least $1 - \delta T$, the risk of H$_2$O (Algorithm 2) with $T$, $\mathbf{Alg}_{\mathrm{off}}$, and $\mathcal{D}_{\mathrm{off}}$ is bounded as[13]*

$$\mathbf{Risk} \leq \frac{2\mathfrak{a}_\gamma}{T} \sum_{h=1}^{H} \sum_{t=1}^{T} \frac{1}{t} \mathsf{CC}_h(\pi^\star, \nu, \gamma^{(t)}) + \underbrace{\widetilde{O}\left( H\left( \frac{C_{\mathsf{cov}}\mathfrak{a}_\gamma}{N} + \mathfrak{b}_\gamma \right) \right)}_{=:\mathsf{err}_{\mathrm{off}}}. \tag{40}$$

**Proof of Theorem F.6.** Recall the definitions $d_h^{(t)} := d_h^{\pi^{(t)}}$, $\widetilde{d}_h^{(t)} := \sum_{s=1}^{t-1} d_h^{(t)}$, and $\bar{d}_h^{(t)} := \frac{1}{t-1} \widetilde{d}_h^{(t)}$. Furthermore, let $d_h^\star := d_h^{\pi^\star}$ for all $h \leq H$. Note that the data distribution for $\mathcal{D}_{\mathrm{hybrid}}^{(t)}$ is $\mu^{(t)} = \{\mu_h^{(t)}\}_{h=1}^{H}$ where $\mu_h^{(t)} = \frac{1}{2}(\nu_h + \bar{d}_h^{(t)})$, where $\nu_h$ is the offline distribution. As a result of Definition 4.2, the offline algorithm $\mathbf{Alg}_{\mathrm{off}}$ invoked on the dataset $\mathcal{D}_{\mathrm{hybrid}}^{(t)}$ outputs a distribution $p_t \sim \Delta(\Pi)$ that satisfies the bound:

$$\mathbb{E}_{\pi^{(t)} \sim p_t}\left[ J(\pi^\star) - J(\pi^{(t)}) \right] \leq \sum_{h=1}^{H} \frac{\mathfrak{a}_\gamma}{t} \left( \mathsf{CC}_h(\pi^\star, \mu^{(t)}, \gamma t) + \mathbb{E}_{\pi^{(t)} \sim p_t}[\mathsf{CC}_h(\pi^{(t)}, \mu^{(t)}, \gamma t)] \right) + \mathfrak{b}_\gamma, \tag{41}$$

with probability at least $1 - \delta$, where $\mathfrak{a}_\gamma$ and $\mathfrak{b}_\gamma$ are the scaling functions for which $\mathbf{Alg}_{\mathrm{off}}$ is CC-bounded at scale $\gamma$. By taking a union bound over $T$, the number of iterations, we have that the event in Eq. (41) occurs for all $t \leq T$ with probability greater than $1 - \delta T$.

Plugging in the definition for the clipped concentrability coefficient above and summing over $t$ from $1, \dots, T$, we get that

$$\mathbf{Reg} = \mathbb{E}\left[ \sum_{t=1}^{T} J(\pi^\star) - J(\pi^{(t)}) \right]$$

$$\leq \sum_{h=1}^{H} \left( \underbrace{\sum_{t=1}^{T} \frac{\mathfrak{a}_\gamma}{t} \left\| \mathsf{clip}_{\gamma t}\left[ \frac{d_h^{\pi^\star}}{\mu_h^{(t)}} \right] \right\|_{1, d_h^\star}}_{(\mathrm{I})} + \underbrace{\sum_{t=1}^{T} \frac{\mathfrak{a}_\gamma}{t} \mathbb{E}_{\pi^{(t)} \sim p_t}\left[ \left\| \mathsf{clip}_{\gamma t}\left[ \frac{d_h^{(t)}}{\mu_h^{(t)}} \right] \right\|_{1, d_h^{(t)}} \right]}_{(\mathrm{II})} \right) + HT\mathfrak{b}_\gamma.$$

For each $h \in [H]$, we bound the two terms I and II separately below.

**Term (I).** Note that $\mu_h^{(t)}(x, a) \geq \nu_h(x, a)/2$ for any $x, a$. Thus,

$$(\mathrm{I}) \leq 2\mathfrak{a}_\gamma \sum_{t=1}^{T} \frac{1}{t} \left\| \mathsf{clip}_{\gamma t}\left[ \frac{d_h^{\pi^\star}}{\nu_h} \right] \right\|_{1, d_h^{\pi^\star}} = 2\mathfrak{a}_\gamma \sum_{t=1}^{T} \frac{1}{t} \mathsf{CC}_h(\pi^\star, \nu, \gamma t).$$

---

[13]We define risk for the hybrid setting as in Eq. (1). Our result is stated as a bound on the risk to the optimal policy $\pi^\star$, but extends to give a bound on the risk of any comparator $\pi$ with $C_\star$ replaced by coverage for $\pi$.

**Term (II).**  We bound this term uniformly for any $\pi^{(t)} \sim p_t$. So, fix $\pi^{(t)}$ and note that

$$
\text{(II)} = \mathfrak{a}_\gamma \sum_{t=1}^{T} \frac{1}{t} \mathbb{E}_{d_h^{(t)}} \left[ \min\left\{ \frac{d_h^{(t)}(x,a)}{\mu_h^{(t)}(x,a)}, \gamma t \right\} \right]
$$

$$
= \mathfrak{a}_\gamma \left( \underbrace{\sum_{t=1}^{T} \frac{1}{t} \mathbb{E}_{d_h^{(t)}} \left[ \frac{d^{(t)}(x,a)}{\mu_h^{(t)}(x,a)} \mathbb{I}\left\{ \frac{d^{(t)}(x,a)}{\mu_h^{(t)}(x,a)} \le \gamma t \right\} \right]}_{\text{(II.A)}} + \underbrace{\sum_{t=1}^{T} \frac{1}{t} \mathbb{E}_{d_h^{(t)}} \left[ \gamma t \cdot \mathbb{I}\left\{ \frac{d^{(t)}(x,a)}{\mu_h^{(t)}(x,a)} > \gamma t \right\} \right]}_{\text{(II.B)}} \right),
$$

where the second line holds since $\min\{u,v\} = u\mathbb{I}\{u \le v\} + v\mathbb{I}\{v < u\}$ for all $u,v \in \mathbb{R}$.

In order to bound the two terms appearing above, we use certain properties of coverability, similar to the analysis of GLOW (Appendix E). For a parameter $\lambda \in (0,1)$, let us define a burn-in time

$$
\tau_h^{(\lambda)}(x,a) = \min\left\{ t \mid \widetilde{d}_h^{(t)}(x,a) \ge \frac{C_{\text{cov}} \cdot \mu_h^\star(x,a)}{\lambda} \right\}, \tag{42}
$$

and observe that

$$
\sum_{t=1}^{T} \mathbb{E}_{d_h^{(t)}}[\mathbb{I}\{t < \tau_h^{(\lambda)}(x,a)\}] = \sum_{x,a} \sum_{t < \tau_h^{(\lambda)}(x,a)} d_h^{(t)}(x,a) \le \frac{2C_{\text{cov}}}{\lambda}, \tag{43}
$$

which holds for any $\lambda \in (0,1)$. This bound can be derived by noting that

$$
\sum_{x,a} \sum_{t < \tau_h^{(\lambda)}(x,a)} d_h^{(t)}(x,a) = \sum_{x,a} \tilde{d}^{(\tau_h^{(\lambda)}(x,a))}(x,a)
$$

$$
= \sum_{x,a} \tilde{d}^{(\tau_h^{(\lambda)}(x,a)-1)}(x,a) + \sum_{x,a} d^{(\tau_h^{(\lambda)}(x,a))}(x,a)
$$

$$
\le \sum_{x,a} \frac{C_{\text{cov}}}{\lambda} \mu_h^\star(x,a) + \sum_{x,a} C_{\text{cov}} \mu_h^\star(x,a)
$$

$$
\le C_{\text{cov}} \left( \frac{1}{\lambda} + 1 \right) \le \frac{2C_{\text{cov}}}{\lambda}.
$$

We also recall the follow bound, which is a corollary of the elliptical potential lemma (Lemma D.5):

$$
\sum_{t=1}^{T} \sum_{x,a} d_h^{(t)}(x,a) \frac{d_h^{(t)}(x,a)}{\widetilde{d}^{(t)}(x,a)} \mathbb{I}\{t > \tau_h^{(\lambda)}(x,a)\} \le \sum_{t=1}^{T} \sum_{x,a} d_h^{(t)}(x,a) \frac{d_h^{(t)}(x,a)}{\widetilde{d}^{(t)}(x,a)} \mathbb{I}\{t > \tau_h^{(1)}(x,a)\} \overset{(i)}{\le} 5\log(T)C_{\text{cov}}. \tag{44}
$$

The inequality (i) can be seen derived by noting that, under the event in the indicator, we have $\widetilde{d}_h^{(t)}(x,a) \ge C_{\text{cov}}\mu_h^\star(x,a)$ and thus $\widetilde{d}_h^{(t)}(x,a) \ge \frac{1}{2}(C_{\text{cov}}\mu_h^\star(x,a) + \widetilde{d}^{(t)}(x,a))$. This gives

$$
\sum_{t=1}^{T} \sum_{x,a} d_h^{(t)}(x,a) \frac{d_h^{(t)}(x,a)}{\widetilde{d}^{(t)}(x,a)} \mathbb{I}\{t > \tau_h^{(1)}(x,a)\} \le 2 \sum_{t=1}^{T} \sum_{x,a} d_h^{(t)}(x,a) \frac{d_h^{(t)}(x,a)}{\widetilde{d}^{(t)}(x,a) + C_{\text{cov}}\mu_h^\star(x,a)},
$$

from which we can repeat the steps from Eq. (19) to Eq. (20).

**Term (II.A).**  To bound this term, we introduce a split according to the burn-in time $\tau_h^{(1)}(x,a)$, i.e.

$$
\text{(II.A)} = \sum_{t=1}^{T} \frac{1}{t} \mathbb{E}_{d_h^{(t)}} \left[ \frac{d_h^{(t)}(x,a)}{\mu_h^{(t)}(x,a)} \mathbb{I}\left\{ \frac{d_h^{(t)}(x,a)}{\mu_h^{(t)}(x,a)} \le \gamma t \right\} \left( \mathbb{I}\{t \le \tau_h^{(1)}(x,a)\} + \mathbb{I}\{t > \tau_h^{(1)}(x,a)\} \right) \right].
$$

The first term is bounded via

$$
\sum_{t=1}^{T} \frac{1}{t} \mathbb{E}_{d_h^{(t)}} \left[ \frac{d_h^{(t)}(x,a)}{\mu_h^{(t)}(x,a)} \mathbb{I}\left\{ \frac{d_h^{(t)}(x,a)}{\mu_h^{(t)}(x,a)} \le \gamma t \right\} \mathbb{I}\{t \le \tau_h^{(1)}(x,a)\} \right] \le \gamma \sum_{t=1}^{T} \mathbb{E}_{d_h^{(t)}} \left[ \mathbb{I}\{t \le \tau_h^{(1)}(x,a)\} \right]
$$

$$
\le 2\gamma C_{\text{cov}},
$$

by Eq. (43) with $\lambda = 1$. The second term is bounded via:

$$\sum_{t=1}^{T} \frac{1}{t} \mathbb{E}_{d_h^{(t)}} \left[ \frac{d_h^{(t)}(x,a)}{\mu_h^{(t)}(x,a)} \mathbb{I}\left\{ \frac{d^{(t)}(x,a)}{\mu_h^{(t)}(x,a)} \leq \gamma t \right\} \mathbb{I}\{t > \tau_h^{(1)}(x,a)\} \right] \leq 2 \sum_{t=1}^{T} \frac{1}{t} \mathbb{E}_{d_h^{(t)}} \left[ \frac{d_h^{(t)}(x,a)}{\bar{d}_h^{(t)}(x,a)} \mathbb{I}\{t > \tau_h^{(1)}(x,a)\} \right]$$

$$\leq 2 \sum_{t=1}^{T} \mathbb{E}_{d_h^{(t)}} \left[ \frac{d_h^{(t)}(x,a)}{\widetilde{d}_h^{(t)}(x,a)} \mathbb{I}\{t > \tau_h^{(1)}(x,a)\} \right]$$

$$\leq 10 \log(T) C_{\mathsf{cov}},$$

by using that $\mu_h^{(t)}(x,a) \geq \frac{\bar{d}_h^{(t)}(x,a)}{2}$ and Equation (44).

Adding these two terms together gives us the upper bound $\text{(II.A)} \leq 2\gamma C_{\mathsf{cov}} + 10 \log(T) C_{\mathsf{cov}}$.

**Term (II.B).** We have

$$\sum_{t=1}^{T} \frac{1}{t} \mathbb{E}_{d_h^{(t)}} \left[ \gamma t \mathbb{I}\left\{ \frac{d_h^{(t)}(x,a)}{\mu_h^{(t)}(x,a)} > \gamma t \right\} \right] = \gamma \sum_{t=1}^{T} \mathbb{E}_{d_h^{(t)}} \left[ \mathbb{I}\left\{ \frac{d_h^{(t)}(x,a)}{\mu_h^{(t)}(x,a)} > \gamma t \right\} \right]$$

$$\overset{(i)}{\leq} \gamma \sum_{t=1}^{T} \mathbb{E}_{d_h^{(t)}} \left[ \mathbb{I}\left\{ \frac{C_{\mathsf{cov}} \mu_h^{\star}(x,a)}{\widetilde{d}_h^{(t)}(x,a)} > \frac{\gamma}{2} \right\} \right]$$

$$\overset{(ii)}{=} \gamma \sum_{t=1}^{T} \mathbb{E}_{d_h^{(t)}} \left[ \mathbb{I}\{t \leq \tau_h^{(\gamma/2)}(x,a)\} \right]$$

$$\leq \gamma \cdot \frac{4 C_{\mathsf{cov}}}{\gamma} = 4 C_{\mathsf{cov}},$$

where the inequality (i) follows from applying the upper bounds $d_h^{(t)}(x,a) \leq C_{\mathsf{cov}} \mu_h^{\star}(x,a)$ and $\mu_h^{(t)}(x,a) \geq \frac{1}{2} \bar{d}_h^{(t)}(x,a)$, and the inequality (ii) follows from the definition of the burn-in time (Eq. (42)) with $\lambda = \gamma/2$.

Combining all the bounds above, we get that

$$\text{(II)} \leq 2\mathfrak{a}_\gamma \left( 2\gamma C_{\mathsf{cov}} + 10 \log(T) C_{\mathsf{cov}} + 2 C_{\mathsf{cov}} \right)$$
$$= 4\mathfrak{a}_\gamma C_{\mathsf{cov}} \left( \gamma + 10 \log(T) + 1 \right).$$

Adding together the terms so far, we can conclude the regret bound:

$$\mathbf{Reg} \leq 2\mathfrak{a}_\gamma \sum_{h=1}^{H} \sum_{t=1}^{T} \frac{1}{t} \mathsf{CC}_h(\pi^\star, \nu, \gamma t) + H \left( 4\mathfrak{a}_\gamma C_{\mathsf{cov}} \left( \gamma + 10 \log(T) + 1 \right) + T \mathfrak{b}_\gamma \right).$$

It follows that the policy $\widehat{\pi} = \mathsf{Unif}(\pi^{(1)}, \ldots, \pi^{(T)})$ satisfies the risk bound

$$\mathbf{Risk} \leq \frac{2\mathfrak{a}_\gamma}{T} \sum_{h=1}^{H} \sum_{t=1}^{T} \frac{1}{t} \mathsf{CC}_h(\pi^\star, \nu, \gamma t) + \underbrace{H \left( 4 \frac{\mathfrak{a}_\gamma C_{\mathsf{cov}}}{T} \left( \gamma + 10 \log(T) + 1 \right) + \mathfrak{b}_\gamma \right)}_{:= \mathsf{err}_{\mathsf{off}}}$$

$$= \frac{2\mathfrak{a}_\gamma}{T} \sum_{h=1}^{H} \sum_{t=1}^{T} \frac{1}{t} \mathsf{CC}_h(\pi^\star, \nu, \gamma t) + \widetilde{O}\left( H\left( \frac{C_{\mathsf{cov}} \mathfrak{a}_\gamma}{T} + \mathfrak{b}_\gamma \right) \right),$$

with probability at least $1 - \delta T$, where in the last line we have used that $\gamma \in [0,1]$. $\qquad\square$

We now prove Theorem 4.1 as a consequence of Theorem F.6.

**Theorem 4.1** (Risk bound for H$_2$O). *Let $T \in \mathbb{N}$ be given, let $\mathcal{D}_{\mathsf{off}}$ consist of $H \cdot T$ samples from data distribution $\nu$, and suppose that $\nu$ satisfies $C_\star$-single-policy concentrability. Let $\mathbf{Alg}_{\mathsf{off}}$ be CC-bounded at scale $\gamma \in (0,1)$ under Assumption($\cdot$), with parameters $\mathfrak{a}_\gamma$ and $\mathfrak{b}_\gamma$. Suppose that $\forall t \in [T]$ and $\pi^{(1)}, \ldots, \pi^{(t)} \in \Pi$, Assumption($\mu^{(t)}, M^\star$) holds for $\mu^{(t)} := \{1/2(\nu_h + 1/t \sum_{i=1}^{t} d_h^{\pi^{(i)}})\}_{h=1}^{H}$.*

*Then, with probability at least $1 - \delta T$, the risk of* $H_2O$ *(Algorithm 2) with $T$, $\mathbf{Alg}_{\text{off}}$, and $\mathcal{D}_{\text{off}}$ is bounded as[14]*

$$\mathbf{Risk} \leq \widetilde{O}\left(H\left(\frac{\mathfrak{a}_\gamma(C_\star + C_{\text{cov}})}{T} + \mathfrak{b}_\gamma\right)\right). \tag{8}$$

**Proof of Theorem 4.1.** Under the assumptions in the theorem statement, Theorem F.6 implies that

$$\mathbf{Risk} \leq \frac{2\mathfrak{a}_\gamma}{T}\sum_{h=1}^{H}\sum_{t=1}^{T}\frac{1}{t}\mathsf{CC}_h(\pi^\star, \nu, \gamma t) + H\left(4\frac{\mathfrak{a}_\gamma C_{\text{cov}}}{T}\left(\gamma + 10\log(T) + 1\right) + \mathfrak{b}_\gamma\right).$$

Since $\max_h \max_{x,a}\left|\frac{d_h^{\pi^\star}(x,a)}{\nu_h(x,a)}\right| \leq C_\star$, we have $\mathsf{CC}_h(\pi^\star, \nu, \gamma t) \leq C_\star$, so we can simplify the first term above to

$$\frac{2\mathfrak{a}_\gamma}{T}\sum_{h=1}^{H}\sum_{t=1}^{T}\frac{1}{t}\mathsf{CC}_h(\pi^\star, \nu, \gamma t) \leq \frac{2\mathfrak{a}_\gamma C_\star}{T}H\sum_{t=1}^{T}\frac{1}{t} \leq \frac{6\mathfrak{a}_\gamma C_\star}{T}H\log(T),$$

using the bound on the harmonic number $\sum_{t=1}^{T}1/t \leq 3\log(T)$. Combining with the remainder of the risk bound in Theorem F.6 gives us

$$\mathbf{Risk} \leq \frac{H\mathfrak{a}_\gamma}{T}(6C_\star\log(T) + 4C_{\text{cov}}(\gamma + 10\log(T) + 1)) + H\mathfrak{b}_\gamma = \widetilde{O}\left(H\left(\frac{\mathfrak{a}_\gamma(C_\star + C_{\text{cov}})}{T} + \mathfrak{b}_\gamma\right)\right),$$

where we have used the fact that $\gamma \in [0, 1]$. $\qquad\square$

**Corollary F.1.** *Let $T \in \mathbb{N}$ and $\mathcal{D}_{\text{off}}$ consist of $H \cdot T$ samples from data distribution $\nu$. Suppose that for all $t \in [T]$ and $\pi^{(1)}, \ldots, \pi^{(t)} \in \Pi$, $\mathsf{Assumption}(\mu^{(t)}, M^\star)$ holds for $\mu^{(t)} := \{1/2(\nu_h + 1/t\sum_{i=1}^{t}d_h^{\pi^{(i)}})\}_{h=1}^{H}$. Let $\mathbf{Alg}_{\text{off}}$ be CC-bounded at scale $\gamma \in [0, 1]$ under $\mathsf{Assumption}(\cdot)$ and with parameters $\mathfrak{a}_\gamma = \frac{\mathfrak{a}}{\gamma}$ and $\mathfrak{b}_\gamma = \mathfrak{b}\gamma$. Consider the $H_2O$ algorithm with inputs $T$, $\mathbf{Alg}_{\text{off}}$, and $\mathcal{D}_{\text{off}}$. Then,*

- *If $\mathbf{Alg}_{\text{off}}$ is CC-bounded at scale $\gamma = \widetilde{\Theta}\left(\sqrt{\mathfrak{a}C_{\text{cov}}/\mathfrak{b}T}\right)$ and $T$ is such that $\gamma \in [0, 1]$, we have*

$$\mathbf{Risk} \leq 2\sqrt{\frac{\mathfrak{a}\mathfrak{b}}{TC_{\text{cov}}}}\sum_{h=1}^{H}\sum_{t=1}^{T}\frac{1}{t}\mathsf{CC}_h(\pi^\star, \nu, \gamma^{(t)}) + \widetilde{O}\left(H\sqrt{\frac{C_{\text{cov}}\mathfrak{a}\mathfrak{b}}{T}}\right).$$

  *with probability greater than $1 - \delta T$.*

- *If $\nu$ satisfies $C_\star$-single-policy concentrability and $\mathbf{Alg}_{\text{off}}$ is CC-bounded at scale $\gamma = \widetilde{\Theta}\left(\sqrt{\mathfrak{a}(C_\star + C_{\text{cov}})/\mathfrak{b}T}\right)$ and $T$ is such that $\gamma \in [0, 1]$, then*

$$\mathbf{Risk} \leq \widetilde{O}\left(H\sqrt{(C_\star + C_{\text{cov}})\mathfrak{a}\mathfrak{b}/T}\right),$$

  *with probability greater than $1 - \delta T$.*

**Proof of Corollary F.1.** We start with the first case. We recall the definition of $\mathsf{err}_{\text{off}}$ appearing in Theorem F.6:

$$\mathsf{err}_{\text{off}} := H\left(\frac{4\mathfrak{a}_\gamma C_{\text{cov}}}{T}\left(\gamma + 8\log(T) + 1\right) + \mathfrak{b}_\gamma\right) = \widetilde{O}\left(H\left(\frac{C_{\text{cov}}\mathfrak{a}_\gamma}{T} + \mathfrak{b}_\gamma\right)\right).$$

and the risk bound from Theorem F.6.

$$\mathbf{Risk} \leq \frac{2\mathfrak{a}_\gamma}{T}\sum_{h=1}^{H}\sum_{t=1}^{T}\frac{1}{t}\mathsf{CC}_h(\pi^\star, \nu, \gamma^{(t)}) + \mathsf{err}_{\text{off}}. \tag{45}$$

---

[14]We define risk for the hybrid setting as in Eq. (1). Our result is stated as a bound on the risk to the optimal policy $\pi^\star$, but extends to give a bound on the risk of any comparator $\pi$ with $C_\star$ replaced by coverage for $\pi$.

Plugging in $\mathfrak{a}_\gamma = \frac{\mathfrak{a}}{\gamma}$ and $\mathfrak{b}_\gamma = \mathfrak{b}\gamma$ into $\mathsf{err}_{\mathsf{off}}$ gives

$$\mathsf{err}_{\mathsf{off}} = H\left(4\frac{\mathfrak{a}}{\gamma T}C_{\mathsf{cov}}\left(\gamma + 8\log(T) + 1\right) + \mathfrak{b}\gamma\right).$$

The above is optimized by picking $\gamma = 2\sqrt{\frac{\mathfrak{a}}{\mathfrak{b}}\frac{C_{\mathsf{cov}}}{T}(8\log(T)+1)}$. Plugging this in gives us

$$\mathsf{err}_{\mathsf{off}} = H\left(4\sqrt{\frac{\mathfrak{a}\mathfrak{b}C_{\mathsf{cov}}(8\log(T)+1)}{T}} + 4\frac{\mathfrak{a}C_{\mathsf{cov}}}{T}\right) = \widetilde{O}\left(H\sqrt{\frac{\mathfrak{a}\mathfrak{b}C_{\mathsf{cov}}}{T}}\right),$$

as desired. For the second result, recall from Theorem 4.1 that the risk bound when $\nu$ satisfied $C_\star$-policy-concentrability is

$$\mathbf{Risk} \le \frac{H\mathfrak{a}_\gamma}{T}(6C_\star\log(T) + 4C_{\mathsf{cov}}(\gamma + 8\log(T) + 1)) + H\mathfrak{b}_\gamma.$$

Plugging in $\mathfrak{a}_\gamma = \frac{\mathfrak{a}}{\gamma}$ and $\mathfrak{b}_\gamma = \mathfrak{b}\gamma$ gives us

$$\mathbf{Risk} \le \frac{H\mathfrak{a}}{T\gamma}(6C_\star\log(T) + 4C_{\mathsf{cov}}(\gamma + 8\log(T) + 1)) + H\mathfrak{b}\gamma.$$

This expression is optimized by $\gamma = \sqrt{\frac{\mathfrak{a}(6C_\star\log(T) + 4C_{\mathsf{cov}}(\gamma + 8\log(T)+1))}{T\mathfrak{b}}}$, which when substituted gives us the risk bound

$$\mathbf{Risk} \le 2H\sqrt{\frac{\mathfrak{a}\mathfrak{b}\left(6C_\star\log(T) + 4C_{\mathsf{cov}}(\gamma + 8\log(T) + 1)\right)}{T}} = \widetilde{O}\left(H\left(\sqrt{\frac{(C_\star + C_{\mathsf{cov}})\mathfrak{a}\mathfrak{b}}{T}}\right)\right),$$

as desired. $\qquad\square$

### F.5 PROOFS FOR H$_2$O EXAMPLES (APPENDIX F.1)

#### F.5.1 SUPPORTING TECHNICAL RESULTS

**Lemma F.2** (Telescoping Performance Difference (Xie and Jiang (2020), Theorem 2); Jin et al. (2021b, Lemma 3.1)))**.** *For any $f \in \mathcal{F}$, we have that*

$$J(\pi^\star) - J(\pi_f) \le \sum_{h=1}^{H} \mathbb{E}_{d_h^{\pi^\star}}[\mathcal{T}_h f_{h+1}(x_h, a_h) - f_h(x_h, a_h)] + \mathbb{E}_{d_h^{\pi_f}}[f_h(x_h, a_h) - \mathcal{T}_h f_{h+1}(x_h, a_h)].$$

This bound follows from a straightforward adaptation of the proof of Xie and Jiang (2020, Theorem 2) to the finite horizon setting.

**Lemma F.3.** *For all policy $\pi \in \Pi$, value function $f \in \mathcal{F}$, timestep $h \in [H]$, data distribution $\mu = \{\mu_h\}_{h=1}^{H}$ where $\mu_h \in \Delta(\mathcal{X} \times \mathcal{A})$, and $\gamma \in \mathbb{R}_+$, we have*

$$\mathbb{E}_{d_h^\pi}[[\Delta_h f](x_h, a_h)] \le \mathbb{E}_{\mu_h}\left[[\Delta_h f](x_h, a_h) \cdot \mathsf{clip}_\gamma\left[\frac{d_h^\pi(x_h, a_h)}{\mu_h(x_h, a_h)}\right]\right] + 2\mathbb{P}^\pi\left[\frac{d_h^\pi(x_h, a_h)}{\mu_h(x_h, a_h)} > \gamma\right].$$

*Similarly,*

$$\mathbb{E}_{d_h^\pi}[-[\Delta_h f](x_h, a_h)] \le \mathbb{E}_{\mu_h}\left[(-[\Delta_h f](x_h, a_h)) \cdot \mathsf{clip}_\gamma\left[\frac{d_h^\pi(x_h, a_h)}{\mu_h(x_h, a_h)}\right]\right] + 2\mathbb{P}^\pi\left[\frac{d_h^\pi(x_h, a_h)}{\mu_h(x_h, a_h)} > \gamma\right],$$

*where recall that $[\Delta_h f](x, a) := f_h(x, a) - [\mathcal{T}_h f_{h+1}](x, a)$.*

**Proof of Lemma F.3.** In the following, we prove the first inequality. The second inequality follows similarly. Using that $|[\Delta_h f](x, a)| \le 1$ for any $x, a \in \mathcal{X} \times \mathcal{A}$, we have

$$\mathbb{E}_{d_h^\pi}[[\Delta_h f](x_h, a_h)] \le \mathbb{E}_{d_h^\pi}[[\Delta_h f](x_h, a_h) \cdot \mathbb{I}\{\mu_h(x_h, a_h) \ne 0\}] + \mathbb{E}_{d_h^\pi}[\mathbb{I}\{\mu_h(x_h, a_h) = 0\}].$$

For the second term, for any $\gamma > 0$,

$$\mathbb{E}_{d_h^\pi}[\mathbb{I}\{\mu_h(x_h, a_h) = 0\}] \le \mathbb{E}_{d_h^\pi}\left[\mathbb{I}\left\{\frac{d_h^\pi(x_h, a_h)}{\mu_h(x_h, a_h)} > \gamma\right\}\right] = \mathbb{P}^\pi\left[\frac{d_h^\pi(x_h, a_h)}{\mu_h(x_h, a_h)} > \gamma\right].$$

For the first term, using that $u \leq \min\{u, v\} + u\mathbb{I}\{u \geq v\}$ for all $u, v \geq 0$, and that $|[\Delta_h f](x, a)| \leq 1$ for any $x, a \in \mathcal{X} \times \mathcal{A}$, we get that

$$
\mathbb{E}_{d_h^\pi}\left[[\Delta_h f](x_h, a_h) \cdot \mathbb{I}\{\mu_h(x_h, a_h) \neq 0\}\right]
$$

$$
= \mathbb{E}_{\mu_h}\left[[\Delta_h f](x_h, a_h) \cdot \frac{d_h^\pi(x_h, a_h)}{\mu_h(x_h, a_h)}\mathbb{I}\{\mu(x_h, a_h) \neq 0\}\right]
$$

$$
\leq \mathbb{E}_{\mu_h}\left[[\Delta_h f](x_h, a_h) \cdot \mathsf{clip}_\gamma\left[\frac{d_h^\pi(x_h, a_h)}{\mu_h(x_h, a_h)}\right]\mathbb{I}\{\mu_h(x_h, a_h) \neq 0\}\right]
$$

$$
+ \mathbb{E}_{\mu_h}\left[\frac{d_h^\pi(x_h, a_h)}{\mu_h(x_h, a_h)}\mathbb{I}\left\{\frac{d_h^\pi(x_h, a_h)}{\mu_h(x_h, a_h)} > \gamma\right\}\mathbb{I}\{\mu_h(x_h, a_h) \neq 0\}\right]
$$

$$
= \mathbb{E}_{\mu_h}\left[[\Delta_h f](x_h, a_h) \cdot \mathsf{clip}_\gamma\left[\frac{d_h^\pi(x_h, a_h)}{\mu_h(x_h, a_h)}\right]\mathbb{I}\{\mu_h(x_h, a_h) \neq 0\}\right] + \mathbb{E}_{d_h^\pi}\left[\mathbb{I}\left\{\frac{d_h^\pi(x_h, a_h)}{\mu_h(x_h, a_h)} > \gamma\right\}\right].
$$

Furthermore, also note that

$$
\mathbb{E}_{\mu_h}\left[[\Delta_h f](x_h, a_h) \cdot \mathsf{clip}_\gamma\left[\frac{d_h^\pi(x_h, a_h)}{\mu_h(x_h, a_h)}\right]\mathbb{I}\{\mu_h(x_h, a_h) = 0\}\right]
$$

$$
= \sum_{(x_h, a_h) \text{ s.t. } \mu_h(x_h, a_h)=0} \mu_h(x_h, a_h) \cdot [\Delta_h f](x_h, a_h) \cdot \mathsf{clip}_\gamma\left[\frac{d_h^\pi(x_h, a_h)}{\mu_h(x_h, a_h)}\right] = 0.
$$

The final bound follows by combining the above three terms. $\qquad \square$

**Lemma F.4.** *For any policy $\pi$, data distribution $\mu = \{\mu_h\}_{h=1}^H$ where $\mu_h \in \Delta(\mathcal{X} \times \mathcal{A})$, scale $\gamma \in \mathbb{R}_+$ and horizon $h \in [H]$, we have*

$$
\mathbb{P}^\pi\left[\frac{d_h^\pi(x_h, a_h)}{\mu_h(x_h, a_h)} > \gamma\right] \leq \frac{2}{\gamma}\left\|\mathsf{clip}_\gamma\left[\frac{d_h^\pi}{\mu_h}\right]\right\|_{1, d_h^\pi}.
$$

**Proof of Lemma F.4.** Note that

$$
\mathbb{P}^\pi\left[\frac{d_h^\pi(x_h, a_h)}{\mu_h(x_h, a_h)} > \gamma\right] = \mathbb{E}_{d_h^\pi}\left[\mathbb{I}\left\{\frac{d_h^\pi(x_h, a_h)}{\mu_h(x_h, a_h)} > \gamma\right\}\right]
$$

$$
\leq \mathbb{E}_{d_h^\pi}\left[\mathbb{I}\left\{\frac{d_h^\pi(x_h, a_h)}{\mu_h(x, a) + \gamma^{-1}d_h^\pi(x_h, a_h)} > \frac{\gamma}{2}\right\}\right]
$$

$$
\leq \frac{2}{\gamma}\mathbb{E}_{d_h^\pi}\left[\frac{d_h^\pi(x_h, a_h)}{\mu_h(x_h, a_h) + \gamma^{-1}d_h^\pi(x_h, a_h)}\right]
$$

$$
\leq \frac{2}{\gamma}\mathbb{E}_{d_h^\pi}\left[\mathsf{clip}_\gamma\left[\frac{d_h^\pi(x_h, a_h)}{\mu_h(x_h, a_h)}\right]\right] = \frac{2}{\gamma}\left\|\mathsf{clip}_\gamma\left[\frac{d_h^\pi}{\mu_h}\right]\right\|_{1, d_h^\pi},
$$

where the first inequality follows from

$$
\gamma^{-1}d_h^\pi(x, a) > \mu_h(x, a) \implies \gamma^{-1}d_h^\pi(x, a) > \frac{1}{2}(\mu_h(x, a) + \gamma^{-1}d_h^\pi(x, a)),
$$

and the second inequality follows by Markov's inequality. $\qquad \square$

**Lemma F.5.** *For any policy $\pi$, data distribution $\mu = \{\mu_h\}_{h=1}^H$ where $\mu_h \in \Delta(\mathcal{X} \times \mathcal{A})$, scale $\gamma \in \mathbb{R}_+$, and horizon $h \in [H]$, we have*

$$
\left\|\mathsf{clip}_\gamma\left[\frac{d_h^\pi}{\mu_h}\right]\right\|_{2, \mu_h}^2 \leq 2\left\|\mathsf{clip}_\gamma\left[\frac{d_h^\pi}{\mu_h}\right]\right\|_{1, d_h^\pi}.
$$

**Proof of Lemma F.5.** Beginning with the left-hand side, we have,

$$
\mathbb{E}_{\mu_h}\left[\min\left\{\frac{d_h^\pi(x_h, a_h)}{\mu_h(x_h, a_h)}, \gamma\right\}^2\right] \overset{(i)}{\leq} 2\,\mathbb{E}_{\mu_h}\left[\left(\frac{d_h^\pi(x_h, a_h)}{\mu_h(x_h, a_h)}\right)^2\mathbb{I}\left\{\frac{d_h^\pi(x_h, a_h)}{\mu_h(x_h, a_h)} \leq \gamma\right\}\right]
$$

$$+ 2\,\mathbb{E}_{\mu_h}\left[\gamma \cdot \frac{d_h^\pi(x_h, a_h)}{\mu_h(x_h, a_h)} \cdot \mathbb{I}\left\{\frac{d_h^\pi(x_h, a_h)}{\mu_h(x_h, a_h)} > \gamma\right\}\right]$$

$$\overset{(ii)}{\leq} 2\,\mathbb{E}_{d_h^\pi}\left[\frac{d_h^\pi(x_h, a_h)}{\mu_h(x_h, a_h)}\mathbb{I}\left\{\frac{d_h^\pi(x_h, a_h)}{\mu_h(x_h, a_h)} \leq \gamma\right\}\right] + 2\,\mathbb{E}_{d_h^\pi}\left[\gamma \cdot \mathbb{I}\left\{\frac{d_h^\pi(x_h, a_h)}{\mu_h(x_h, a_h)} > \gamma\right\}\right]$$

$$\overset{(iii)}{\leq} 2\,\mathbb{E}_{d_h^\pi}\left[\min\left\{\frac{d_h^\pi(x_h, a_h)}{\mu_h(x_h, a_h)}, \gamma\right\}\right].$$

In (i), we have used that for all $u, v \in \mathbb{R}^+$, $\min\{u, v\} \leq u\mathbb{I}\{u \leq v\} + \sqrt{uv}\mathbb{I}\{v < u\}$ and that $(u+v)^2 \leq 2(u^2 + v^2)$, thus that $\min\{u, v\}^2 \leq 2(u^2\mathbb{I}\{u \leq v\} + uv\mathbb{I}\{v < u\})$. In (ii), we have done a change of measure from $\mu_h$ to $d_h^\pi$. In (iii), we have used that $\min\{u, v\} = u\mathbb{I}\{u \leq v\} + v\mathbb{I}\{v < u\}$.

$\square$

### F.5.2 PROOFS FOR MABO.CR (PROOF OF THEOREM F.1)

Suppose the dataset $\mathcal{D} = \{\mathcal{D}_h\}_{h \leq H}$ is sampled from the offline distribution $\mu^{(1:n)}$. In this section, we analyze our regularized and clipped variant of MABO (Eq. (9)). We analyze MABO.CR under the following density ratio assumption. Recall for any sequence of data distributions $\mu^{(1)}, \ldots \mu^{(n)}$, we denote the mixture distribution by $\mu^{(1:n)} = \{\mu_h^{(1:n)}\}_{h=1}^H$, defined by $\mu_h^{(1:n)} := \frac{1}{n}\sum_{i=1}^n \mu_h^{(i)}$.

**Assumption F.4.** *For a given sequence of data distributions $\mu^{(1)}, \ldots \mu^{(n)}$, we have that for all $\pi \in \Pi$, and for all $h \in [H]$,*

$$\frac{d_h^\pi}{\mu_h^{(1:n)}} \in \mathcal{W}.$$

We first note the following bound for the hypothesis $\widehat{f}$ returned by MABO.CR.

**Lemma F.6.** *Let $\mathcal{D} = \{\mathcal{D}_h\}_{h=1}^H$ be a dataset consisting of $H \cdot n$ samples from $\mu^{(1)}, \ldots, \mu^{(n)}$. Suppose that $\mathcal{F}$ satisfies Assumption 2.1 and that $\mathcal{W}$ satisfies Assumption F.4 with respect to $\mu^{(1:n)}$. Let $\widehat{f}$ be the function returned by MABO.CR, given in Eq. (9), when executed on $\{\mathcal{D}_h\}_{h \leq H}$ with parameters $\gamma, \mathcal{F}$, and the augmented weight function class $\overline{\mathcal{W}}$ defined in Eq. (34). Then, with probability at least $1 - \delta$, we have*

$$\sum_{h=1}^H \left|\mathbb{E}_{\mu_h^{(1:n)}}\left[[\Delta_h \widehat{f}](x_h, a_h) \cdot \check{w}_h(x_h, a_h)\right]\right| \leq \sum_{h=1}^H \frac{20}{\gamma^{(n)}}\,\mathbb{E}_{\mu_h^{(1:n)}}\left[(\check{w}_h(x_h, a_h))^2\right] + \frac{7}{18}H^2\beta^{(n)}, \quad (46)$$

*for all $w \in \mathcal{W}$, where $\beta^{(n)} := \frac{36\gamma^{(n)}}{n-1}\log(24|\mathcal{F}||\mathcal{W}|H^2/\delta)$.*

**Proof of Lemma F.6.** Repeating the argument of Lemma E.2 (a), we can establish that $Q^\star$ satisfies the following bound for all $h \in [H], w \in \overline{\mathcal{W}}$:

$$\widehat{\mathbb{E}}_{\mathcal{D}_h}\left[([\widehat{\Delta}_h Q^\star](x_h, a_h, r_h, x'_{h+1})) \cdot \check{w}_h(x_h, a_h)\right] \leq \widehat{\mathbb{E}}_{\mathcal{D}_h}\left[\alpha^{(n)} \cdot (\check{w}_h(x_h, a_h))^2\right] + \beta^{(n)}, \quad (47)$$

with probability at least $1 - \delta$, where $\alpha^{(n)} = 8/\gamma^{(n)}$ and $\beta^{(n)} = 36\gamma^{(n)}/n{-}1\log(6|\mathcal{F}||\overline{\mathcal{W}}|H/\delta) \leq 36\gamma^{(n)}/n{-}1\log(24|\mathcal{F}||\mathcal{W}|H^2/\delta)$. Going forward, we condition on the event that this holds. Now, since the right-hand side of Eq. (47) is independent of the sign of $\check{w}_h$, we can apply this to $-\check{w} \in \overline{\mathcal{W}}$ to conclude that

$$\left|\widehat{\mathbb{E}}_{\mathcal{D}_h}\left[([\widehat{\Delta}_h Q^\star](x_h, a_h, r_h, x'_{h+1})) \cdot \check{w}_h(x_h, a_h)\right]\right| \leq \widehat{\mathbb{E}}_{\mathcal{D}_h}\left[\alpha^{(n)} \cdot (\check{w}_h(x_h, a_h))^2\right] + \beta^{(n)}.$$

Summing over $h \in [H]$ and taking the max over $w \in \overline{\mathcal{W}}$, we can conclude that:

$$\max_{w \in \overline{\mathcal{W}}}\sum_{h=1}^H\left|\widehat{\mathbb{E}}_{\mathcal{D}_h}\left[([\widehat{\Delta}_h Q^\star](x_h, a_h, r_h, x'_{h+1})) \cdot \check{w}_h(x_h, a_h)\right]\right| - \widehat{\mathbb{E}}_{\mathcal{D}_h}\left[\alpha^{(n)} \cdot (\check{w}_h(x_h, a_h))^2\right] \leq H\beta^{(n)}.$$

By Assumption 2.1 and the definition of the hypothesis $\widehat{f}$ returned by MABO.CR, we have:

$$\max_{w \in \overline{\mathcal{W}}}\sum_{h=1}^H\left|\widehat{\mathbb{E}}_{\mathcal{D}_h}\left[([\widehat{\Delta}_h \widehat{f}](x_h, a_h, r_h, x'_{h+1})) \cdot \check{w}_h(x_h, a_h)\right]\right| - \widehat{\mathbb{E}}_{\mathcal{D}_h}\left[\alpha^{(n)} \cdot (\check{w}_h(x_h, a_h))^2\right] \leq H\beta^{(n)},$$

and in particular the bound that for all $w \in \overline{\mathcal{W}}$

$$\sum_{h=1}^{H} \widehat{\mathbb{E}}_{\mathcal{D}_h} \left[ \left( [\widehat{\Delta}_h \widehat{f}](x_h, a_h, r_h, x'_{h+1}) \cdot \check{w}_h(x_h, a_h) \right) \right] \leq \sum_{h=1}^{H} \widehat{\mathbb{E}}_{\mathcal{D}_h} \left[ \alpha^{(n)} \cdot \left( \check{w}_h(x_h, a_h) \right)^2 \right] + H\beta^{(n)}.$$

Repeating the argument for [Lemma E.2](#) (b), we can conclude that for all $w \in \overline{\mathcal{W}}$

$$\sum_{h=1}^{H} \mathbb{E}_{\mu_h^{(1:n)}} \left[ [\Delta_h \widehat{f}](x_h, a_h) \cdot \check{w}_h(x_h, a_h) \right] \leq \sum_{h=1}^{H} \frac{20}{\gamma^{(n)}} \mathbb{E}_{\mu_h^{(1:n)}} \left[ (\check{w}_h(x_h, a_h))^2 \right] + \frac{7H\beta^{(n)}}{18}.$$

Applying this to $w^{(h)} \in \overline{\mathcal{W}}$ and to $-w^{(h)} \in \overline{\mathcal{W}}$, and again noting that the right-hand side is independent of the sign of $\check{w}$, we can conclude that for each $h \in [H], w \in \overline{\mathcal{W}}$:

$$\left| \mathbb{E}_{\mu_h^{(1:n)}} \left[ [\Delta_h \widehat{f}](x_h, a_h) \cdot \check{w}_h(x_h, a_h) \right] \right| \leq \frac{20}{\gamma^{(n)}} \mathbb{E}_{\mu_h^{(1:n)}} \left[ (\check{w}_h(x_h, a_h))^2 \right] + \frac{7H\beta^{(n)}}{18},$$

Summing over $h \in [H]$ gives the desired bound. $\qquad\square$

**Theorem F.1** (MABO.CR is CC-bounded)**.** *Let* $\mathcal{D} = \{\mathcal{D}_h\}_{h=1}^{H}$ *consist of* $H \cdot n$ *samples from* $\mu^{(1)}, \ldots, \mu^{(n)}$. *For any* $\gamma \in \mathbb{R}_+$, *the* MABO.CR *algorithm ([Eq. (9)](#)) with parameters* $\mathcal{F}$, *augmented class* $\overline{\mathcal{W}}$ *defined in [Eq. (34)](#) in [Appendix F.1.1](#), and* $\gamma$ *is* CC-*bounded at scale* $\gamma$ *under the* Assumption *that* $Q^\star \in \mathcal{F}$ *and that for all* $\pi \in \Pi$ *and* $h \in [H]$, $d_h^\pi / \mu_h^{(1:n)} \in \mathcal{W}$.

**Proof of [Theorem F.1](#).** Let $\widehat{\pi} = \pi_{\widehat{f}}$. Using the performance difference lemma ([Lemma F.2](#)), we note that

$$J(\pi^\star) - J(\widehat{\pi}) \leq \sum_{h=1}^{H} \mathbb{E}_{d_h^{\pi^\star}} [-[\Delta_h \widehat{f}](x_h, a_h)] + \sum_{h=1}^{H} \mathbb{E}_{d_h^{\widehat{\pi}}} [[\Delta_h \widehat{f}](x_h, a_h)]. \tag{48}$$

However, note that due to [Lemma F.3](#) and [Lemma F.4](#), we have that

$$\mathbb{E}_{d_h^{\widehat{\pi}}} [([\Delta_h \widehat{f}](x_h, a_h))] \leq \underbrace{\mathbb{E}_{\mu_h^{(1:n)}} \left[ ([\Delta_h \widehat{f}](x_h, a_h)) \mathsf{clip}_{\gamma n} \left[ \frac{d_h^{\widehat{\pi}}(x_h, a_h)}{\mu_h^{(1:n)}(x_h, a_h)} \right] \right]}_{\text{(I)}} + \frac{4}{\gamma n} \left\| \mathsf{clip}_{\gamma n} \left[ \frac{d_h^{\widehat{\pi}}}{\mu_h^{(1:n)}} \right] \right\|_{1, d_h^{\widehat{\pi}}}, \tag{49}$$

and,

$$\mathbb{E}_{d_h^{\pi^\star}} [(-[\Delta_h \widehat{f}](x_h, a_h))] \leq \underbrace{\mathbb{E}_{\mu_h^{(1:n)}} \left[ (-[\Delta_h \widehat{f}](x_h, a_h)) \mathsf{clip}_{\gamma n} \left[ \frac{d_h^{\pi^\star}(x_h, a_h)}{\mu_h^{(1:n)}(x_h, a_h)} \right] \right]}_{\text{(II)}} + \frac{4}{\gamma n} \left\| \mathsf{clip}_{\gamma n} \left[ \frac{d_h^{\pi^\star}}{\mu_h^{(1:n)}} \right] \right\|_{1, d_h^{\pi^\star}}. \tag{50}$$

We bound the terms (I) and (II) separately below. Before we delve into these bounds, note that using [Lemma F.6](#), we have with probability at least $1 - \delta$,

$$\max_{w \in \mathcal{W}} \sum_{h=1}^{H} \left( \left| \mathbb{E}_{\mu_h^{(1:n)}} \left[ \check{w}_h([\Delta_h \widehat{f}](x_h, a_h)) \right] \right| - \frac{20}{\gamma^{(n)}} \mathbb{E}_{\mu_h^{(1:n)}} \left[ (\check{w}_h(x_h, a_h))^2 \right] \right) \leq \frac{7}{18} H^2 \beta^{(n)}, \tag{51}$$

where $\beta^{(n)} := \frac{36\gamma n}{n-1} \log(24|\mathcal{F}||\mathcal{W}|H^2/\delta)$. Moving forward, we condition on the event under which [Eq. (51)](#) holds.

**Bound on Term (I).** Define $w$ via $w_h := \frac{d_h^{\widehat{\pi}}}{\mu_h^{(1:n)}}$ and $\check{w}_h := \mathsf{clip}_{\gamma n} \left[ \frac{d_h^{\widehat{\pi}}}{\mu_h^{(1:n)}} \right]$, and note that due to [Assumption F.4](#), we have that $w \in \mathcal{W}$. Thus, using [(51)](#), we get that

$$\sum_{h=1}^{H} \mathbb{E}_{\mu_h^{(1:n)}} [(-[\Delta_h \widehat{f}](x_h, a_h)) \check{w}_h(x_h, a_h)] \leq \sum_{h=1}^{H} \left| \mathbb{E}_{\mu_h^{(1:n)}} [(-[\Delta_h \widehat{f}](x_h, a_h)) \check{w}_h(x_h, a_h)] \right|$$

$$\leq \sum_{h=1}^{H} \frac{20}{\gamma^{(n)}} \mathbb{E}_{\mu_h^{(1:n)}}\left[(\breve{w}_h(x_h, a_h))^2\right] + \frac{7}{18} H^2 \beta^{(n)}$$

$$= \sum_{h=1}^{H} \frac{20}{\gamma^{(n)}} \left\|\mathsf{clip}_{\gamma n}[w_h]\right\|_{2,\mu_h^{(1:n)}}^2 + \frac{7}{18} H^2 \beta^{(n)}$$

$$\leq \sum_{h=1}^{H} \frac{40}{\gamma^{(n)}} \left\|\mathsf{clip}_{\gamma n}[w_h]\right\|_{1,d_h^{\widehat{\pi}}} + \frac{7}{18} H^2 \beta^{(n)},$$

where the last step follows by Lemma F.5 and the definition of $w_h$.

**Bound on Term (II).** Define $w_h^\star \coloneqq \frac{d_h^{\pi^\star}}{\mu_h^{(1:n)}}$ and $\breve{w}_h^\star \coloneqq \mathsf{clip}_{\gamma n}\left[\frac{d_h^{\pi^\star}}{\mu_h^{(1:n)}}\right]$, and again note that due to Assumption F.4, we have that $w_h \in \mathcal{W}_h$. Thus, using (51), and repeating the same arguments as above, we get that

$$\text{(II)} \leq \sum_{h=1}^{H} \frac{40}{\gamma^{(n)}} \left\|\mathsf{clip}_{\gamma n}[w_h^\star]\right\|_{1,d_h^{\pi^\star}} + \frac{7}{18} H^2 \beta^{(n)},$$

Plugging the two bounds above in (53) and (54), and then further in (52), and using the definitions for $w_h$ and $w_h^\star$, we get that with probability at least $1 - \delta$,

$$J(\pi^\star) - J(\widehat{\pi}) \leq \sum_{h=1}^{H} \frac{40}{\gamma n} \left\|\mathsf{clip}_{\gamma n}\left[\frac{d_h^{\pi^\star}}{\mu_h^{(1:n)}}\right]\right\|_{1,d_h^{\pi^\star}} + \sum_{h=1}^{H} \frac{5}{\gamma n} \left\|\mathsf{clip}_{\gamma n}\left[\frac{d_h^{\widehat{\pi}}}{\mu_h^{(1:n)}}\right]\right\|_{1,d_h^{\widehat{\pi}}} + \frac{14}{18} H^2 \beta^{(n)}$$

$$= \sum_{h=1}^{H} \frac{40}{\gamma^{(n)}} \left(\left\|\mathsf{clip}_{\gamma n}\left[\frac{d_h^{\pi^\star}}{\mu_h^{(1:n)}}\right]\right\|_{1,d_h^{\pi^\star}} + \left\|\mathsf{clip}_{\gamma n}\left[\frac{d_h^{\widehat{\pi}}}{\mu_h^{(1:n)}}\right]\right\|_{1,d_h^{\widehat{\pi}}}\right) + \frac{14}{18} H^2 \beta^{(n)}$$

$$= \sum_{h=1}^{H} \frac{40}{\gamma^{(n)}} \left(\mathsf{CC}_h(\pi^\star, \mu^{(1:n)}, \gamma n) + \mathsf{CC}_h(\widehat{\pi}, \mu^{(1:n)}, \gamma n)\right) + 28 H^2 \gamma \frac{n}{n-1} \log(24|\mathcal{F}||\mathcal{W}|H^2/\delta)$$

$$\leq \sum_{h=1}^{H} \frac{40}{\gamma n} \left(\mathsf{CC}_h(\pi^\star, \mu^{(1:n)}, \gamma n) + \mathsf{CC}_h(\widehat{\pi}, \mu^{(1:n)}, \gamma n)\right) + 56 H^2 \gamma \log(24|\mathcal{F}||\mathcal{W}|H^2/\delta),$$

which establishes that the algorithm is $\mathsf{CC}$-bounded for scale $\gamma$, with scaling functions $\mathfrak{a}_\gamma = \frac{40}{\gamma}$ and $\mathfrak{b}_\gamma = 56 H^2 \gamma \log(24|\mathcal{F}||\mathcal{W}|H^2/\delta)$.

$\square$

**Theorem 4.2** (Risk bound for HYGLOW). *Let $\varepsilon > 0$ be given, let $\mathcal{D}_{\mathrm{off}}$ consist of $H \cdot T$ samples from data distribution $\nu$, and suppose that $\nu$ satisfies $C_\star$-single-policy concentrability. Suppose that $Q^\star \in \mathcal{F}$ and that for all $t \in [T]$, $\pi \in \Pi$, and $h \in [H]$, we have $d_h^\pi/\mu_h^{(t)} \in \mathcal{W}$, where $\mu_h^{(t)} \coloneqq 1/2(\nu_h + 1/t \sum_{i=1}^{t} d_h^{\pi^{(i)}})$. Then, HYGLOW (Algorithm 3) with inputs $T = \widetilde{\Theta}\left((H^4(C_{\mathsf{cov}}+C_\star)/\varepsilon^2) \cdot \log(|\mathcal{F}||\mathcal{W}|/\delta)\right)$, $\mathcal{F}$, augmented $\overline{\mathcal{W}}$ defined in Eq. (34), $\gamma = \widetilde{\Theta}\left(\sqrt{(C_\star+C_{\mathsf{cov}})/TH^2 \log(|\mathcal{F}||\mathcal{W}|/\delta)}\right)$, and $\mathcal{D}_{\mathrm{off}}$ returns an $\varepsilon$-suboptimal policy with probability at least $1 - \delta T$ after collecting $N = \widetilde{O}\left(\frac{H^2(C_{\mathsf{cov}}+C_\star)}{\varepsilon^2} \log(|\mathcal{F}||\mathcal{W}|/\delta)\right)$ trajectories.*

**Proof of Theorem 4.2.** This follows by combining Theorem F.1 with Corollary F.1. $\square$

### F.5.3 PROOFS FOR FITTED Q-ITERATION (FQI)

In this section we prove Theorem F.3.

We quote the following generalization bound for least squares regression in the adaptive setting.

**Lemma F.7** (Least squares generalization bound; Song et al. (2023, Lemma 3)). *Let $R > 0, \delta \in (0, 1)$, and $\mathcal{H} : \mathcal{X} \mapsto [-R, R]$ a class of real-valued functions. Let $\mathcal{D} = \{(x_1, y_1) \dots (x_T, y_T)\}$ be a dataset of $T$ points where $x_t \sim \rho_t(x_{1:t-1}, y_{1:t-1})$ and $y_t = h^\star(x_t) + \varepsilon_t$ for some realizable $h^\star \in \mathcal{H}$ and $\varepsilon_t$ is conditionally mean-zero, i.e. $\mathbb{E}[y_t \mid x_t] = h^\star(x_t)$. Suppose $\max_t |y_t| \leq R$ and $\max_x |h^\star(x)| \leq R$. Then the least squares solution $\widehat{h} \in \arg\min_{h \in \mathcal{H}} \sum_{t=1}^T (h(x_t) - y_t)^2$ satisfies that with probability at least $1 - \delta$,*

$$\sum_{t=1}^T \mathbb{E}_{x \sim \rho_t} \left[ (\widehat{h}(x) - h^\star(x))^2 \right] \leq 256 R^2 \log(2|\mathcal{H}|/\delta).$$

Using the above theorem, we can show the following concentration result for FQI.

**Lemma F.8** (Concentration bound for FQI). *With probability at least $1 - \delta$, we have that for all $h \in [H]$,*

$$\mathbb{E}_{\mu_h^{(1:n)}} \left[ ([\Delta_h \widehat{f}](x_h, a_h))^2 \right] = \frac{1}{n} \sum_{i=1}^n \mathbb{E}_{(x_h, a_h) \sim \mu_h^{(i)}} \left[ \left( \widehat{f}_h(x_h, a_h) - [\mathcal{T}_h \widehat{f}_{h+1}](x_h, a_h) \right)^2 \right]$$

$$\leq 1024 \frac{\log(2|\mathcal{F}|H/\delta)}{n}.$$

**Proof of Lemma F.8.** Fix $h + 1$. Consider the regression problem induced by the dataset $\mathcal{D}_h = \{(z_h^{(i)}, y_h^{(i)})\}_{i=1}^n$ where $z_h^{(i)} = (x_h^{(i)}, a_h^{(i)}) \sim \mu_h^{(i)}$ and $y_h^{(i)} = r^{(i)} + \max_{a'} \widehat{f}_{h+1}(x_{h+1}^{(i)}, a')$. This problem is realizable via the regression function $\mathbb{E}[y_h^{(i)} \mid z_h^{(i)}] = h^\star(z_h^{(i)}) = \mathcal{T}\widehat{f}_{h+1}(z_h^{(i)}) \in \mathcal{F}$, and satisfies that $|y_h^{(i)}| \leq 2, |h^\star(z_h^{(i)})| \leq 2$. In this regression problem, the least squares solution from Lemma F.7 is precisely the FQI solution, so by Lemma F.7 we have

$$\mathbb{E}_{(x_h, a_h) \sim \mu_h^{(1:n)}} \left( \widehat{f}_h(x_h, a_h) - [\mathcal{T}\widehat{f}_{h+1}](x_h, a_h) \right)^2 = \frac{1}{n} \sum_{i=1}^n \mathbb{E}_{(x_h, a_h) \sim \mu_h^{(i)}} \left( \widehat{f}_h(x_h, a_h) - [\mathcal{T}\widehat{f}_{h+1}](x_h, a_h) \right)^2$$

$$\leq 1024 \frac{\log(2|\mathcal{F}|/\delta)}{n},$$

with high probability. Taking a union bound over $h \in [H]$ gives the desired result. $\qquad\square$

**Theorem F.3.** *The FQI algorithm is $\mathsf{CC}$-bounded under Assumption F.2 with scaling functions $\mathfrak{a}_\gamma = \frac{6}{\gamma}$ and $\mathfrak{b}_\gamma = 1024 \log(2n|\mathcal{F}|)\gamma$, for all $\gamma > 0$ simultaneously. As a consequence, when invoked within $H_2O$, we have $\mathbf{Risk} \leq \widetilde{O}\big(H\sqrt{(C_\star + C_{\mathsf{cov}}) \log(|\mathcal{F}|\delta^{-1})/T}\big)$ with probability at least $1 - \delta T$.*

**Proof of Theorem F.3.** Let $\widehat{\pi} = \pi_{\widehat{f}}$. Using the performance difference lemma (given in Lemma F.2), we note that

$$J(\pi^\star) - J(\widehat{\pi}) \leq \sum_{h=1}^H \mathbb{E}_{d_h^{\pi^\star}} [-[\Delta_h \widehat{f}](x_h, a_h)] + \sum_{h=1}^H \mathbb{E}_{d_h^{\widehat{\pi}}} [[\Delta_h \widehat{f}](x_h, a_h)]. \tag{52}$$

However, note that due to Lemma F.3 and Lemma F.4, we have that

$$\mathbb{E}_{d_h^{\widehat{\pi}}} [[\Delta_h \widehat{f}](x_h, a_h)] \leq \underbrace{\mathbb{E}_{\mu_h^{(1:n)}} \left[ ([\Delta_h \widehat{f}](x_h, a_h)) \mathsf{clip}_{\gamma n} \left[ \frac{d_h^{\widehat{\pi}}(x_h, a_h)}{\mu_h^{(1:n)}(x_h, a_h)} \right] \right]}_{(\mathrm{I})} + \frac{4}{\gamma n} \left\| \mathsf{clip}_{\gamma n} \left[ \frac{d_h^{\widehat{\pi}}}{\mu_h^{(1:n)}} \right] \right\|_{1, d_h^{\widehat{\pi}}}, \tag{53}$$

and,

$$\mathbb{E}_{d_h^{\pi^\star}} [(-[\Delta_h \widehat{f}](x_h, a_h))] \leq \underbrace{\mathbb{E}_{\mu_h^{(1:n)}} \left[ (-[\Delta_h \widehat{f}](x_h, a_h)) \mathsf{clip}_{\gamma n} \left[ \frac{d_h^{\pi^\star}(x_h, a_h)}{\mu_h^{(1:n)}(x_h, a_h)} \right] \right]}_{(\mathrm{II})} + \frac{4}{\gamma n} \left\| \mathsf{clip}_{\gamma n} \left[ \frac{d_h^{\pi^\star}}{\mu_h^{(1:n)}} \right] \right\|_{1, d_h^{\pi^\star}}. \tag{54}$$

**Bound on Term (I).** Note that

$$\mathbb{E}_{\mu_h^{(1:n)}}\left[([\Delta_h\widehat{f}](x_h,a_h))\mathsf{clip}_{\gamma n}\left[\frac{d_h^{\widehat{\pi}}(x_h,a_h)}{\mu_h^{(1:n)}(x_h,a_h)}\right]\right] \leq \sqrt{\mathbb{E}_{\mu_h^{(1:n)}}\left[\left([\Delta_h\widehat{f}](x_h,a_h)\right)^2\right]\mathbb{E}_{\mu_h^{(1:n)}}\left[\left(\mathsf{clip}_{\gamma n}\left[\frac{d_h^{\widehat{\pi}}(x_h,a_h)}{\mu_h^{(1:n)}(x_h,a_h)}\right]\right)^2\right]}$$

$$\leq \sqrt{2048\frac{\log(2|\mathcal{F}|H)}{n}}\left\|\mathsf{clip}_{\gamma n}\left[\frac{d_h^{\widehat{\pi}}}{\mu_h^{(1:n)}}\right]\right\|_{2,\mu_h^{(1:n)}},$$

where the second line follows from Cauchy-Schwarz and the last line follows by Lemma F.8. AM-GM inequality implies that

$$\sqrt{\frac{2048\log(2|\mathcal{F}|H)}{n}}\left\|\mathsf{clip}_n\left[\frac{d_h^{\widehat{\pi}}}{\mu_h^{(1:n)}}\right]\right\|_{2,\mu_h^{(1:n)}} \leq \frac{1}{2}\left(\frac{1}{\gamma n}\left\|\mathsf{clip}_{\gamma n}\left[\frac{d_h^{\widehat{\pi}}}{\mu_h^{(1:n)}}\right]\right\|_{2,\mu_h^{(1:n)}}^2 + 2048\log(2|\mathcal{F}|H)\gamma\right)$$

$$\leq \frac{1}{2}\left(\frac{4}{\gamma n}\left\|\mathsf{clip}_{\gamma n}\left[\frac{d_h^{\widehat{\pi}}}{\mu_h^{(1:n)}}\right]\right\|_{1,d_h^{\widehat{\pi}}} + 2048\log(2|\mathcal{F}|H)\gamma\right),$$

where the last line is due to Lemma F.5.

**Bound on Term (II).** Repeating the same argument above for $\pi^\star$, we get that

$$(\text{II}) \leq \frac{2}{\gamma n}\left\|\mathsf{clip}_{\gamma n}\left[\frac{d_h^{\pi^\star}}{\mu_h^{(1:n)}}\right]\right\|_{1,d_h^{\pi^\star}} + 1024\log(2|\mathcal{F}|H)\gamma.$$

Combining the above bounds implies that

$$J(\pi^\star) - J(\widehat{\pi}) \leq \sum_{h=1}^{H}\frac{2}{\gamma n}\left(\left\|\mathsf{clip}_{\gamma n}\left[\frac{d_h^{\pi^\star}}{\mu_h^{(1:n)}}\right]\right\|_{1,d_h^{\pi^\star}} + \left\|\mathsf{clip}_{\gamma n}\left[\frac{d_h^{\widehat{\pi}}}{\mu_h^{(1:n)}}\right]\right\|_{1,d_h^{\widehat{\pi}}}\right) + 2048\log(2|\mathcal{F}|H)\gamma$$

$$= \sum_{h=1}^{H}\frac{2}{\gamma n}\left(\mathsf{CC}_h(\pi^\star,\mu^{(1:n)},\gamma) + \mathsf{CC}_h(\widehat{\pi},\mu^{(1:n)},\gamma)\right) + 2048\log(2|\mathcal{F}|H)\gamma,$$

which shows that FQI is CC-bounded at scale $\gamma$ under Assumption F.2, with scaling functions $\mathfrak{a}_\gamma = \frac{2}{\gamma}$ and $\mathfrak{b}_\gamma = 2048\log(2|\mathcal{F}|H)\gamma$.

It remains to show the stated risk bound when FQI is applied within $H_2O$. This follows by applying Corollary F.1. □

### F.5.4 PROOFS FOR MODEL-BASED MLE

In this section we prove Theorem F.5. We quote the following generalization bound for maximum likelihood estimation (MLE) in the adaptive setting.

**Lemma F.9** (MLE generalization bound; Agarwal et al. (2020, Theorem 18)). *Consider a sequential conditional probability estimation setting with an instance space $\mathcal{Z}$ and a target space $\mathcal{Y}$ and conditional density $p(y \mid z)$. We are given $\mathcal{F} : (\mathcal{Z} \times \mathcal{Y}) \to \mathbb{R}$. We are given a dataset $D = \{(z_t, y_t)\}_{t=1}^T$ where $z_t \sim \mathcal{D}_t = \mathcal{D}_t(z_{1:t-1}, y_{1:t-1})$ and $y_t \sim p(\cdot \mid z_t)$. Fix $\delta \in (0, 1)$, assume $|\mathcal{F}| < \infty$ and there exists $f^\star \in \mathcal{F}$ such that $f^\star(y \mid z) = p(y \mid z)$. Then, with probability at least $1 - \delta$,*

$$\sum_{t=1}^{T}\mathbb{E}_{z\sim\rho_t}\left\|\widehat{f}(\cdot \mid z) - f^\star(\cdot \mid z)\right\|_{\mathrm{tv}}^2 \leq 2\log(|\mathcal{F}|/\delta),$$

*where*

$$\widehat{f} \in \arg\max_{f\in\mathcal{F}}\sum_{t=1}^{T}\log f(y_t \mid z_t).$$

Recall the notation that each model $M \in \mathcal{M}$ defines a conditional probability distribution of the form $M(r, x' \mid x, a)$. We apply the above generalization bound to our setting to obtain the following.

**Corollary F.2.** *Under [Assumption F.3](#), we have that the model-based maximum likelihood estimator of Equation [(38)](#) satisfies*

$$\forall h \in [H]: \ \mathbb{E}_{\mu_h^{(1:n)}}\left[\left\|\widehat{M}_h(\cdot \mid x_h, a_h) - M_h^\star(\cdot \mid x_h, a_h)\right\|_{\mathrm{tv}}^2\right] \leq 2\frac{\log(|\mathcal{M}|H/\delta)}{n}.$$

Note that we have taken an extra union bound over $H$ so that the MLE succeeds at each layer.

This is easily seen to imply an error bound between the associated Bellman operators.

**Corollary F.3.** *Let $[\widehat{\mathcal{T}}f](x,a) = \mathbb{E}_{(r,x')\sim\widehat{M}(\cdot|x,a)}[r + \max_{a'} f(x',a')]$ denote the Bellman optimality operator of $\widehat{M}$, and $\mathcal{T}$ denote the Bellman optimality operator of $M^\star$. Then we have that for all $h \in [H]$ and for all $f: \mathcal{X} \times \mathcal{A} \to [0,1]$,*

$$\mathbb{E}_{\mu_h^{(1:n)}}\left[\left([\widehat{\mathcal{T}}_h f](x_h, a_h) - [\mathcal{T}_h f](x_h, a_h)\right)^2\right] \leq 8\frac{\log(|\mathcal{M}|H/\delta)}{n}.$$

**Proof of [Corollary F.3](#).** Notice that

$$[\widehat{\mathcal{T}}_h f](x,a) - [\mathcal{T}_h f](x,a) = \mathbb{E}_{(r,x')\sim\widehat{M}(x,a)}\left[r + \max_{a'} f(x',a')\right] - \mathbb{E}_{(r,x')\sim M^\star(x,a)}\left[r + \max_{a'} f(x',a')\right]$$

$$\leq 2\left\|\widehat{M}_h(\cdot \mid x,a) - M_h^\star(\cdot \mid x,a)\right\|_{\mathrm{tv}}.$$

$\square$

**Theorem F.5.** *The model-based MLE algorithm is* CC-*bounded under [Assumption F.3](#) for all $\gamma > 0$ simultaneously, with scaling functions $\mathfrak{a}_\gamma = \frac{6}{\gamma}$ and $\mathfrak{b}_\gamma = 8\log(|\mathcal{M}|H/\delta)\gamma$. As a consequence, when invoked within* $H_2O$, *we have* $\mathbf{Risk} \leq \widetilde{O}\big(H\sqrt{(C_\star+C_{\mathrm{cov}})\log(|\mathcal{M}|H\delta^{-1})/T}\big)$ *with probability at least $1 - \delta T$.*

**Proof of [Theorem F.5](#).** We note that [Corollary F.2](#) implies a squared Bellman error bound for $Q_{\widehat{M}}^\star$, the optimal value function for $\widehat{M}$, since

$$\forall x, a: \ \left([\widehat{\mathcal{T}}_h Q_{\widehat{M},h+1}^\star](x,a) - [\mathcal{T}_h Q_{\widehat{M},h+1}^\star](x,a)\right)^2 = \left(Q_{\widehat{M},h}^\star(x,a) - [\mathcal{T}_h Q_{\widehat{M},h+1}^\star](x,a)\right)^2,$$

by the optimality equation for $Q_{\widehat{M}}^\star$. Thus we have

$$\forall h \in [H]: \ \mathbb{E}_{\mu_h^{(1:n)}}\left[\left(Q_{\widehat{M},h}^\star(x_h,a_h) - [\mathcal{T}_h Q_{\widehat{M},h+1}^\star](x_h,a_h)\right)^2\right] \leq 8\frac{\log(|\mathcal{M}|H/\delta)}{n}. \tag{55}$$

This is enough to repeat the proof of CC-boundedness for FQI ([Theorem F.3](#)), with $Q_{\widehat{M}}^\star$ taking the place of the FQI solution $\widehat{f}$. Indeed, the only algorithmic property that we used for FQI was [Lemma F.8](#), which also holds for $Q_{\widehat{M}}^\star$ by Equation [(55)](#). Tracking the slightly different constants resulting gives us the desired values for $\mathfrak{a}_\gamma$ and $\mathfrak{b}_\gamma$. $\square$

