# OpenReview forum: "Harnessing Density Ratios for Online Reinforcement Learning"
_ICLR.cc/2024/Conference — ICLR 2024 spotlight_

### Official Review · Reviewer_xDdp · 2023-10-30

**Soundness:** 3 good
**Presentation:** 3 good
**Contribution:** 3 good
**Rating:** 8
**Confidence:** 2

**Summary:**

This paper provides theoretical results adapting the idea of density ratio modeling from offline RL to the setting of online RL and provides sample complexity guarantees for the proposed algorithms. The main contribution of the paper is the algorithm called GLOW which provides polynomial sample complexity bounds in the online RL setting by minimizing the weighted average Bellman error using clipped density ratios along with realizability assumptions in function approximation. To further improve the sample efficiency, authors propose a meta-algorithm H2O that can reuse any offline algorithm to output an online exploration policy in the Hybrid RL setting (i.e. online RL with some available offline data).



*I would also like to note here that I am perhaps not the target audience of this work and though I am aware of some of the related works on this topic, I may not be the best judge of this paper's contributions.*

**Strengths:**

1. This paper exhaustively compares to the relevant prior work in the field and positions the contributions of this work with the literature in offline and Hybrid RL. In several parts of the paper, authors intuitively explain some of the challenges faced in adapting the density ratio modeling to online RL and the need for certain regularization formulations to achieve desired approximation error bounds. These add to the readability of the paper.

2. This paper claims to be the first to provide theoretical guarantees showing that value function realizability and density ratio realizability alone are sufficient for sample-efficient RL under coverability. It is a step towards unifying some of the advances made in offline RL with the requirements of sample-efficient online RL algorithms.

**Weaknesses:**

While the paper focuses on providing theoretical bounds regarding sample efficiency of the proposed algorithms, it lacks any attempts at empirical verification of the same. Comparing to Song et al. which the authors highlight for introducing the Hybrid RL setting, I would expect similar experiments (eg. Song et al. present experimental results in Montezuma's revenge) in this paper to demonstrate the performance of the theoretically motivated framework in practice. Perhaps it would also help to include a discussion of the practicality of the assumptions made in the theoretical proofs and what, if any, are the challenges of implementing the proposed algorithms in standard RL benchmarks.

**Questions:**

1. Why did the authors not consider empirical verification of their proposed framework in Sec 4? Is there any understanding of the validity of the assumptions made in the paper when applied to RL algorithms in practice?

2. In the first line on page 4, "that $\frac{d^\pi_h}{d^{\pi'}_h}$ for all ...", is there some missing text after $\frac{d^\pi_h}{d^{\pi'}_h}$?

---

> ### Author Response · Authors · 2023-11-22
> **Reply to Reviewer xDdp**
>
> We thank the reviewer for their positive comments and for their feedback. We reply to their questions and comments below.
>
> >While the paper focuses on providing theoretical bounds regarding sample efficiency of the proposed algorithms, it lacks any attempts at empirical verification of the same.
>
> As the reviewer recognized, this is a theoretical paper, with a focus on getting a statistically efficient algorithm for the online RL setting under only realizability-type assumptions. We believe these theoretical contributions alone are sufficient for publication. Our results for the Hybrid RL setting do provide some computational efficiency results, however few RL algorithms with rigorous guarantees can be implemented as-is in complex environments, and often need nontrivial adaptations to implement (e.g. the practical algorithm for HyQ given in Algorithm 3 in Song. et. al. 2022 [1] is quite different from its theoretical version given in Algorithm 1).
>
>  > Comparing to Song et al. which the authors highlight for introducing the Hybrid RL setting, I would expect similar experiments (eg. Song et al. present experimental results in Montezuma's revenge) in this paper to demonstrate the performance of the theoretically motivated framework in practice.
>
> While the paper of Song et al. 2022 was entirely focused on the hybrid RL setting and on demonstrating that hybrid RL can be computationally efficient, by contrast we have focused on providing both statistical results in the online setting and computational results in the hybrid setting. Thus, we consider the empirical implementation of HyGlow out of the scope of our current work. We are optimistic that our HyGlow algorithm will work similarly to the HyQ algorithm of Song. et. al. 2022, although verifying this in practice is left for future research.
>
> > Perhaps it would also help to include a discussion of the practicality of the assumptions made in the theoretical proofs
>
> Theoretically, our assumptions are quite general and capture many prior settings (see Appendix B and D.1). We also argue that we can expect them to hold in reasonable practical problems via the following heuristic argument. The coverability assumption states that there exists some unknown data distribution which provides coverage over our policy set. Should we be provided with this distribution, offline RL algorithms are known to work in conjunction with our function approximation assumptions. Thus, when our assumptions do not hold, this states that offline RL can not solve this MDP under any possible training distribution (even the most favorable one). This indicates that if the coverability assumption does not hold, the MDP is “degenerate” in the sense that sampling data from any possible data distribution and training on that data will always fail. We will expand the discussion in the text on the practicality of our assumptions.
>
> >In the first line on page 4, "that d^π/ d^{π′} for all ...", is there some missing text after d^π/d^{π′}?
>
> We have fixed the typo, thank you for pointing it out. It should have said “we require realizability of $d^\pi_h/d^{\pi’}_h$ for all $\pi, \pi’ \in \Pi$.”
>
> References:
> [1] Song, Yuda, Yifei Zhou, Ayush Sekhari, J. Andrew Bagnell, Akshay Krishnamurthy, and Wen Sun. "Hybrid rl: Using both offline and online data can make rl efficient." arXiv preprint arXiv:2210.06718 (2022).

---

### Official Review · Reviewer_reZc · 2023-10-31

**Soundness:** 3 good
**Presentation:** 3 good
**Contribution:** 3 good
**Rating:** 6
**Confidence:** 3

**Summary:**

This paper endeavors to establish a bridge between offline and online reinforcement learning. More specifically, it introduces a novel algorithm named GLOW, which excels at conducting sample-efficient online exploration while assuming the presence of value-function and density-ratio realizability. Furthermore, the authors extend these findings to encompass the hybrid reinforcement learning setting.

**Strengths:**

This paper is well-crafted, presenting its content in a clear and easily understandable manner. The majority of the results are extensively scrutinized and discussed, adding to the paper's overall quality. The introduction of a new density-based online algorithm effectively bridges the gap between offline and online RL. Furthermore, the paper's concerted effort to tackle the enduring and vital issue of removing the completeness assumption is a critical contribution to the field. This problem has long been a pressing concern in online reinforcement learning and is integral to the significance of this work.

**Weaknesses:**

- Bypassing the need for strong completeness-type assumptions by introducing an additional density ratio realizability assumption is not a novel approach, particularly in the context of offline reinforcement learning (see [Zhan et al., 2021] and a missing related work [1]). So the results in this work are not very surprising.

- Dentity-based algorithms may not be as suitable for online reinforcement learning, given the perceived strength of Assumption 2.2. While the authors have included two examples in Appendix D.1, these examples are already familiar to us and do not pose a significant challenge. If the authors could provide additional examples, such as those involving low (Bellman) eluder dimension or bilinear class, it would make the results more remarkable.

[1] Importance Weighted Actor-Critic for Optimal Conservative Offline Reinforcement Learning. Hanlin Zhu, Paria Rashidinejad, Jiantao Jiao

**Questions:**

see the Weakness part.

---

> ### Author Response · Authors · 2023-11-22
> **Reply to Reviewer reZc**
>
> We thank the reviewer for their positive comments and for their feedback. We reply to their questions and comments below.
>
> > Bypassing the need for strong completeness-type assumptions by introducing an additional density ratio realizability assumption is not a novel approach, particularly in the context of offline reinforcement learning [...]. So the results in this work are not very surprising.
>
> We disagree with the claim that our results are not very surprising. The reviewer seems to be glancing over the distinction between online RL and offline RL. As we have discussed in the text, density ratios have indeed appeared and been fruitful in offline RL:  “Notably, a promising emerging paradigm in offline RL makes use of the ability to model density ratios [...] for the underlying MDP.” (pg 1). However, in offline RL, the learner is provided with a dataset that by assumption has good coverage properties. This assumption is essential to the very existence of density ratios.  The key challenge in adopting these methods for online RL is that one is not given such a dataset and any dataset collected via exploration can not be assumed to have good coverage. Dealing with these issues requires us to introduce novel algorithmic ideas and new analysis techniques. To our knowledge, our paper is the *only* work that shows that density ratios have provable benefits in online RL.
>
> > Dentity-based algorithms may not be as suitable for online reinforcement learning, given the perceived strength of Assumption 2.2. While the authors have included two examples in Appendix D.1, these examples are already familiar to us and do not pose a significant challenge.
>
> We would like to push back on the notion that density ratios are not suitable for online RL. In particular, our setting subsumes many specific examples that satisfy our realizability assumptions and could not be captured by prior work. To this end, we disagree with the claim that “these examples are familiar and do not pose a challenge”: Example D.2 is novel – we show for the first time that we can address the problem of rich observation RL when the latent MDP is coverable, and in which we only make assumptions about the latent function classes. In particular, this setting is not guaranteed to satisfy the Bellman completeness assumption used in Xie et al. (2023) (notably, completeness in latent space does not imply completeness in observation space), and hence is out of the reach of prior work. We are happy to expand the discussion around the novelty of Example D.2 in the final version of the paper.
> Regarding the strength of assumption 2.2, we believe that the reviewer is referring to our need to realize the density ratios for all pairs of policies. We remark that this assumption is weaker than needing to realize the density ratios between all policies and any reference distribution (Remark B.2 in the appendix).  In addition, note that in the online RL setting, a coverability assumption with respect to only a single policy (e.g. pi^\star) is easily seen to be useless (this assumption would state that there exists an unknown distribution which covers pi^\star, which is always trivially satisfied by the distribution for pi^\star itself).
>
> >  If the authors could provide additional examples, such as those involving low (Bellman) eluder dimension or bilinear class, it would make the results more remarkable.
>
> The low Bellman eluder dimension condition and the bilinear class condition are orthogonal to our setting. That is, they do not imply and are not implied by coverability + our realizability assumptions (this is discussed in Appendix B, as well as in the prior work [1]).
>
> References:
> [1] Tengyang Xie, Dylan J Foster, Yu Bai, Nan Jiang, and Sham M Kakade. The role of coverage in online reinforcement learning. International Conference on Learning Representations, 2023.

---

> > ### Comment · Reviewer_reZc · 2023-11-22
> >
> > Thank you for your detailed response. After reading the author's response and the comments from other reviewers, I decided to maintain the positive score.

---

### Official Review · Reviewer_4ukw · 2023-11-08

**Soundness:** 4 excellent
**Presentation:** 3 good
**Contribution:** 3 good
**Rating:** 6
**Confidence:** 3

**Summary:**

The convergence of offline and online reinforcement learning theories has led to the emergence of density ratio modeling in offline RL. However, this concept has been absent in online RL due to the challenge of collecting a comprehensive exploratory dataset. This paper introduces GLOW, an online RL algorithm that can explore and learn efficiently when there is an exploratory distribution with good coverage (coverability condition), even in the presence of value functions and density ratios. GLOW handles unbounded density ratios through truncation and employs optimism for exploration, but it is computationally inefficient. To address this, a more efficient variant called HYGLOW is introduced for Hybrid RL, combining online RL with offline data. HYGLOW is derived from a meta-algorithm called $H_2O$, which offers a provable reduction from hybrid RL to offline RL.

**Strengths:**

1. The paper is fairly well organized and the problem is well motivated (to analyze the notion of density ratio in online RL).

2. The theoretical guarantee is solid and the explanation of concepts is also very comprehensive.

**Weaknesses:**

While the paper primarily delves into theoretical aspects and introduces the GLOW algorithm with statistical guarantees, it's worth noting that GLOW is computationally inefficient. Therefore, there is a need to develop more computationally efficient variants to showcase the algorithm's practical effectiveness.

**Questions:**

The idea of using density ratios in online RL is new. However, I am still curious about the specific benefits for online RL. Does it enlarge range of MDPs which can be solved by online RL algorithms (it may be computationally inefficient)? What is the advantage of using density ratios for guiding exploration in online RL compared to other classical methods like Upper Confidence Bound ?

---

> ### Author Response · Authors · 2023-11-22
> **Reply to Reviewer 4ukw**
>
> We thank the reviewer for their positive comments and for their feedback. We reply to their questions and comments below.
>
> >  …it's worth noting that GLOW is computationally inefficient. Therefore, there is a need to develop more computationally efficient variants to showcase the algorithm's practical effectiveness.
>
> We believe that our statistical efficiency results on their own are a contribution which merits acceptance. Statistical considerations preclude computational ones, and a major contribution of our work is to establish for the first time that the use of density ratios in online RL is viable and worthy of further study (including computational questions). Computational inefficiency is overwhelmingly common for online RL algorithms under general function approximation (e.g. OLIVE [1], GOLF [2], BiLin-UCB [3], etc.), and our work builds on this line of research, though deriving an efficient counterpart to our algorithm is of course a fascinating direction for future research.
>
> Towards bridging this gap, we also provided new results for the Hybrid RL setting (Section 4) which do yield a computationally efficient version of GLOW, under the additional assumption that the learner has access to both offline datasets and online interactions. In this setting, we can remove the optimism step, which is the only computationally intensive component of our algorithm.
>
> A computationally efficient algorithm in the fully online setting would require algorithmic ideas beyond optimism (e.g. policy covers). These exist in simpler settings (namely tabular and linear MDPs) but are not known to work in the general setting of coverability with arbitrary function classes.
>
> > However, I am still curious about the specific benefits for online RL. Does it enlarge range of MDPs which can be solved by online RL algorithms (it may be computationally inefficient)?
>
> Yes, this does indeed enlarge the class of MDPs which we can solve in online RL. Specifically, this is the first provable sample complexity guarantee for coverable MDPs that does not rely on the Bellman Completeness assumption, which was specifically mentioned as an open problem in Xie et al. 2023 [4]. In Appendix D.1 we also provide some more specific examples of new MDP classes that can be solved by our algorithm and which were not previously known to be tractable.
>
> References:
> [1] Jiang N, Krishnamurthy A, Agarwal A, Langford J, Schapire RE. Contextual decision processes with low bellman rank are pac-learnable. InInternational Conference on Machine Learning 2017 Jul 17 (pp. 1704-1713). PMLR.
>
> [2] Jin C, Liu Q, Miryoosefi S. Bellman eluder dimension: New rich classes of rl problems, and sample-efficient algorithms. Advances in neural information processing systems. 2021 Dec 6;34:13406-18.
>
> [3] Du S, Kakade S, Lee J, Lovett S, Mahajan G, Sun W, Wang R. Bilinear classes: A structural framework for provable generalization in rl. InInternational Conference on Machine Learning 2021 Jul 1 (pp. 2826-2836). PMLR.
>
> [4] Xie, Tengyang, Dylan J. Foster, Yu Bai, Nan Jiang, and Sham M. Kakade. "The role of coverage in online reinforcement learning." ICLR 2023.

---

### Meta-Review · Area_Chair_RxbJ · 2023-12-09

**Metareview:**

a) claims: The paper proposes an algorithm for online RL with provable statistical guarantees that makes use of density ratios, an approach that has previously proven useful in offline RL.  A more computationally efficient version for Hybrid RL is also presented.

b) strengths: All the reviews noted that this was clear progress on a long standing and important issue.

c) weaknesses: There were some concerns about empirical efficiency and evaluation.

**Justification For Why Not Higher Score:**

n/a

**Justification For Why Not Lower Score:**

The reviewers all agreed that this is a clearly written paper that makes solid theoretical progress on an important problem.

---

### Decision · Program_Chairs · 2024-01-16

Accept (spotlight)